

# Implementation and evaluation of the GEOS-Chem chemistry module version 13.1.2 within the Community Earth System Model v2.1

Thibaud M. Fritz[1], Sebastian D. Eastham[1,2]*, Louisa K. Emmons[3], Haipeng Lin[4], Elizabeth W. Lundgren[4], Steve Goldhaber[3], Steven R. H. Barrett[1,2], Daniel J. Jacob[4]

[1]Laboratory for Aviation and the Environment, Department of Aeronautics and Astronautics, Massachusetts Institute of Technology, Cambridge, MA 02139, USA

[2]Joint Program on the Science and Policy of Global Change, Massachusetts Institute of Technology, Cambridge, MA 02139, USA

[3]Atmospheric Chemistry Observations and Modeling Laboratory, National Center for Atmospheric Research, Boulder, CO, USA

[4]John A. Paulson School of Engineering and Applied Sciences, Harvard University, Cambridge, MA 02138, USA

*Correspondence to: Sebastian D. Eastham (seastham@mit.edu)

**Short summary**. We bring the state-of-the-science chemistry module GEOS-Chem into the Community Earth System Model (CESM). We show that some known differences between results from GEOS-Chem and CESM's CAM-chem chemistry module may be due to the configuration of model meteorology rather than inherent differences in the model chemistry. This is a significant step towards a truly modular ESM and allows two strong but currently separate research communities to benefit from each other's advances.

**Abstract.** We implement the GEOS-Chem chemistry module as a chemical mechanism in the Community Earth System Model version 2 (CESM). Our implementation allows the state-of-the-science GEOS-Chem chemistry module to be used with identical emissions, meteorology, and climate feedbacks as the CAM-chem chemistry module within CESM. We use coupling interfaces to allow GEOS-Chem to operate almost unchanged within CESM. Aerosols are converted at each time step between the GEOS-Chem bulk representation and the size-resolved representation of CESM's Modal Aerosol Model (MAM4). Land type information needed for dry deposition calculations in GEOS-Chem is communicated through a coupler, allowing online land-atmosphere interactions. Wet scavenging in GEOS-Chem is replaced with the Neu and Prather scheme, and a common emissions approach is developed for both CAM-chem and GEOS-Chem in CESM.

We compare how GEOS-Chem embedded in CESM (C-GC) compares to the existing CAM-chem chemistry option (C-CC) when used to simulate atmospheric chemistry in 2016, with identical meteorology and emissions. We compare atmospheric composition and deposition tendencies between the two simulations and evaluate the residual differences between C-GC compared to its use as a standalone chemistry transport model (S-GC). We find that stratospheric ozone agrees well between the three models with differences of less than 10% in the core of the ozone layer, but that ozone at lower altitudes is generally lower in C-GC than in either C-CC or S-GC due to greater tropospheric concentrations of bromine. This difference is not





uniform, with C-GC ozone 30% lower in the southern hemisphere than in S-GC but within 10% in the northern hemisphere,
suggesting differences in the effects of anthropogenic emissions. Aerosol concentrations in C-GC agree with those in S-GC at
low altitudes in the tropics but are over 100% greater in the upper troposphere due to differences in the representation of
convective scavenging. We also find that water vapor concentrations vary substantially between the standalone and CESM-
implemented version of GEOS-Chem, as the simulated hydrological cycle in CESM diverges from that represented in the
source MERRA-2 meteorology.
Our implementation of GEOS-Chem as a chemistry option in CESM (including full chemistry-climate feedbacks) is publicly
available and is being considered for inclusion in the CESM main code repository. This work is a significant step in the MUlti-
Scale Infrastructure for Chemistry and Aerosols (MUSICA) project, enabling two communities of atmospheric researchers
(CESM and GEOS-Chem) to share expertise through a common modeling framework and thereby accelerate progress in
atmospheric science.

## 1    Introduction

Accurate representation and understanding of atmospheric chemistry in global Earth System Models (ESMs) has been
recognized as an urgent priority in geoscientific model development. The National Research Council (NRC) report on a
National Strategy for Advancing Climate Modeling (Bretherton et al., 2012) stresses the need for including comprehensive
atmospheric chemistry in the next generation of ESMs. The NRC report on the Future of Atmospheric Chemistry (NRC, 2016)
identifies the integration of atmospheric chemistry into weather and climate models as one of its five priority science areas.
This work responds to those needs, presenting the implementation of the state-of-science model GEOS-Chem as an
atmospheric chemistry module within the Community Earth System Model (CESM).
GEOS-Chem is a state-of-the-science global atmospheric chemistry model developed and used by over 150 research groups
worldwide (http://geos-chem.org). It has wide appeal among atmospheric chemists because it is a comprehensive, state-of-
science, open-access, well-documented modeling resource that is easy to use and modify but also has strong central
management, version control, and user support. The model is managed at Harvard by a GEOS-Chem Support Team with
oversight from an international GEOS-Chem Steering Committee. Documentation and communication with users is done
through extensive web and wiki pages, email lists, newsletters, and benchmarking. Grass-roots model development is done by
users, and inclusion into the standard model is prioritized by Working Groups reporting to the Steering Committee. The model
can simulate tropospheric and stratospheric oxidant-aerosol chemistry, aerosol microphysics, and budgets of various gases .
Simulations can be conducted on a wide range of computing platforms with either shared-memory (OpenMP) or distributed
memory (MPI) parallelization – with this latter implementation referred to as GEOS-Chem High Performance, or GCHP
(Eastham et al., 2018).






For the general atmospheric chemistry problem involving $K$ atmospheric species coupled by chemistry and/or aerosol
microphysics, GEOS-Chem solves the system of $K$ coupled continuity equations

$$\frac{\partial n_i}{\partial t} = -\nabla \cdot (n_i \mathbf{U}) + P_i(\mathbf{n}) - L_i(\mathbf{n}) \tag{1}$$



where $\mathbf{n} = (n_1, \ldots n_K)^T$ is the number density vector representing the concentrations of the $K$ species, $\mathbf{U}$ is the 3-D wind vector,
and $P_i$ and $L_i$ are local production and loss terms for species $i$ including emissions, deposition, chemistry, and aerosol physics.
The transport term $-\nabla \cdot (n_i \mathbf{U})$ includes advection by grid-resolved winds as well as parameterized subgrid turbulent motions
(boundary layer mixing, convection). The local term $P_i(\mathbf{n})$ –$L_i(\mathbf{n})$ couples the continuity equations across species through
chemical kinetics and aerosol physics.

Standard application of the GEOS-Chem model as originally described by Bey et al. (2001) is off-line, meaning that the model
does not simulate its own atmospheric dynamics. Instead, it uses winds and other meteorological variables archived from the
Goddard Earth Observation System (GEOS) of the NASA Global Modeling and Assimilation Office (GMAO). These archives
are produced by GEOS ESM simulations with assimilated meteorological observations, currently at a horizontal resolution of
0.25°×0.3125°. GEOS-Chem simulations can be conducted at that native resolution or at coarser resolution (by conservative
re-gridding of meteorological fields). Long et al. (2015) developed an on-line capability for GEOS-Chem to be used as a
chemical module in ESMs, with initial application to the GEOS ESM.  In that configuration, GEOS-Chem only solves the
local terms of the continuity equation

$$\frac{\partial n_i}{\partial t} = P_i(\mathbf{n}) - L_i(\mathbf{n}) \tag{2}$$



and delivers the updated concentrations to the ESM for computation of transport through its atmospheric dynamics. On-line
simulation avoids the need for a meteorological data archive and the associated model transport errors (Jöckel et al., 2001; Yu
et al., 2018). It also enables fast coupling between chemistry and dynamics.

Transformation of GEOS-Chem to a grid-independent structure was performed transparently, such that the standard GEOS-
Chem model uses the exact same code for on-line and off-line applications. This includes a mature implementation within the
GEOS ESM. It was applied recently to a year-long tropospheric chemistry simulation with ≈12 km (cubed-sphere c720) global
resolution, and is now being used for global air quality forecasting and chemical data assimilation (Keller et al., 2017; Hu et
al., 2018; Keller et al., 2021). However, the only implementations of GEOS-Chem which are currently publicly available are
either designed to run "offline", driven by archived meteorological data from the NASA Goddard Earth Observing System



(GEOS) (Bey et al., 2001; Eastham et al., 2018), or operate at regional scale and do not extend to global simulation (Lin et al.,
2020; Feng et al., 2021).

Integration of GEOS-Chem as a chemistry option within an open-access, global ESM responds to the aforementioned calls
from the NRC. One of the most widely used open-access ESM is the Community Earth System Model (CESM) (Hurrell et al.,
2013). CESM is fully coupled and state-of-science. It produces its own meteorology based on fixed sea surface temperatures
or with a fully interactive ocean model. It can also be nudged to observed meteorology including from GEOS. The CESM
configuration with chemistry covering the troposphere and stratosphere is referred to as CAM-chem (Community Atmosphere
Model with chemistry) (Tilmes et al., 2016; Lamarque et al., 2012). CAM-chem is a state-of-science model of atmospheric
chemistry; it has participated (along with CESM's WACCM model which extends to the lower thermosphere) in many
international model intercomparison activities such as ACCMIP, CCMI, POLMIP, HTAP2, GeoMIP and CMIP6, and has a
large international user community. CAM-chem also has a very different development heritage from GEOS-Chem, with each
model providing better performance in comparison to observations in different areas (Park et al., 2021; Emmons et al., 2015;
Nicely et al., 2017; Jonson et al., 2018). The fundamental differences in implementation of almost every atmospheric process
between GEOS-Chem and CAM-chem mean that it is difficult to disentangle the root causes of these differences.

Modular Earth system models can resolve this issue. Allowing individual scientific components to be swapped freely allows
researchers to evaluate exactly what effect that component has in isolation, while also giving a single user base access to a
larger portfolio of options. If two different models each implement five processes in different ways, a researcher must learn to
use both in order to compare their results and cannot isolate the effect of any one process with confidence. If process options
are implemented in the same framework, this problem is avoided. Such modularity is becoming increasingly possible with the
availability of Earth system infrastructure such as the Earth System Modeling Framework (ESMF) and the National Unified
Operational Prediction Capability (NUOPC), which describe common interfaces for Earth system modeling components (Hill
et al., 2004; Sandgathe et al., 2011). The Multi-Scale Infrastructure for Chemistry and Aerosols (MUSICA) builds upon this
trend with process-level modularization, with the goal of allowing researchers to select from a range of community-developed
options when performing atmospheric simulations.

This work integrates the GEOS-Chem chemistry module into CESM as an alternative option to CAM-chem. Our
implementation allows researchers to select either model to simulate gas-phase and aerosol chemistry throughout the
troposphere and stratosphere, while other processes such as advection, broadband radiative transfer, convective transport, and
emissions are handled nearly identically. We demonstrate this capability by comparing simulations of the year 2016 as
generated by GEOS-Chem and CAM-chem operating within CESM, with the chemical module being the only difference.
Estimates of atmospheric composition are compared between the two models and against a simulation in the standalone GEOS-





Chem High Performance (GCHP) chemistry transport model (CTM). Finally, we evaluate the accuracy of the three approaches
against observations of atmospheric composition and deposition.

Section 2 provides a technical description of the implementation of GEOS-Chem into CESM. Section 3 then describes a two-
year simulation performed in CESM with GEOS-Chem; CESM with CAM-chem; and the standalone GEOS-Chem CTM. This
includes model setup (Section 3.1), intercomparison (Section 3.2), and evaluation against surface and satellite measurements
(Section 3.3).

## 138   2      Coupling between GEOS-Chem and CESM

We first describe the interface used within CESM when using either the CAM-chem or GEOS-Chem options (Section 2.1).
Unless otherwise stated, "GEOS-Chem" refers to the grid independent chemistry module which is common to all
implementations, including standalone GEOS-Chem with OpenMP (Classic) or MPI (GCHP) parallelization, NASA GMAO's
GEOS ESM, and WRF-GC. We then briefly summarize the chemistry and processes represented by the CAM-chem and
GEOS-Chem options within CESM (Section 2.2). This is followed by a description of differences between the implementation
of GEOS-Chem in CESM and its stand-alone code (Section 2.3), differences in the data flow through CESM when using
GEOS-Chem as opposed to CAM-chem (Section 2.4), and finally the installation and compilation process (Section 2.5).

### 146   2.1    Interface

Our approach embeds a full copy of the GEOS-Chem chemistry module source code (version 13.1.2) within CESM (version
2.1.1). All modifications made to the GEOS-Chem source code have been propagated to the GEOS-Chem main code branch
(https://github.com/geoschem/geos-chem) to ensure future compatibility between CESM and GEOS-Chem. Information is
passed between the CESM Community Atmosphere Model (CAM) version 6 (CAM6) and the GEOS-Chem routines through
an interface layer developed as part of this work. A schematic representation of the implementation is provided in Figure 1.






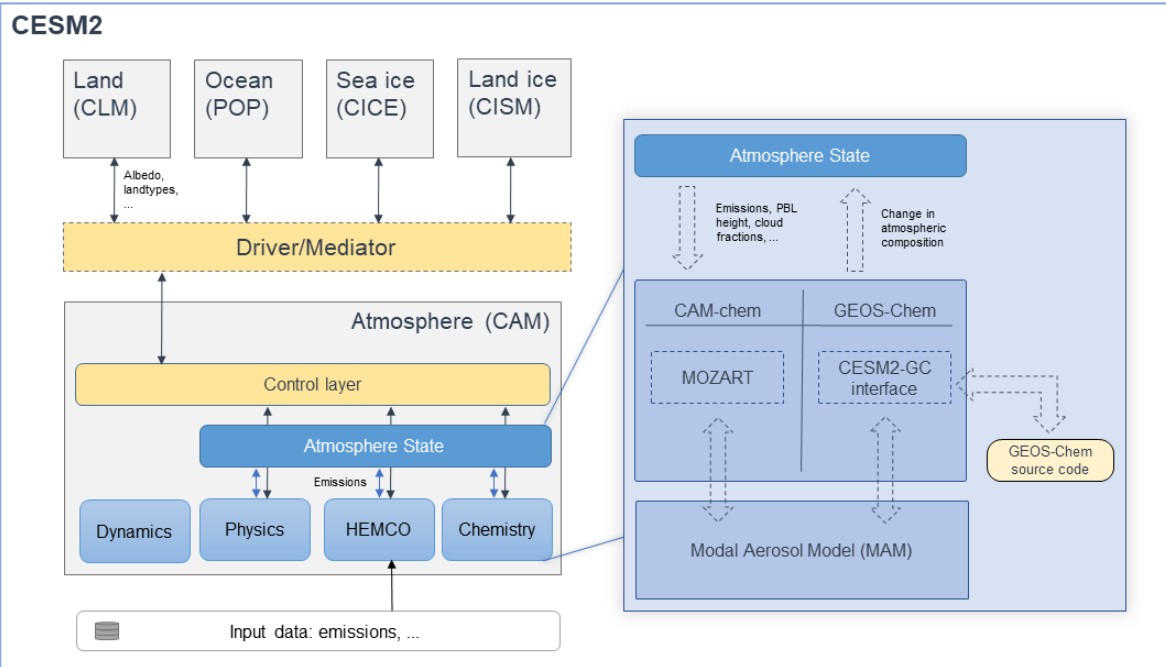


*Figure 1. Architectural overview of CESM when running with either the GEOS-Chem or CAM-chem chemistry options. The left section shows the architecture of CESM, where the five major Earth system components are connected through the driver/mediator. The work presented here changes only the contents of the atmosphere component (CAM). Regardless of the chemistry option used, dynamics, physics, and emissions (HEMCO) are handled identically. Each component modifies the "Atmosphere State" while communication occurs through the control layer. The choice of chemistry module is confined to the "Chemistry" subcomponent, where either CAM-chem or GEOS-Chem can be chosen. In each case, data are transmitted between the "Atmosphere State" and the chemistry module, which interacts in turn with the Modal Aerosol Model.*

At each time step, CESM calls the coupling interface which fills in the meteorological variables required by either CAM-chem or GEOS-Chem. Atmospheric transport and physics are identical whether using CAM-chem or GEOS-Chem to simulate atmospheric chemistry. The interface passes species concentrations from CAM to GEOS-Chem, which are then modified by GEOS-Chem and passed back to CAM. Meteorological data and land data are also passed to GEOS-Chem through the same interface. The routine calls in CAM when using either GEOS-Chem or CAM-chem are identical, with the appropriate chemistry module defined at compilation time such that the calls are routed to the appropriate routines.

The interface handles the conversion of meteorological variables and concentrations of atmospheric constituents between the state variables in CAM and those used in GEOS-Chem. Since GEOS-Chem operates in a "grid-independent" fashion, changes in the grid specification and other upstream modifications to CESM do not necessitate any changes to this interface (Long et al., 2015). Our version of CESM 2.1.1 is modified such that emissions are handled by the Harmonized Emissions Component (HEMCO), which operates independently of the chemistry module and can provide emissions data to either CAM-chem or GEOS-Chem equally (Lin et al., 2021).




The interface code is kept in the `src/chemistry/geoschem` subfolder, which also contains a copy of the source code for
GEOS-Chem. Unlike the implementation of GEOS-Chem within GEOS, we do not use ESMF. However, we plan to develop
a NUOPC-based interface as part of future work.
**2.2    Processes represented by CAM-chem and GEOS-Chem**
CAM-chem uses the Model for OZone And Related chemical Tracers (MOZART) family of chemical mechanisms to simulate
atmospheric chemistry (Emmons et al., 2020). The tropospheric-stratospheric MOZART-TS1 scheme which we demonstrate
in our intercomparison involves 186 gas-phase chemical species and includes stratospheric bromine, chlorine, and fluorine
chemistry. MOZART-TS1 does not include detailed tropospheric halogen chemistry or short-lived halogen sources such as
sea salt bromine, although these will be available in a future release (Badia et al., 2021; Fernandez et al., 2021). Photolysis
rates are calculated using a lookup table, based on calculations with the Tropospheric Ultraviolet and Visible (TUV) radiation
model (Kinnison et al., 2007). Wet deposition is calculated using the Neu and Prather (2012) scheme for both convective and
large-scale precipitation. Dry deposition velocities over land are calculated for each land type by the Community Land Model
(CLM) in CESM using the Wesely (1989) resistance scheme with updates described by Emmons et al. (2020). Deposition
velocities over the ocean are calculated separately in CAM-chem. Aerosols are represented using the 4-mode Modal Aerosol
Model (MAM4), which includes sulfate, black carbon, primary, and secondary organic aerosols (Mills et al., 2016).
Ammonium and ammonium nitrate aerosols are calculated with a parameterization using the bulk aerosol scheme (Tilmes et
al., 2016). Secondary organic aerosols are simulated using a 5-bin volatility basis set (VBS) scheme, formed from terpenes,
isoprene, specific aromatics and lumped alkanes through reaction with OH, $O_3$ and $NO_3$, with unique yields for each reach and
bin (Tilmes et al., 2019).  This more detailed scheme differs from the default MAM SOA scheme that is used in CAM6 (without
interactive chemistry). Aerosol deposition, including dry and wet deposition, and gravitational settling (throughout the
atmosphere) are calculated in the MAM code of CESM. CAM-chem also uses a volatility basis set (VBS) approach for SOA
with five volatility bins, covering saturation concentrations with logarithmic spacing from 0.01 to 100 $\mu g/m^3$. CAM-chem
explicitly represents Aitken and accumulation mode SOA using two separate tracers for each volatility bin but does not include
an explicit representation of non-volatile aerosol.

GEOS-Chem uses a set of chemical mechanisms implemented with the Kinetic PreProcessor (KPP) (Damian et al., 2002). The
standard chemical mechanism has evolved continuously from the tropospheric gas-phase scheme described by Bey et al. (2001)
and now includes aerosol chemistry (Park, 2004), stratospheric chemistry (Eastham et al., 2014), and a sophisticated
tropospheric-stratospheric halogen chemistry scheme (Wang et al., 2019). The scheme present in GEOS-Chem 13.1.2 includes
299 chemical species. Additional "specialty simulations" such as an aerosol-only option and a simulation of the global mercury
cycle are present in GEOS-Chem but are not implemented into CESM in this work. Photolysis rates are calculated using the
Fast-JX v7 model (Wild et al., 2000; Fast-JX v7.0a). When implemented standalone, wet deposition is calculated for large-





scale precipitation using separate approaches for water-soluble aerosols (Liu et al., 2001) and gases (Amos et al., 2012) with
calculation of convective scavenging performed inline with convective transport. A different approach is used to simulate wet
scavenging for the implementation of GEOS-Chem in CESM (see Section 2.3.4). Dry deposition is calculated using the Wesely
(1989) scheme (Wang et al., 1998), but with updates for $HNO_3$ (Jaeglé et al., 2018), aerosols (Jaeglé et al., 2011; Alexander
et al., 2005; Fairlie et al., 2007; Zhang et al., 2001), and over ocean (Pound et al., 2020). The representation of aerosols in
GEOS-Chem varies by species. Sulfate-ammonium-nitrate aerosol is represented using a bulk scheme (Park, 2004), with gas-
particle partitioning determined using ISORROPIA II (Fountoukis and Nenes, 2007). Modal and sectional size-resolved
aerosol schemes are available for GEOS-Chem (Kodros and Pierce, 2017; Yu and Luo, 2009), but are disabled by default and
not used in this work. Sea salt aerosol is represented using two (fine and coarse) modes (Jaeglé et al., 2011), while dust is
represented using four size bins (Fairlie et al., 2007). We use the "complex SOA" chemistry mechanism in GEOS-Chem when
running in CESM, as this uses a volatility basis set (VBS) representation of secondary organic aerosol which is broadly
compatible with that used in CAM-chem (Pye and Seinfeld, 2010; Marais et al., 2016; Pye et al., 2010). The complex SOA
VBS scheme uses four volatility bins covering saturation concentrations on a logarithmic scale from 0.1 to 100 µg/m$^3$. Two
classes of SOA are represented in this fashion: those derived from terpenes (TSOA) and those derived from aromatics (ASOA).
For each "class" of SOA, two tracers are used to represent each volatility bin (one holding the gas phase mass, the other holding
the condensed phase mass). The only exception is the lowest-volatility aromatic aerosol, which is considered to be non-volatile
and therefore has no gas-phase tracer. Two additional SOA tracers, representing isoprene-derived and glyoxal-derived SOA,
are not represented using a VBS approach.

Additional differences between the two chemistry modules include the use of different Henry's law coefficients, gravitational
settling schemes, representation of polar stratospheric clouds, and heterogeneous chemistry. Full descriptions of the two
models are available at https://geos-chem.seas.harvard.edu/narrative and in Emmons et al. (2020).
**2.3    Representation of atmospheric processes in GEOS-Chem when running in CESM**
Some processes cannot be easily transferred from standalone GEOS-Chem to its implementation in CESM, due to factors such
as the different splitting of convective transport in the two models. Processes which vary in their implementation between the
standalone and CESM implementations of GEOS-Chem are described below.
**2.3.1    Aerosol coupling in CESM with GEOS-Chem**
Since GEOS-Chem and CESM use different approaches to represent aerosols, there is no straight-forward translation between
the GEOS-Chem representation and that used elsewhere in CESM. We implement an interface between the CESM and GEOS-
Chem representations, so that GEOS-Chem's processing of aerosols is most accurately represented without compromising the
microphysical simulations and radiative interactions of aerosol calculated elsewhere in CESM.



CESM uses the 4-mode version of the Modal Aerosol Model (MAM4) to represent the aerosol size distribution and perform
aerosol microphysics (Liu et al., 2016). This represents the mass of sulfate aerosols, secondary organic matter (in five volatility
basis set bins), primary organic matter, black carbon, soil dust, and sea salt with advected tracers for each mode (accumulation,
Aitken, coarse, and primary carbon), although some species are considered only in a subset of the four modes. A tracer is also
implemented for the number of aerosol particles in each mode, resulting in a total of 18 tracers. As discussed above, GEOS-
Chem instead represents sulfate, nitrate, and ammonium aerosol constituents with three tracers; fresh and aged black and
organic carbon with four tracers; fine and coarse sea salt as two tracers; and different sizes of dust with four tracers. Six
additional tracers are used to track the bromine, iodine, and chlorine content of each mode of sea salt aerosol, with two more
used to track overall alkalinity. Gas-phase sulfuric acid is assumed to be negligible in the troposphere and is estimated using
an equilibrium calculation in the stratosphere (Eastham et al., 2014). The GEOS-Chem mechanism therefore represents greater
chemical complexity but reduced size resolution compared to the aerosol representation in MAM4.

Accordingly, when receiving species concentrations from CESM, the interface to GEOS-Chem lumps all modes of the MAM
aerosol into the corresponding GEOS-Chem tracer. This includes gas-phase $H_2SO_4$, in the case of the GEOS-Chem sulfate
(SO4) tracer. Aerosol constituents which are not represented explicitly by MAM (e.g. nitrates) are not included in this
calculation. The relative contribution of each mode is stored during this "lumping" process for each grid cell. Once calculations
with GEOS-Chem are complete, the updated concentration of the lumped aerosol is repartitioned into the MAM tracers based
on the stored relative contributions in each grid cell.

For secondary organic aerosols (SOA), additional steps are needed. For the bins covering saturation concentrations of 1 µg/m³
and greater, we assume that the relevant volatility bin in MAM4 is equal to the sum of the two classes in GEOS-Chem covering
the same saturation concentrations. For example, the tracers TSOA1 and ASOA1 in GEOS-Chem are combined to estimate
the total quantity of the Aitken and accumulation modes for species "soa3" in MAM4. Partitioning between the two modes
(when transferring from GEOS-Chem to MAM4) is calculated based on the relative contribution of each constituent to the
total prior to processing by GEOS-Chem. Partitioning between the two classes (when transferring from MAM4 to GEOS-
Chem) is calculated based on the relative contribution of each constituent to the total at the end of the previous time step. For
the lowest-volatility species, we split the lowest volatility bin concentrations (and non-volatile species) from GEOS-Chem
between the two lowest volatilities in MAM4. A full mapping for all species is provided in Table 1.

*Table 1. Mapping between tracers used to represent SOA in GEOS-Chem and CAM-chem (CESM). Translation between GEOS-Chem and*
*MAM4 is performed by preserving the relative contributions provided during the previous transfer.*

| GEOS-Chem species | Mapping to CAM-chem species | Saturation concentration range (µg/m³) | Phase |
|---|---|---|---|
| TSOA0 + ASOAN | soa1_a1 + soa1_a2 + soa2_a1 + soa2_a2 | 0 – 0.1 | Aerosol |
| TSOA1 + ASOA1 | soa3_a1 + soa3_a2 | 0.1 – 1.0 | Aerosol |



| TSOA2 + ASOA2 | soa4_a1 + soa4_a2 | 1.0 – 10 | Aerosol |
|---|---|---|---|
| TSOA3 + ASOA3 | soa5_a1 + soa5_a2 | 10 – 100 | Aerosol |
| TSOG0 | SOAG0 + SOAG1 | 0 – 0.1 | Gas |
| TSOG1 + ASOG1 | SOAG2 | 0.1 – 1.0 | Gas |
| TSOG2 + ASOG2 | SOAG3 | 1.0 – 10 | Gas |
| TSOG3 + ASOG3 | SOAG4 | 10 – 100 | Gas |

Finally, MAM simulates some chemical processing on and in the aerosol. This includes the reaction of sulfur dioxide with hydrogen peroxide and ozone in clouds, which is already included in the GEOS-Chem chemistry mechanism. We therefore disable in-cloud sulfur oxidation in MAM4 when using the GEOS-Chem chemistry component in CESM, consistent with the GEOS-Chem CTM. A comparison of the effect of each approach is provided in the Supplementary Information.

### 2.3.2 Dry deposition

Dry deposition velocities over land are calculated in CESM for each atmospheric constituent by the Community Land Model (CLM) using a species database stored by the coupler. GEOS-Chem is also able to calculate its own dry deposition velocities (see Section 2.2), in situations where a land model is not available such as when running as a CTM. We thus implement different options to compute dry deposition velocities when running CESM with the GEOS-Chem chemistry option:

1. Dry deposition velocities over land are computed by CLM and are passed to CAM through the coupler. They are then merged with dry deposition velocities computed over ocean and ice by GEOS-Chem, identical to the procedure used in CAM-chem. Each of these are scaled by the land and ocean/ice fraction respectively.

2. GEOS-Chem computes dry deposition at any location using the land types and leaf area indices from CLM, which are passed through the coupler.

3. GEOS-Chem obtains "offline" land types and leaf area indices and computes the dry deposition velocities similarly to GEOS-Chem Classic.

This allows researchers to experiment with different dry deposition options, ranging from that most consistent with the approach used in CAM-chem (option 1) to that most consistent with stand-alone GEOS-Chem (option 3). For this work we use option 2, but option 1 will be brought as standard into the CESM main code to reduce data transfer requirements.

### 2.3.3 Emissions

The Harmonized Emissions Component (HEMCO) is used to calculate emissions in standalone GEOS-Chem (Keller et al., 2014), and HEMCO v3.0 was recently implemented as an option for CAM-chem (Lin et al., 2021). HEMCO offers the possibility for the user to read, regrid, overlay, and scale emission fluxes from different archived emissions inventories at runtime. Emissions extensions allow for the computation of emissions that depend on meteorology or surface characteristics (e.g. lightning, dust emissions). Some extensions have also been designed to calculate subgrid-scale chemical processes, such as non-linear chemistry in ship plumes (Vinken et al., 2011).



297

The GEOS-Chem CTM implementations use archived ("offline") inventories of natural emissions, calculated at native resolution using the NASA GEOS MERRA-2 and GEOS-FP meteorological fields. This ensures that the emissions are calculated consistently regardless of grid resolution. These archived emissions fields can be used within CESM but we also preserve the option for users to employ "online" emissions inventories where relevant. This enables feedbacks between climate and emissions to be calculated. For instance, lightning $NO_x$ emissions, dust and sea salt emissions, and biogenic emissions are all computed online using parameterizations from CAM and CLM. CAM computes lightning $NO_x$ emissions based on the lightning flash frequency, which is estimated following the model cloud height, with different parameterizations over ocean and land. The NO lightning production rate in CAM is assumed proportional to the discharged energy, with $10^{17}$ atoms of nitrogen released per Joule (Price et al., 1997). The lightning $NO_x$ emissions are then allocated vertically from the surface to the local cloud top based on the distribution described by Pickering et al. (1998). For biogenic emissions, we use the online Model of Emissions of Gases and Aerosols from Nature version 2.1 (MEGANv2.1), as established in CLM (Guenther et al., 2012). Aerosol mass and number emissions are passed directly to MAM constituents. Global anthropogenic emissions can be specified from any of the standard GEOS-Chem inventories, but default to the Community Emissions Data System (CEDS) inventory (Hoesly et al., 2018). Sulfur emissions from the CEDS inventory are partitioned into size-resolved aerosol (mass and number) and $SO_2$ (Emmons et al., 2020). In CAM, volcanic out-gassing of $SO_2$ is provided from the GEIA inventory with 2.5% emitted as sulfate aerosol (Andres and Kasgnoc, 1998), while eruptive emissions are provided from the VolcanEESM database (Neely and Schmidt, 2016). The option is also available through HEMCO to use the "AeroCom" volcanic emissions, which are derived from OMI observations of $SO_2$ (Ge et al., 2016; Carn et al., 2015).

Although we use HEMCO with both model configurations, there remain differences between the representation of emissions in CAM-chem and in GEOS-Chem when run within CESM. This is because of differences in the species present in their respective mechanisms. For instance, emissions of iodocarbons ($CH_3I$, $CH_2I_2$, $CH_2ICl$, $CH_2IBr$) and inorganic iodine (HOI, $I_2$) are not available in CAM-chem since iodine is not explicitly modeled in CESM v2.1.1. VOC lumping is also performed differently (see the Supplemental Information for more detail).

Where the emitted species are present in both chemical mechanisms, the emissions calculated by HEMCO in CESM are identical whether running with either GEOS-Chem and CAM-chem. If the HEMCO implementations of lightning, dust, sea salt, and biogenic emissions are used, emissions will be identical between CESM and the standalone GEOS-Chem CTM.

### 2.3.4 Wet deposition and convection

For both GEOS-Chem and CAM-chem within CESM, convective scavenging and transport are handled separately. Wet deposition is performed using the Neu scheme (Neu and Prather, 2012), which simulates uptake and removal of soluble species by large-scale and convective precipitation. Unlike in the Liu et al. (2001) approach implemented in the GEOS-Chem





standalone code, removal of soluble gases within convective updrafts is not explicitly simulated in either CAM-chem or GEOS-
Chem when embedded in CESM. When using the CAM-chem mechanism within CESM, the Neu scheme is used to perform
washout of soluble gaseous species, while wet deposition of MAM aerosols is handled by MAM. When running CESM with
the GEOS-Chem chemistry mechanism, the Neu scheme also performs wet scavenging for aerosols which are not represented
by MAM4 (e.g. nitrate). For all such aerosols we assume a Henry's law coefficient equal to that for $HNO_3$.

### 2.3.5    Surface boundary conditions

In CESM, surface boundary mixing ratios of long-lived greenhouse gases (methane, $N_2O$, and chlorofluorocarbons) are set to
the fields specified for CMIP6 historical conditions and future scenarios (Meinshausen et al. 2017). For whichever scenario is
chosen, the boundary conditions overwrite those set by the GEOS-Chem chemistry module or by the HEMCO emissions
component.

### 2.4    Changes to the data flow in CESM when running with GEOS-Chem

In CESM, data such as the Henry's law coefficients required to calculate dry deposition velocities and wet scavenging rates
for each species are defined at compile time. For species that are common to GEOS-Chem and CAM-chem but where these
factors differ, the GEOS-chem values are used by default. The CAM-Chem values are listed alongside them in the source code
to allow users to switch if desired. Additionally, we modify CAM, CLM and CIME such that the land model can pass land
type information and leaf area indices to the atmosphere model to compute dry deposition velocities. This could be a potential
solution for dry deposition of aerosols in MAM, which currently uses fixed land types independent of the ones used in CLM
(Liu et al., 2012). However, this comes at the cost of passing land information through the coupler at every time step.

### 2.5    Installation and compilation process

The interface between CESM and GEOS-Chem, as well as the GEOS-Chem source code, is automatically downloaded when
CAM checks out its external repositories. The versions of GEOS-Chem and of the coupling interface can be changed by
modifying the `Externals_CAM.cfg` and by running the `checkout_externals` command.

When creating a new case, the user chooses the atmospheric chemistry mechanism (GEOS-Chem or CAM-chem). The
chemistry option is defined by the name of the CESM configuration (component set, or "compset"), making the process of
creating a run directory almost identical when choosing either GEOS-Chem or CAM-chem. Whereas chemistry options in
CAM-chem are set explicitly using namelist files, certain options in GEOS-Chem are set using ASCII text input files which
are read during the initialization sequence. The installation and build infrastructure of CIME will therefore copy any GEOS-
Chem specific text input files to the case directory when setting up a simulation which includes GEOS-Chem. This currently
includes emissions specifications read by HEMCO, although this is expected to change as HEMCO becomes the standard
emissions option for both CAM-chem and GEOS-Chem.




Although CESM supports both shared-memory parallelization (OpenMP) and distributed memory parallelization (MPI),
GEOS-Chem implemented in CESM does not currently support OpenMP. When running CESM with the GEOS-Chem
chemistry model, the number of OpenMP threads per MPI task is therefore set to one.

Although a complete copy of the GEOS-Chem source code is downloaded (to ensure , not all files present in the GEOS-Chem
source code directory are compiled. For instance, the files pertaining to the GEOS-Chem advection scheme are not needed as
advection is performed by CAM, and therefore the GEOS-Chem advection routines are not compiled. To do this we implement
a new feature in CIME to use `.exclude` files which list files not needed during compilation. CIME reads each `.exclude`
file at compile time and searches subdirectories recursively from the location of the exclude file, preventing any named file
from being included in compilation. For example, an `.exclude` file is provided in the chemistry coupling interface folder
for GEOS-Chem that lists the files to exclude in the GEOS-Chem source code directories.
**3    Model evaluation**
We simulate a two-year period with GEOS-Chem embedded in CESM (hereafter C-GC), to support two evaluations. First, we
perform a comparison of its output to that generated by two other model configurations (Section 3.2). By comparing the results
to those produced for the same period by CESM with CAM-chem (hereafter C-CC), we can perform the first comparison of
GEOS-Chem and CAM-chem when run as chemistry modules within the same ESM. Any differences between these two
simulations can only be the result of differences between the two chemical modules and their implementations in CESM. This
includes not only differences in the gas-phase chemical mechanism, but also in the implementation of photolysis calculations,
heterogeneous chemistry, aerosol microphysics, and the chemical kinetics integrator itself. We also compare output to that
produced by the standalone GEOS-Chem (hereafter S-GC). This enables us to evaluate the effect of using CESM's grid
discretization, advection, aerosols, and representation of meteorology compared to that used in the GEOS-Chem CTM.

Second, we evaluate the performance of C-GC by comparing output to observational data (Section 3.3). We also include
comparisons of data from the C-CC and S-GC configurations, to provide insight into the relative performance of the model
and the root cause of disagreements with observations.
**3.1    Simulation setup**
All simulations cover the period January 1$^{st}$ 2015 to December 31$^{st}$ 2016, with the first year discarded as spinup. For C-CC,
the standard restart file provided with CESM is used to provide initial conditions. For S-GC, we use a restart file provided with
version 13.1.2 of the GEOS-Chem chemistry module. For C-GC, we use initial conditions which are taken from the S-GC
restart file where possible, but fill missing species (e.g. MAM4 aerosol tracers) using data from the C-CC restart file. Both



simulations performed with CESM v2.1.1 (C-GC and C-CC) use a horizontal resolution of 1.9°×2.5° on 56 hybrid pressure
levels, extending from the surface to 1.65 hPa. Aerosols are represented in CESM using the 4-mode version of the modal
aerosol model, MAM4 (Liu et al., 2012). In C-GC, we use the complex SOA chemistry scheme (Pye and Seinfeld, 2010; Pye
et al., 2010; Marais et al., 2016). In C-CC, we use the MOZART-TS1 chemistry scheme (Emmons et al., 2020).

Standalone GEOS-Chem (S-GC) simulations are performed using the GEOS-Chem High Performance (GCHP) configuration,
using a C48 cubed-sphere grid (approximately equivalent to a 2°×2.5° horizontal grid) on 72 hybrid pressure levels extending
up to 1 Pa. In GCHP, chemistry is performed up to 1 hPa (~50 km) with simplified parameterizations used above that point.
Aerosols are represented using GEOS-Chem's "native" scheme, without translation to or from MAM4. As in C-GC, we use
the complex SOA scheme.

All three model configurations are driven using meteorological data from the Modern Era Retrospective analysis for Research
and Applications, version 2 (MERRA-2). In S-GC all meteorological fields are explicitly specified by MERRA-2, using the
same 72-layer vertical grid. The only exception is the specific humidity in the stratosphere, which is computed online. In C-
CC and C-GC, we use the "specified dynamics" (SD) configuration of CAM6 in which 3-D temperature, 3-D wind velocities,
surface pressure, surface temperature, surface sensible heat flux, surface latent heat flux, surface water flux, and surface
stresses are provided by MERRA-2 on a truncated 56-layer vertical grid. These variables are nudged with a relaxation time of
50 hours, resulting in a relatively "loose" nudging strength. All other fields (e.g. cloud fraction) are computed using the CAM
physics routines. This includes convection. Whereas S-GC computes convective transport from archived convective mass
fluxes and calculates scavenging within the updraft (Wu et al., 2007), convective transport in both C-CC and C-GC is
calculated in CAM6 using the CLUBB-SGS scheme for shallow convection and the Zhang-McFarlane scheme for deep
convection. Scavenging within the convective updraft is not simulated explicitly.

Water vapor in C-GC is initialized from the specific humidity "Q" restart variable, which is identical to the one used for C-
CC; after this point humidity is calculated based on the moist processes represented explicitly in CAM's physics package. The
GEOS-Chem CTM does not calculate water vapor in the troposphere, instead prescribing specific humidity directly from
MERRA-2 output. Mixing ratios of water vapor in C-CC and C-GC are therefore identical to that in S-GC at initialization
time, but from that point onwards may diverge.

Emissions are harmonized between the three models, with all three configurations using HEMCO to calculate emissions fluxes.
Surface anthropogenic emissions are provided from CEDS and are identical between all three models, apart from small
differences in effective emissions from ships due to parameterized plume processing (Vinken et al., 2011). Simulated
anthropogenic surface emissions of nitrogen oxides are 145-148 Tg($NO_2$) in each of the three models. Aviation emissions are



calculated in all three models based on the AEIC 2005 emission inventory, contributing a further 2.7 Tg $NO_2$ in addition to
other species (Simone et al., 2013).

Lightning emissions are calculated in C-CC and C-GC using the online parameterization described in Section 2.3.3, while
lightning emissions in S-GC are calculated using archived flash densities and cloud top heights (Murray et al., 2012). Total
emissions are 13 Tg NO in all three models, with less than 2% difference in total. Biogenic emissions are calculated in C-CC
and C-GC using the embedded MEGAN emissions module in CESM, which differs slightly from the implementation in S-GC
and will produce different emissions due to different vegetation distributions. Total biogenic emissions in S-GC and C-GC are
shown in Table 2. In all three simulations we use the "AeroCom" volcano emissions implemented in HEMCO.

*Table 2. Annual global biogenic emission totals in standalone GEOS-Chem (S-GC) compared to in GEOS-Chem implemented in CESM (C-*
*GC).*

| Species | Name in GEOS-Chem | S-GC (Tg/year) | C-GC (Tg/year) |
|---|---|---|---|
| Acetone | ACET | 48.2 | 42.7 |
| Acetic acid | ACTA | - | 3.86 |
| Acetaldehyde | ALD2 | 17.9 | 20.8 |
| Lumped alkanes >= $C_4$ | ALK4 | - | 0.16 |
| Ethylene | C2H4 | - | 30.4 |
| Ethane | C2H6 | 0.21 | 0.34 |
| Propane | C3H8 | - | 0.03 |
| Formaldehyde | CH2O | - | 5.14 |
| Carbon monoxide | CO | - | 88.8 |
| Ethanol | EOH | 17.9 | 20.8 |
| Limonene | LIMO | 9.11 | 11.0 |
| α/β-pinene, sabinene, carene | MTPA | 81.5 | 98.6 |
| Other monoterpenes | MTPO | 38.6 | 40.8 |
| Isoprene | ISOP | 397.6 | 502 |
| Methanol | MOH | - | 119 |
| Toluene | TOLU | - | 1.57 |
| Lumped alkenes >= $C_3$ | PRPE | 24.2 | 22.3 |


Mobilization of mineral dust is calculated in all three models using the DEAD scheme (Zender, 2003). In C-CC and C-GC,
the online implementation in CESM is employed, resulting in total natural mineral dust emissions of 5984 Tg/year. A brief
discussion of dust emissions in CESM is provided in the SI. In S-GC, natural mineral dust emissions are calculated online





using the same scheme but with a different scaling and at a slightly different grid resolution, resulting in total emissions of
1390 Tg/year.

Emissions of sea salt are calculated online in CESM for C-GC and C-CC, while S-GC uses a pre-calculated (offline) inventory
of sea-salt emissions, as well as sea-salt bromine and chloride. Emissions of sea-salt bromine in C-GC are calculated based on
the offline inventory rather than the calculated emissions of sea salt, and therefore do not scale correctly with the estimated
sea-salt emissions from CESM  (see Table 3). This will be resolved as part of future work.

*Table 3. Annual global emissions of sea salt aerosols (fine and coarse) and bromine in sea salt for C-GC and S-GC. The names of the tracers*
*used to represent these species in GEOS-Chem are provided in brackets.*

| Species | C-GC (Tg/year) | S-GC (Tg/year) |
|---|---|---|
| Fine sea-salt (SALA) | 93.0 | 59.1 |
| Coarse sea-salt (SALC) | 2780 | 3576 |
| Bromine in fine sea-salt (BrSALA) | 0.166 | 0.126 |
| Bromine in coarse sea-salt (BrSALC) | 10.1 | 7.54 |


Finally, for long-lived species such as CFCs we use the shared socio-economic pathway 2-4.5 (SSP2-4.5) set of surface
boundary conditions in both C-GC and C-CC. In comparisons against S-GC we use historical emissions from the World
Meteorological Organization's 2018 assessment of ozone depletion (Fahey et al., 2018). However, this difference is unlikely
to significantly affect simulation output given the short duration of the simulations.
**3.2    Model intercomparison**
We first compare the global distribution of ozone and aerosols between C-GC, S-GC, and C-CC. Section 3.2.1 evaluates the
vertical and latitudinal distribution of ozone and two related species (water vapor and the hydroxyl radical), followed by the
global distribution of ozone at the surface in each model configuration (Section 3.2.2). A similar evaluation of differences in
zonal mean and surface aerosol concentrations follows (Section 3.2.3).

To understand the causes of these differences, we compare the global distribution of reactive nitrogen and halogen species in
each model configuration (Section 3.2.4). When comparing halogen distributions we consider only bromine and chlorine
distributions, as iodine is not simulated in this version of CAM-chem. Differences in the total atmospheric burden and vertical
distribution of these families provides information regarding differences in removal processes. Differences in their internal
partitioning (e.g. between $NO_x$ and $HNO_3$) provide information regarding the representation of atmospheric chemistry.



### 3.2.1 Ozone

Figure 2 shows the annual mean mixing ratio of stratospheric ozone simulated by each of the three model configurations. At 10 hPa in the tropics, where ozone mixing ratios reach their peak, the three configurations agree to within 10% suggesting a reasonable representation of stratospheric ozone. However, near the tropopause the three configurations diverge. C-GC simulates mixing ratios of ozone around the tropopause which are 20% lower than C-CC at all latitudes. This difference may be the result of greater mixing ratios of reactive bromine in the C-GC troposphere than in C-CC (see Section 3.2.4.2).

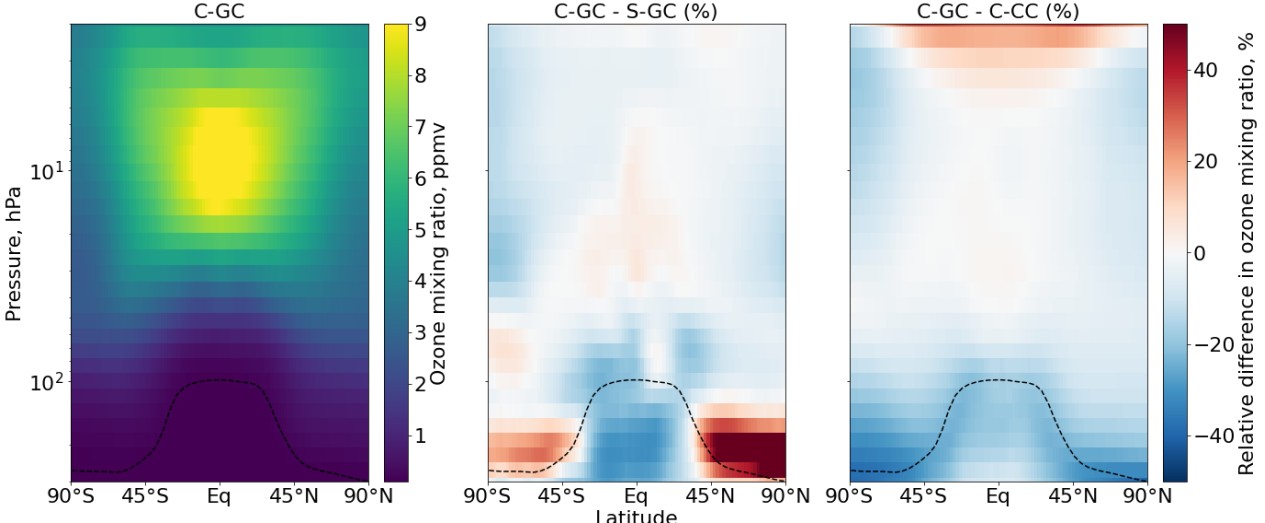

*Figure 2. Comparison of stratospheric ozone simulated with CESM running GEOS-Chem (C-GC) to standalone GEOS-Chem (S-GC) and CESM running CAM-chem (C-CC). Left column: absolute values estimated with C-GC. Center column: relative difference between C-GC and S-GC. Right column: relative difference between C-GC and C-CC. Red (blue) shading means that C-GC estimated a higher (lower) value than the other model.*

Comparison of C-GC to S-GC shows a different pattern, with mixing ratios 20% lower than S-GC near the tropical tropopause but more than 50% greater in the extratropical lower stratosphere. The absence of this pattern from the comparison against C-CC implies that the cause is likely to be related to factors which are common between C-GC and C-CC, such as the representation of meteorology.

To quantify and understand these differences in stratospheric ozone, we analyze concentrations of three different related compounds from the surface to the stratosphere: ozone itself, the hydroxyl (OH) radical, and water vapor. Since OH is produced from water vapor and (indirectly) ozone, these three compounds can collectively be used to understand some of the differences between C-GC, S-GC, and C-CC. Later analyses will focus on $NO_x$, bromine, and chlorine, each of which also strongly affect tropospheric and stratospheric concentrations of ozone.





The upper row of Figure 3 shows the distribution of ozone as represented by C-GC (left), and the difference when compared
to S-GC (center) or C-CC (right). Comparing first to C-CC, C-GC estimates mixing ratios of ozone which are 30% lower at
the surface (averaged over all latitudes) and throughout the extratropical troposphere. This is consistent with previous work
which showed that ozone simulated by GEOS-Chem to match the KORUS-AQ campaign had a normalized mean bias of -
26%, compared to -9% in CAM-chem. The lower concentrations of ozone are evident in both hemispheres, although
differences fall to zero in the tropical free troposphere. Ozone mixing ratios around the tropopause are also lower in C-GC
than in C-CC by 15-20%.

Comparing to the differences between C-GC and S-GC provides some insight into possible causes for these discrepancies.
Near-surface ozone in C-GC in the southern hemisphere is also 30-40% lower than in S-GC, suggesting a potential common
cause for the differences with C-CC. However, in the northern extratropical troposphere below 400 hPa, zonal mean differences
between C-GC and S-GC are consistently less than 10%. Ozone concentrations are also lower in the tropical mid-troposphere
in C-GC than in S-GC by 15-25%, whereas concentrations were well matched in this region between C-GC and C-CC. In the
lower stratosphere, ozone concentrations in C-GC are instead greater than in S-GC, with the difference in the northern
extratropical lower stratosphere exceeding 50%. The global ozone burden in C-GC is within 1.5% of that estimated by S-GC,
while C-CC has a total atmospheric ozone burden 15% greater than C-GC.

Differences in tropospheric $NO_y$ and halogens, in particular the higher loading of BrO in C-GC, may explain some of these
differences (see Section 3.2.4). However, another possible factor in these differences in ozone is differences in water vapor
distribution. The bottom row of Figure 3 shows the annual average simulated distribution of water vapor in C-GC, and the
difference relative to S-GC and C-CC. Water vapor concentrations are approximately equal between C-GC and C-CC, since
the representation of moist physics in the two models is identical. However, differences of up to 20% arise around the
tropopause, possibly due to the different representation of stratospheric water chemistry and settling of stratospheric aerosol
(including ice).

The differences between C-GC and S-GC are larger. Outside of the tropics and below the tropopause, water vapor
concentrations are up to 30% greater in C-GC than in S-GC. Differences are smaller in the tropics, but in the tropical upper
troposphere water vapor concentrations are instead 15% lower in C-GC than in S-GC. This may be part of the reason that
water vapor concentrations in the extratropical lower stratosphere are more than 50% lower in C-GC than in S-GC, since the
tropical upper troposphere is the source of water vapor to the stratosphere. This is the same region in which C-GC calculates
ozone mixing ratios which are more than 50% greater than in S-GC, potentially due to the lower concentration of water vapor
(an indirect sink for ozone).



These differences arise due to the different representation of moist processes between CAM's physics package (used in both
C-GC and C-CC), and GEOS, which produces MERRA-2 and therefore is represented in S-GC. For example, although total
annual average precipitation agrees to within 10% between the models, the mean volumetric cloud fraction in C-GC and C-
CC is 15%, compared to 8% in S-GC. Meanwhile the area-averaged cloud water content and cloud ice content are 57% and
38% greater in S-GC than in C-GC (or C-CC).





*Figure 3. Comparison of atmospheric composition simulated with CESM running GEOS-Chem (C-GC) to standalone GEOS-Chem (S-GC) and CESM running CAM-chem (C-CC). Different rows show different constituents, while different columns show different model results. Top row: ozone. Middle row: OH radical. Bottom row: water vapor. Left column: absolute values estimated with C-GC. Center column:*





*relative difference between C-GC and S-GC. Right column: relative difference between C-GC and C-CC. Red (blue) shading means that C-*
*GC estimated a higher (lower) value than the other model.*
Differences in ozone and water vapor result in differences in concentrations of OH, as shown in the middle row of  Figure 3.
The global OH atmospheric burden is ~10% lower in C-GC than in S-GC, but this difference is not evenly distributed.
Differences in OH concentrations can be roughly considered to be the product of differences in ozone and differences in water
vapor, since both are needed to create OH (along with UV radiation). In the tropical troposphere, OH concentrations are more
than 50% lower in C-GC than in S-GC, likely due to a relative lack of both ozone and water vapor. However in the northern
mid- and upper latitudes below 900 hPa, OH concentrations are 10-20% greater in C-GC than in S-GC. This reflects the greater
water vapor concentrations and roughly equal ozone concentrations between the two models.
**3.2.2    Surface ozone**
Figure 4 compares the simulated, annually-averaged surface ozone mixing ratios as estimated by C-GC, S-GC, and C-CC. We
find that, when globally averaged, C-GC predicts a lower surface ozone mixing ratio than either C-CC or S-GC. The difference
compared to S-GC peaks over Southern Africa and over Northern India, reaching an absolute difference of 12 ppbv. Averaged
over each hemisphere, C-GC estimates a lower surface ozone mixing ratio than S-GC by 4.9 ppbv and 2.2 ppbv in the Southern
Hemisphere and Northern Hemisphere respectively. This varies between the land and oceans. In the Northern Hemisphere, we
observe no difference in surface ozone mixing ratio over the oceans, while a decrease of ~3 ppbv can be found over North
America, Europe and East Asia.

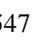
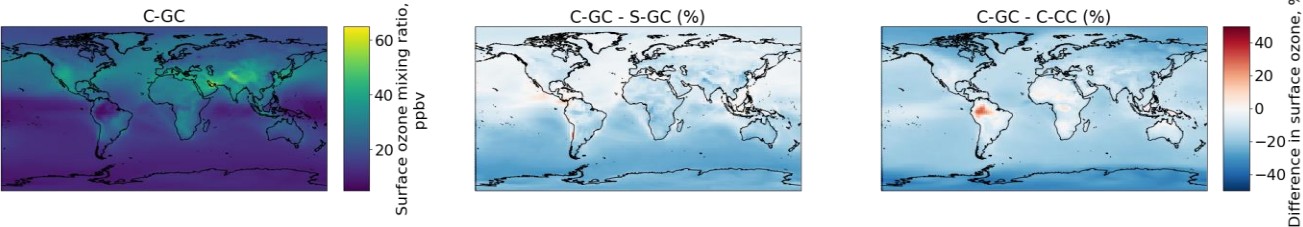


*Figure 4. Comparison of the annually averaged surface ozone mixing ratios simulated with CESM running GEOS-Chem (C-GC) to*
*standalone GEOS-Chem (S-GC) and CESM running CAM-chem (C-CC). Red (blue) shading means that C-GC estimated a higher (lower)*
*value than the other model.*
The difference between C-GC and C-CC does not show the same hemispheric asymmetry, and a larger difference over oceans
than over land. We find that C-GC estimates 5.4 and 7.9 ppbv less ozone than C-CC in the Southern and Northern Hemispheres
respectively. The pattern indicated in Figure 4 suggests that bromine from sea salt may be the principal cause of the differences
in surface ozone between C-GC and C-CC, whereas differences between C-GC and S-GC are likely to be related to
anthropogenic emissions given the hemispheric asymmetry. The 20-30% increase in ozone over the Amazon in C-GC related
to C-CC may instead be related to differences in biogenic emissions.




In addition to annual averages, we also consider seasonal variations of surface ozone. Figure 5 presents parity plots of monthly-
averaged surface ozone mixing ratios for January and July comparing C-GC to S-GC and C-CC, after outputs from all three
model configurations were remapped to a common 2°×2.5° grid. In January, we find a correlation coefficient of 0.87 and slope
of 0.93 between C-GC and S-GC. In July this agreement is worsened, with a correlation coefficient of 0.76 and a slope of 0.91.
This indicates that the sources of differences in surface ozone mixing ratios between C-GC and S-GC are magnified during
boreal summer. There is also a distinctive "hot spot" in the July parity plot, with a large cluster of grid cells showing mixing
ratios in the range 20-25 ppbv in S-GC but 15-20 ppbv in C-GC. Further research is needed to establish the origin of this
cluster, which does not occur during boreal winter.


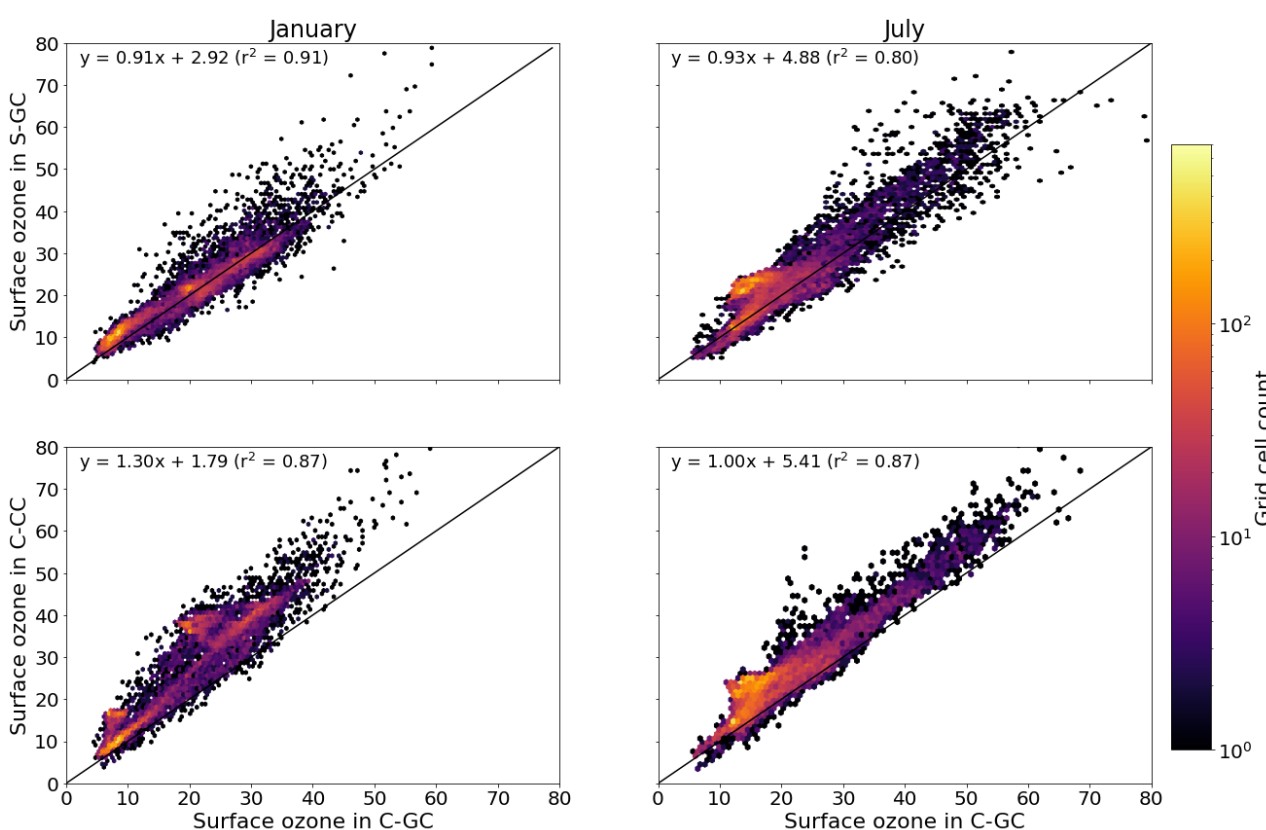


*Figure 5. Parity plots of surface ozone mixing ratios, expressed in ppbv, for January (left) and July (right) comparing C-GC on the X axis to S-GC (top) and C-CC (bottom) on the Y axis. Fitting parameters are shown in the top left corner for both months. All panels share the same color scale.*

Comparison between C-GC and C-CC shows a different pattern. The line of best fit between C-CC and C-GC indicates 30%
greater ozone in C-CC in January than in C-GC (y ~ 1.3x), but no such normalized mean bias is present in July (y ~ 1.0x). As
with the comparison of C-GC to S-GC, the absolute bias is greater in July than in January, but the correlation between C-CC



and C-GC does not worsen between the two months ($r^2 = 0.87$). This may indicate the strength of the effect of meteorology
and non-chemistry processes in the seasonality of simulated surface ozone.

### 3.2.3    Aerosols

Figure 6 shows the zonal mean mass concentration of sulfate aerosol as simulated in each of the three model configurations.
In C-GC and C-CC, this is calculated as the sum across all aerosol size bins, whereas S-GC uses a bulk representation.

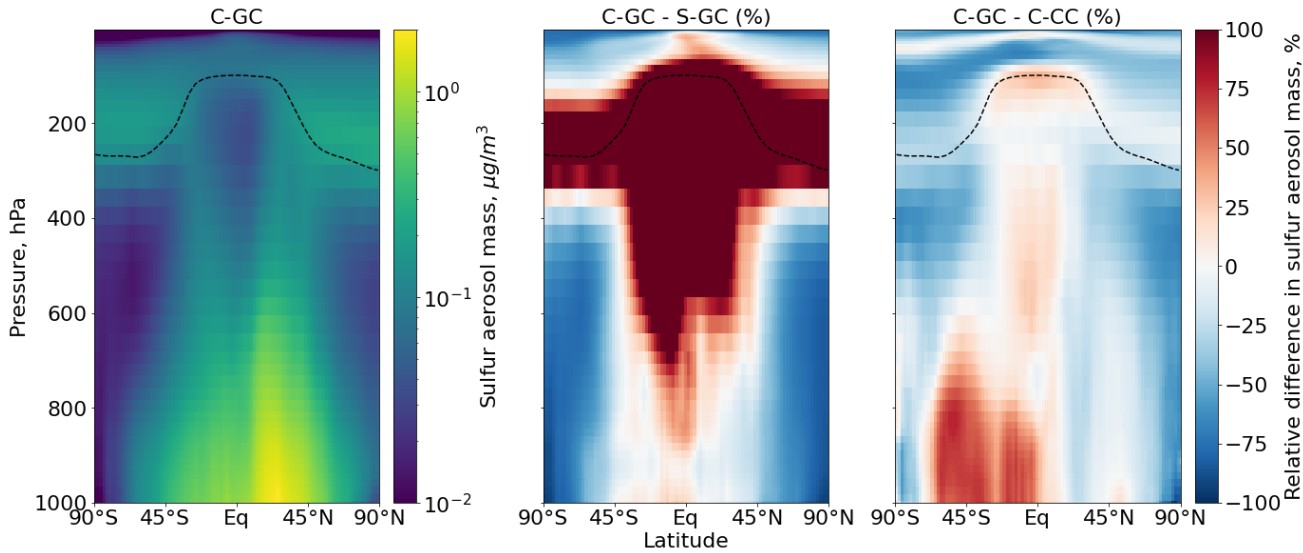


*Figure 6. Comparison of sulfate aerosol mass concentration as simulated with CESM running GEOS-Chem (C-GC) to standalone GEOS-Chem (S-GC) and CESM running CAM-chem (C-CC). Left: absolute values estimated with C-GC. Center: relative difference between C-GC and S-GC. Right: relative difference between C-GC and C-CC. Red (blue) shading means that C-GC estimated a higher (lower) value than the other model. Differences are restricted to ±100% for clarity.*

Between 45°S and 45°N, and below 800 hPa, C-GC more closely follows S-GC with regards to sulfate aerosol mass. Compared
to C-CC, sulfate aerosol mass is ~50% greater in Southern latitudes with differences being greatest over the emission. This is
despite emissions of DMS from the ocean being calculated the same way in all three model configurations. However, elsewhere
the concentration of sulfate in C-GC more closely follows that in C-CC, likely due to the common representation of sulfate
aerosol in MAM4 and differences in the representation of convective scavenging between CESM and standalone GEOS-Chem.
Concentrations of sulfate in the tropical upper troposphere and extratropical lower stratosphere in C-GC exceed those in S-GC
by over 100%, whereas comparison to C-CC show differences of ±25%.





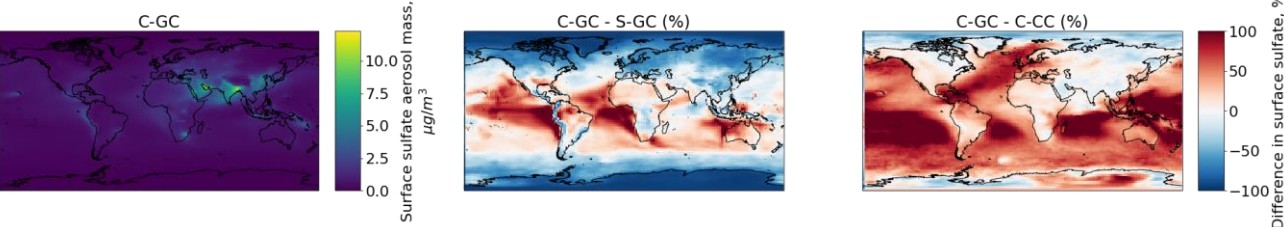


*Figure 7. Comparison of the annually averaged surface mass concentration of sulfate aerosol simulated with CESM running GEOS-Chem (C-GC) to standalone GEOS-Chem (S-GC) and CESM running CAM-chem (C-CC). Red (blue) shading means that C-GC estimated a higher (lower) value than the other model.*

This is further illustrated in Figure 7, which shows the surface concentration of sulfate aerosol in each model configuration. C-GC simulated greater concentrations in the intertropical convergence zone than in S-GC, but in these regions agrees more closely with C-CC. Elsewhere in the tropics the agreement between C-GC and S-GC is stronger, whereas surface concentrations of sulfate aerosol over (e.g.) the Southern Pacific exceed those in C-CC by over 100%. At high latitudes and over land the agreement between C-GC and C-CC is again stronger than in S-GC, although this varies by location.

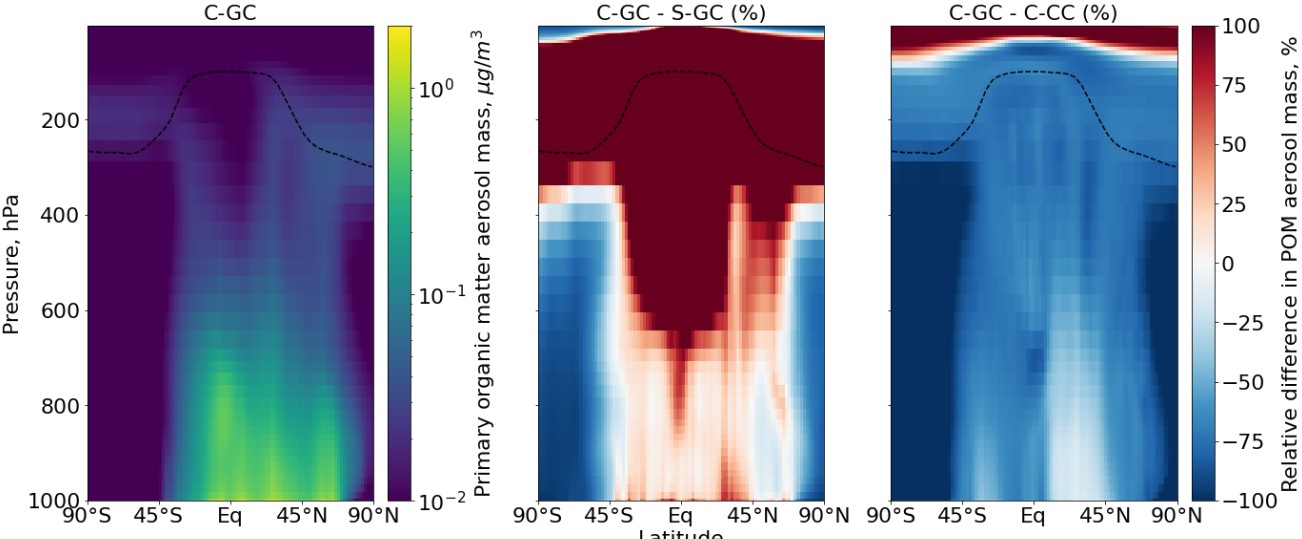

603

*Figure 8. Comparison of primary organic matter aerosol mass concentration as simulated with CESM running GEOS-Chem (C-GC) to standalone GEOS-Chem (S-GC) and CESM running CAM-chem (C-CC). Left: absolute values estimated with C-GC. Center: relative difference between C-GC and S-GC. Right: relative difference between C-GC and C-CC. Red (blue) shading means that C-GC estimated a higher (lower) value than the other model.*

We also show the zonal mean concentrations of primary organic matter (POM) aerosol in each configuration (Figure 8). POM in C-GC and C-CC is calculated as the sum of the POM aerosol size bins, whereas in S-GC it is the sum of the hydrophobic and hydrophilic organic carbon species. As with sulfate aerosol, C-GC and S-GC agree to within 25-50% in the tropics below 800 hPa, but C-GC simulates concentrations of POM which are over 100% greater than S-GC in the tropical upper troposphere and extratropical lower stratosphere. This is again likely due to differences in the representation of convective scavenging. C-





GC also simulates concentrations of POM which are lower than C-CC throughout the entire troposphere. This is likely due to
differences in the implementation of POM emissions between C-CC and C-GC, where emissions of POM in C-CC are 29%
lower and occur as accumulation-mode rather than primary organic mode aerosol.
**3.2.4    Reactive nitrogen (NO$_y$), bromine (Br$_y$), and chlorine (Cl$_y$)**
To better understand the source of differences in ozone and aerosols described above, we now investigate differences in
reactive nitrogen (NO$_y$) and halogen families (Br$_y$ and Cl$_y$).

**3.2.4.1    Reactive nitrogen (NO$_y$)**
We compare the total concentration and partitioning of reactive nitrogen species in each model configuration, including
nitrogen oxides (NO$_x$) and its reservoir species (collectively NO$_y$). We first compare results in the stratosphere, followed by
an evaluation of concentrations and partitioning below 100 hPa. Concentrations of nitrate aerosol concentrations are estimated
in CAM-chem using a simplified approximation (Lamarque et al., 2012), and particulate nitrate is typically not considered to
be simulated by CAM-chem (e.g. Park et al (2021)). We therefore do not include it in this analysis.

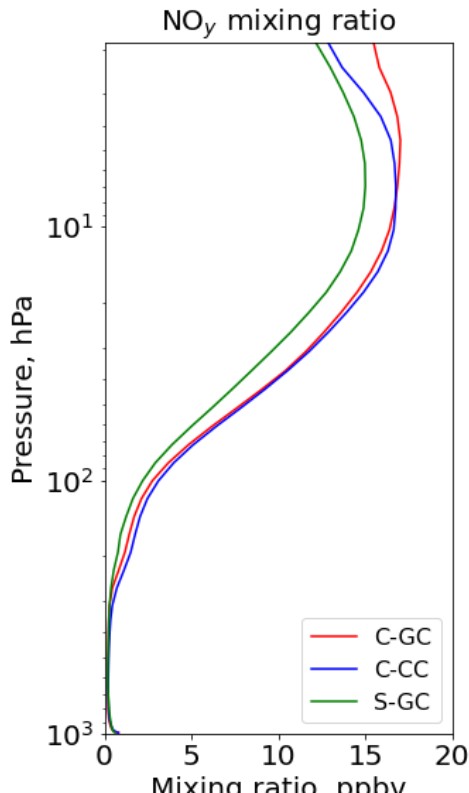
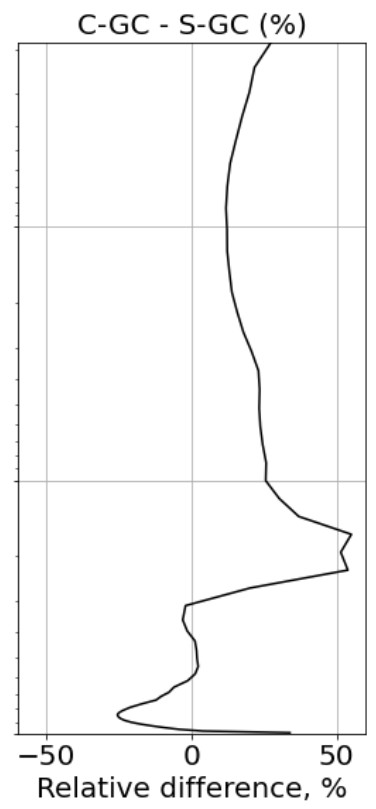
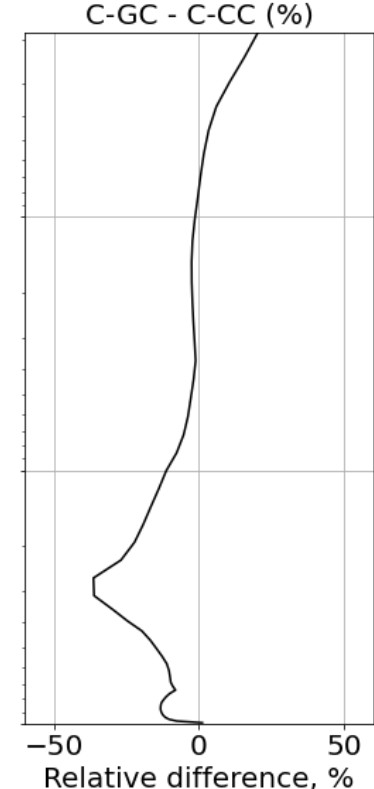




*Figure 9. Global annual mean mixing ratio of total reactive nitrogen (NO$_y$) as a function of altitude. Left: Vertical profile of NO$_y$ mixing ratio for C-GC (red), C-CC (blue), and S-GC (green). Middle: Relative difference in NO$_y$ mixing ratio between C-GC and S-GC. Right: Relative difference in NO$_y$ mixing ratio between C-GC and C-CC.*

Figure 9 shows global mean NO$_y$ at each altitude for C-GC, C-CC, and S-GC. Comparing C-GC to C-CC first, differences in total NO$_y$ are less than ±30% at all altitudes. Between 100 and 10 hPa, C-GC differs from C-CC by less than 10%. Above 10 hPa, the vertical profile more closely matches S-GC than C-CC. The difference between C-GC and C-CC increase from -2% at 10 hPa to +20% at the top of the model, compared to an increase from 10% to 25% when comparing C-GC to S-GC. At lower altitudes C-GC more closely follows C-CC than S-GC, with differences between C-GC and S-GC exceeding 50% between 200 and 300 hPa. The global NO$_y$ burden in C-GC (2.74 TgN) is closer to that in S-GC (2.84 TgN) than C-CC (3.01 Tg), likely due to the stronger influence of the troposphere on this quantity.

Figure 10 shows the speciation of NO$_y$ as a function of altitude in each model from the surface to 0.1 hPa. At altitudes above 100 hPa, the dominant contributors to NO$_y$ in all three model configurations are NO, NO$_2$, HNO$_3$, and N$_2$O$_5$, although ClNO$_3$ contributes significantly between approximately 80 and 5 hPa. Between 10 and 200 hPa ratios of NO to NO$_2$ are approximately consistent between the models, lying in the range 0.35 to 0.50. This suggests broad consistency in actinic flux and ozone concentrations, given their role in controlling NO:NO$_2$ ratios in the stratosphere (Cohen and Murphy, 2003).

By contrast, partitioning between NO$_x$ and HNO$_3$ differs significantly between the three models. At 10 hPa, HNO$_3$ constitutes 20% of total NO$_y$ in C-GC but 23% in both C-CC and S-GC. This fraction increases with decreasing altitude at differing rates. At 200 hPa, HNO$_3$ constitutes 60 and 63% of NO$_y$ in C-GC and S-GC respectively, but 78% of NO$_y$ in C-CC. One possible cause of these discrepancies is heterogeneous chemistry. GEOS-Chem (in both S-GC and C-GC) uses a different representation of N$_2$O$_5$ hydrolysis than CAM-chem, but the CESM-driven simulation include a more detailed representation of the sulfate aerosol size distribution through MAM4.





651

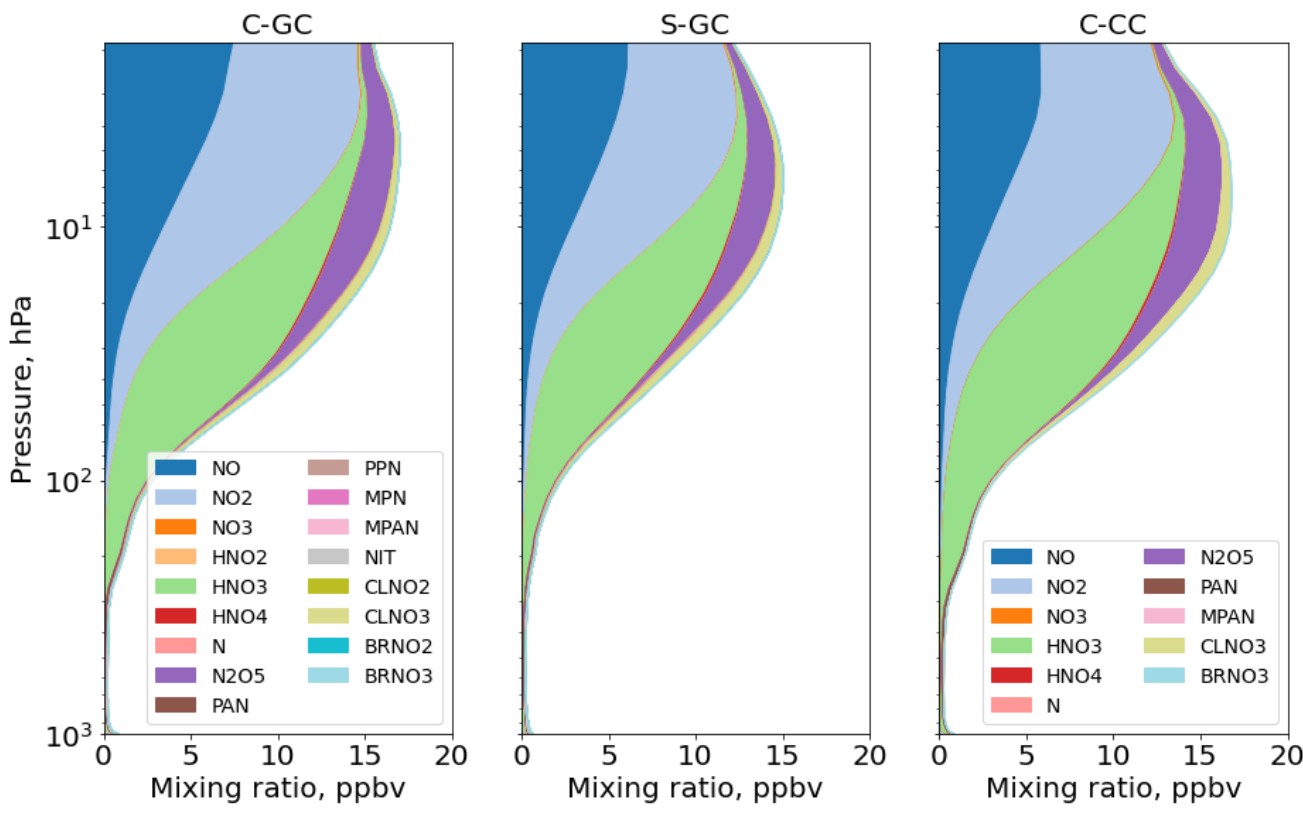

652

*Figure 10. Global annual mean speciation of NOy as a function of altitude. Results are shown from C-GC (left), S-GC (middle), and C-CC (right) from the surface up to the model top (~2 hPa). Values correspond to the number of N atoms present, such that (e.g.) the mixing ratio of $N_2O_5$ is multiplied by 2.*

Figure 11 provides a closer look at the speciation of $NO_y$ at altitudes below 100 hPa. $NO_y$ above 200 hPa is predominantly $NO_x$, $HNO_3$, and $N_2O_5$, at altitudes. However, between 200 and 900 hPa the dominant contributors are $HNO_3$ and peroxyacetyl nitrate (PAN), although the C-GC and S-GC simulations also show a significant contribution from nitrate aerosol (NIT) and $BrNO_3$. Below 900 hPa, NO and $NO_2$ once again become significant contributors to total $NO_y$. At these lower altitudes C-GC more closely follows C-CC than S-GC, with differences in total $NO_y$ between C-GC and S-GC exceeding 50% between 200 and 300 hPa. Since surface emissions of $NO_x$ are nearly identical between the three configurations and lightning $NO_x$ emissions are identical between C-GC and C-CC, differences below 100 hPa are most likely related to the representation of wet deposition and of nitrate aerosol.






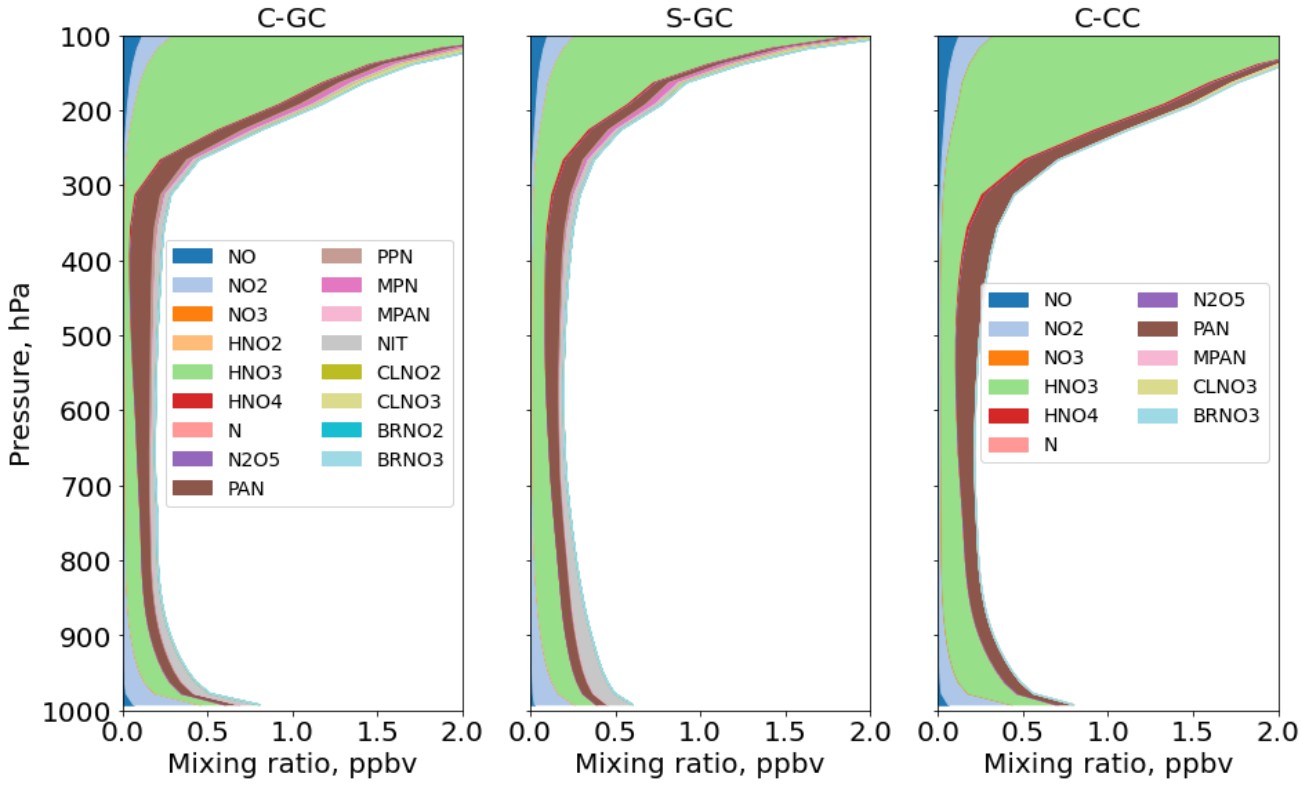


*Figure 11. As in Figure 10, but showing only the $10^3$-$10^2$ hPa pressure range.*

Although the total $NO_y$ mixing ratio in C-GC generally more closely follows C-CC than S-GC as discussed, the speciation in
C-GC more closely follows that in S-GC at lower altitudes. At 200 hPa, the combination of $NO_x$, $HNO_3$, and PAN make up
86% of total $NO_y$ in C-GC and 84% in S-GC, but 96% in C-CC. At 500 hPa the contributions are 78%, 85%, and 97%
respectively. However, concentrations of PAN in C-GC more closely follow C-CC than S-GC. At 500 hPa, total PAN in C-
GC is 3% lower than the value in C-CC, but exceeds the value in S-GC by 38%. This may be due to the greater emissions of
biogenic VOCs in CESM than in the standalone GEOS-Chem (see Table 2), resulting in more $NO_x$ being bound into PAN for
long-range transport. We also find that $HNO_3$ concentrations in the mid-troposphere are lower in C-GC than in either C-CC or
S-GC. At 500 hPa, $HNO_3$ mixing ratios in C-GC are 43% lower than in S-GC and 52% lower than in C-CC. This does not
account for the conversion of $HNO_3$ in C-GC and S-GC to nitrate aerosol (NIT), which is not represented in C-CC.

Differences in mid-tropospheric $HNO_3$ between the models are most likely due to differences in the representation of wet
scavenging. In C-CC and C-GC, scavenging of gaseous species is handled by the Neu scheme, while scavenging of modal
aerosols is performed by MAM (Neu and Prather, 2012). Any aerosol species not handled by MAM, such as nitrate in C-GC,
are also scavenged using the Neu scheme. In C-GC and C-CC, the Neu scheme calculations are performed at the same time as




the chemistry and after convective transport, while scavenging of MAM aerosols is performed before. Thus, all species that
undergo wet deposition in the Neu scheme are not removed during convective transport. This leads to soluble species and
aerosols being carried to higher altitudes without being convectively scavenged.

*Figure 12. Annual zonal mean of nitric acid wet removal tendencies for C-GC (left), S-GC (middle), and C-CC (right).*


Figure 12 shows the calculated wet removal rate of HNO$_3$ in all three models. Positive values correspond to rain re-evaporation
at low altitudes re-releasing dissolved HNO$_3$. The Neu scavenging scheme in C-GC and C-CC results in an HNO$_3$ wet removal
rate which is four times higher in C-GC than in S-GC. This likely explains the greater depletion of HNO$_3$ in the mid-troposphere
calculated by C-GC compared to S-GC, as shown in Figure 11. Wet scavenging in C-CC is faster yet, with HNO$_3$ wet removal
rates approximately six times greater than in S-GC, and 50% greater than in C-GC. This is in part because the mixing ratio (or
fraction of total NO$_y$) of HNO$_3$ in the mid- and upper-troposphere as modeled in C-CC is greater than in either C-GC or S-GC,
but also because C-GC and S-GC simulate nitrate aerosol explicitly.





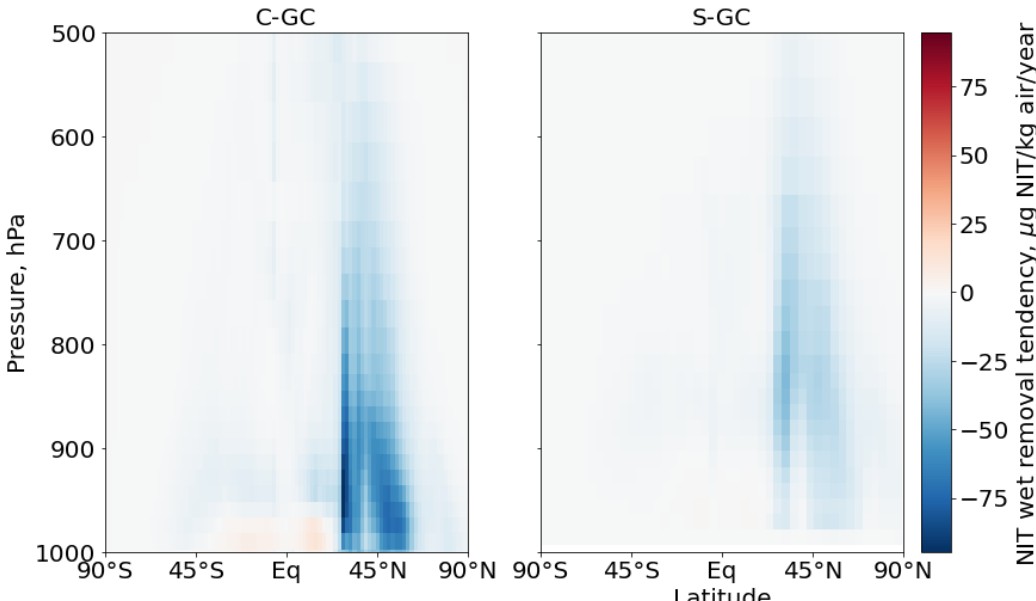

*Figure 13. Annual zonal mean of nitrate aerosol (NIT) wet removal tendencies for C-GC (left) and S-GC (right). Nitrate aerosols are not*
*modeled in CAM-chem.*
The application of the Neu scheme to remove nitrate aerosol also affects removal of total $NO_y$ in C-GC. Figure 13 shows the
annual mean wet removal rates of the nitrate aerosol tracer NIT in C-GC and S-GC. The Neu scheme removes aerosol more
rapidly than the scheme used in S-GC, and at lower altitudes.

Total $HNO_3$ removal in each model configuration are shown in Table 4. The total removal rate of $NO_3^-$ is lowest in S-GC and
highest in C-CC, consistent with the finding that total $NO_y$ burdens are lower in S-GC than C-GC or C-CC. However, the
removal rate of nitrate aerosol is lower in C-GC than in S-GC despite the greater wet removal rates shown in Figure 13 for C-
GC. A possible explanation is that washout rates of nitrate aerosol are sufficiently high in both C-GC and S-GC that all nitrate
aerosol is effectively removed, but that the faster washout of $HNO_3$ in C-GC results in less nitrate aerosol being available for
removal.

*Table 4. Total wet removal tendency of HNO₃ and nitrate aerosol in each model configuration. All values are given in units of Tg NO₃/yr.*

|  | C-GC | S-GC | C-CC |
|---|---|---|---|
| $HNO_3$ | 82.0 | 71.3 | 119.6 |
| Nitrate aerosol | 20.4 | 22.7 | - |
| Total $NO_3^-$ | 102.4 | 94.0 | 119.6 |




### 3.2.4.2 Reactive bromine (Br$_y$)

Figure 14 shows the annual average mixing ratio of total reactive bromine as a function of altitude in each of the three models. This does not include long-lived species such as halons or CH$_3$Br. A full listing is included in the legend of Figure 15.

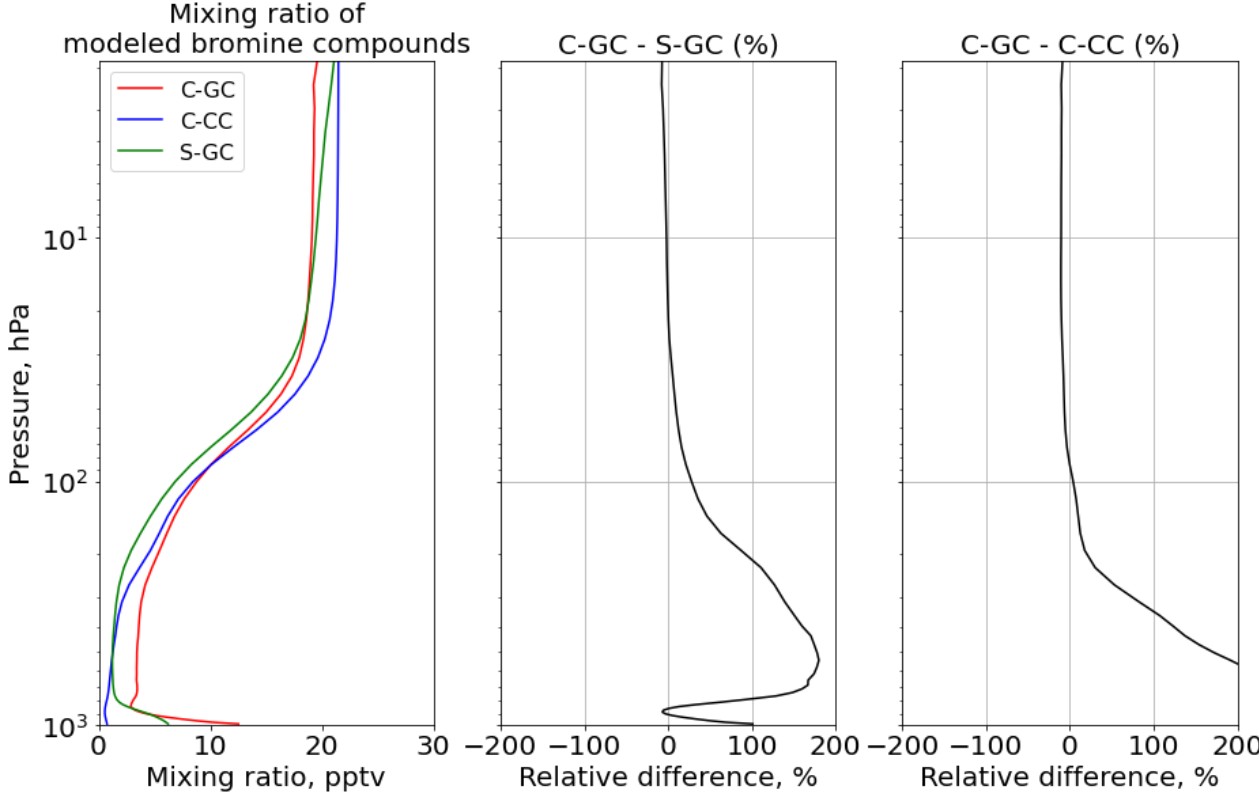

*Figure 14. Global annual mean mixing ratio of reactive bromine as a function of altitude. Left: Vertical profile of total gaseous inorganic and organic bromine mixing ratio for C-GC (red), C-CC (blue), and S-GC (green). Middle: Relative difference in bromine-containing species mixing ratio between C-GC and S-GC. Right: Relative difference in bromine-containing species mixing ratio between C-GC and C-CC. Although relative differences between C-GC and C-CC exceed 1000% near the surface, the limits on the rightmost panel are clipped to allow comparison to the center panel.*

Globally averaged, total Br$_y$ in C-GC is maximized at the surface, exceeding that from S-GC by 100%. This is partially explained by the greater emissions of sea salt bromine, although C-GC's annual emission of sea salt bromine is only 36% greater than that in S-GC (see Table 3). Since C-CC does not include short-lived bromine sources such as sea salt bromine, the difference between C-GC and C-CC exceeds 1000% at the surface.

In all three models the mixing ratio increases monotonically with altitude above 800 hPa due to the reaction of CH$_3$Br with OH. Br$_y$ falls sharply from 12 pptv at the surface in C-GC to 3 pptv at 900 hPa, but then increases again to 10 pptv at 100 hPa. This pattern is similar to that displayed by S-GC, although the decrease from the surface is less sharp and the absolute value lower in S-GC. Above 100 hPa the increase with altitude decreases, with values between 20 hPa and 2 hPa remaining roughly





constant in the range of 16-20 pptv. This is similar to the behavior shown by C-CC but differs from S-GC, in which $Br_y$
continues to rise with altitude – albeit more slowly. The net effect is that total $Br_y$ in C-GC exceeds both C-CC and S-GC
below 100 hPa, but is lower than the value in either model above 10 hPa.

In addition to differences in total $Br_y$, the partitioning of $Br_y$ also varies between the three models (Figure 15). The additional
near-surface bromine present in C-GC and S-GC is due to the presence of $Br_2$ and sea salt bromine (BrSALA and BrSALC,
representing bromine in fine and coarse-mode sea salt respectively). This provides a source of active bromine in the planetary
boundary layer which is not represented in C-CC, but in forms which are rapidly washed out. The greater concentrations of
$Br_y$ near the surface as calculated by C-GC compared to S-GC are likely due to the greater emissions of sea salt bromine, as
shown in Table 3.

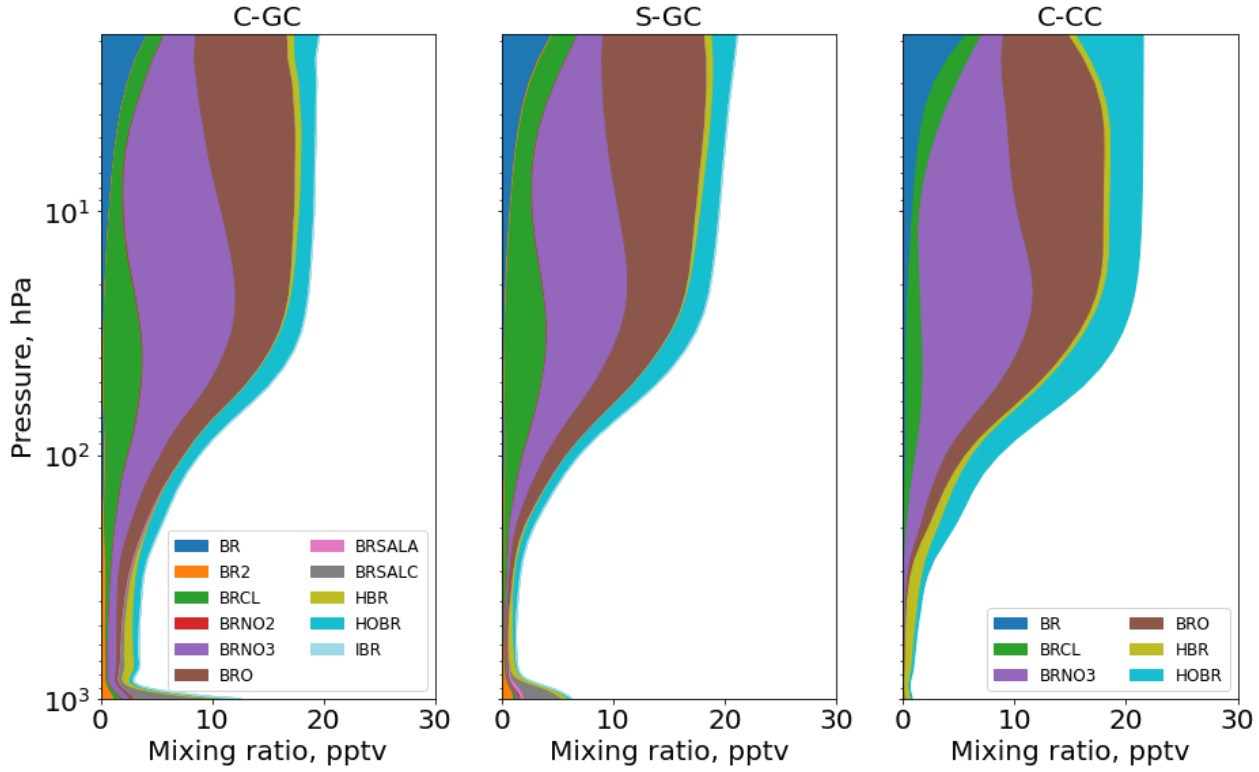


*Figure 15. Global annual mean speciation of total organic and inorganic bromine as a function of altitude. Results are shown from C-GC*
*(left), S-GC (middle), and C-CC (right), from the surface up to the model top (~2 hPa). Values correspond to the number of Br atoms present,*
*such that (e.g.) the mixing ratio of $Br_2$ is multiplied by 2.*
$Br_y$ in the model stratosphere is dominated by the same species in all three configurations: $BrO_x$ (Br + BrO), BrCl, $BrNO_3$,
HBr, and HOBr. The most significant difference is the greater proportion of HOBr in C-CC (~15%) than in S-GC or C-GC (8-

751     10%).

 

Between 30 hPa and the top of the boundary layer, the three models show divergent results. The only significant sources of atmospheric $Br_y$ in C-CC are $CH_3Br$, $CH_2Br_2$, and very long-lived bromine species such as halons which are insoluble. As a result, tropospheric $Br_y$ concentrations increase only slowly from the surface up to 300 hPa, at which point HOBr, BrO, and $BrNO_3$ begin to form in significant quantities. In C-GC and S-GC, these sources of bromine are supplemented by bromine from sea salt and surface $Br_2$ emissions. Mid-tropospheric $Br_y$ concentrations are therefore largely set by the quantity of sea salt bromine emitted, and by the fraction of that bromine which can be released to an insoluble form (e.g. $Br_2$) before the sea salt is washed out of the atmosphere.

The greater concentration of mid-tropospheric $Br_y$ in C-GC than in S-GC is likely due to differences in wet scavenging. Figure 16 shows the wet removal tendencies of bromine in fine sea salt (BrSALA) from large-scale and convective precipitation as calculated by C-GC and S-GC. We find that there is greater wet deposition of fine sea salt bromine in S-GC than in C-GC, despite removal rates below 900 hPa being greater in C-GC. Since total emissions of BrSALA are also 26% lower in S-GC than in C-GC (Table 3), the slower mid-tropospheric mid-tropospheric removal of bromine in C-GC explains the greater simulated concentration of $Br_y$ in the mid troposphere.

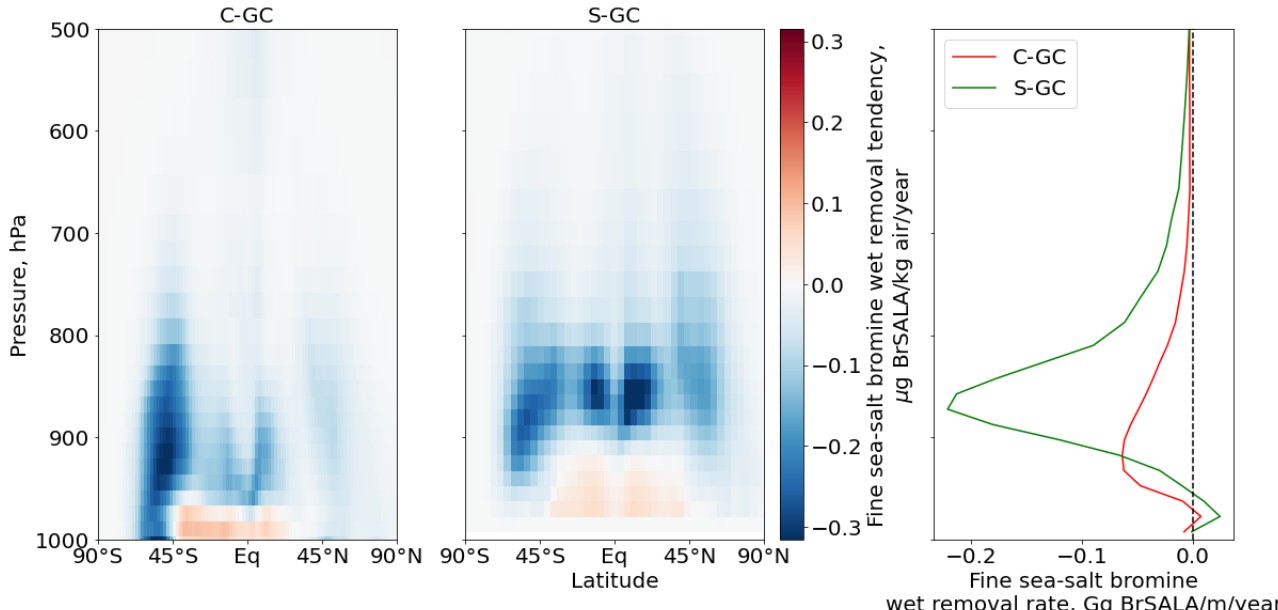

*Figure 16. Zonal mean wet removal tendency of bromine carried in fine sea salt. Left and middle: Removal rates calculated by C-GC (left) and S-GC (middle). Right: Annual mean of fine sea salt bromine aerosol wet removal rate for C-GC (red), S-GC (green). Bromine in sea-salt aerosol is not modeled in CAM-chem.*

C-GC also calculates wet deposition of non-MAM aerosols from both convective and large-scale precipitation independent of convective transport, whereas S-GC calculates convective scavenging as part of convective transport. This means that soluble species can be transported in convective updrafts in C-GC, unlike in S-GC.






### 3.2.4.3  Reactive chlorine ($Cl_y$)


We now focus on atmospheric chlorine by comparing its vertical profile and partitioning in all three models. Annually-
averaged vertical profiles of reactive chlorine ($Cl_y$) are displayed in Figure 17, excluding source species such as chlorocarbons.
A full list of the species used to define $Cl_y$ in each configuration is provided in Figure 18.

As with total $Br_y$, total $Cl_y$ has the same pattern of vertical distribution as S-GC up to around 10 hPa, but follows the pattern
of C-CC above this point. The dominant factor in differences below 100 hPa is the lack of short-lived chlorine species such as
sea salt in C-CC, which are the dominant source of chlorine to the lower troposphere. Above 10 hPa, the relative difference in
$Cl_y$ between C-GC and S-GC increases slowly from 2% at 10 hPa to 5% at 2 hPa, while the difference relative to C-CC remains
at 19-20%.

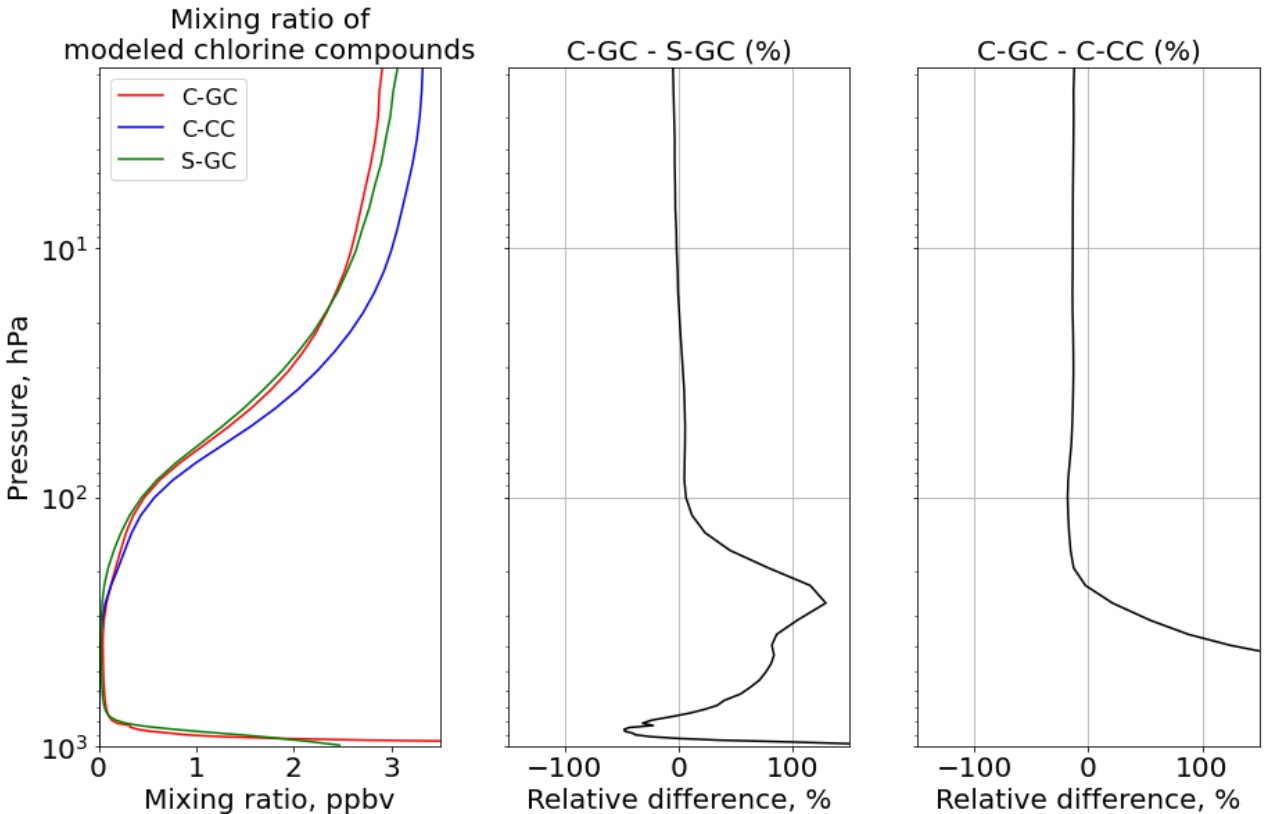


*Figure 17. Comparison of annual average vertical profiles of chlorine-containing compounds in the three models. Left: Vertical profile of total gaseous chlorine mixing ratio for C-GC (red), S-GC (green), and C-CC (blue). Middle: Relative difference in $Cl_y$ mixing ratio between C-GC and S-GC. Right: Relative difference in $Cl_y$ mixing ratio between C-GC and C-CC. Although relative differences between C-GC and C-CC exceed 1000% near the surface, the limits on the rightmost panel are clipped to allow comparison to the center panel.*



Figure 18 shows the speciation of $Cl_y$ as a function of altitude in each model. The greater near-surface chlorine simulated by
C-GC and S-GC relative to C-CC is mostly made up of HCl and chlorine in sea salt (SALACL and SALCCL). In the
stratosphere there is no clear difference between partitioning in C-GC and S-GC. However, production of upper tropospheric
and lower stratospheric HCl and $ClNO_3$ from chlorine source compounds appears to occur faster and at lower altitudes in C-
CC. At 50 hPa total $Cl_y$ in C-CC is 15% greater than in C-GC and S-GC, but the mean mixing ratio of HCl in C-CC is 45%
greater. Differences in $ClNO_3$ reach their peak at higher altitudes, around 20-30 hPa.

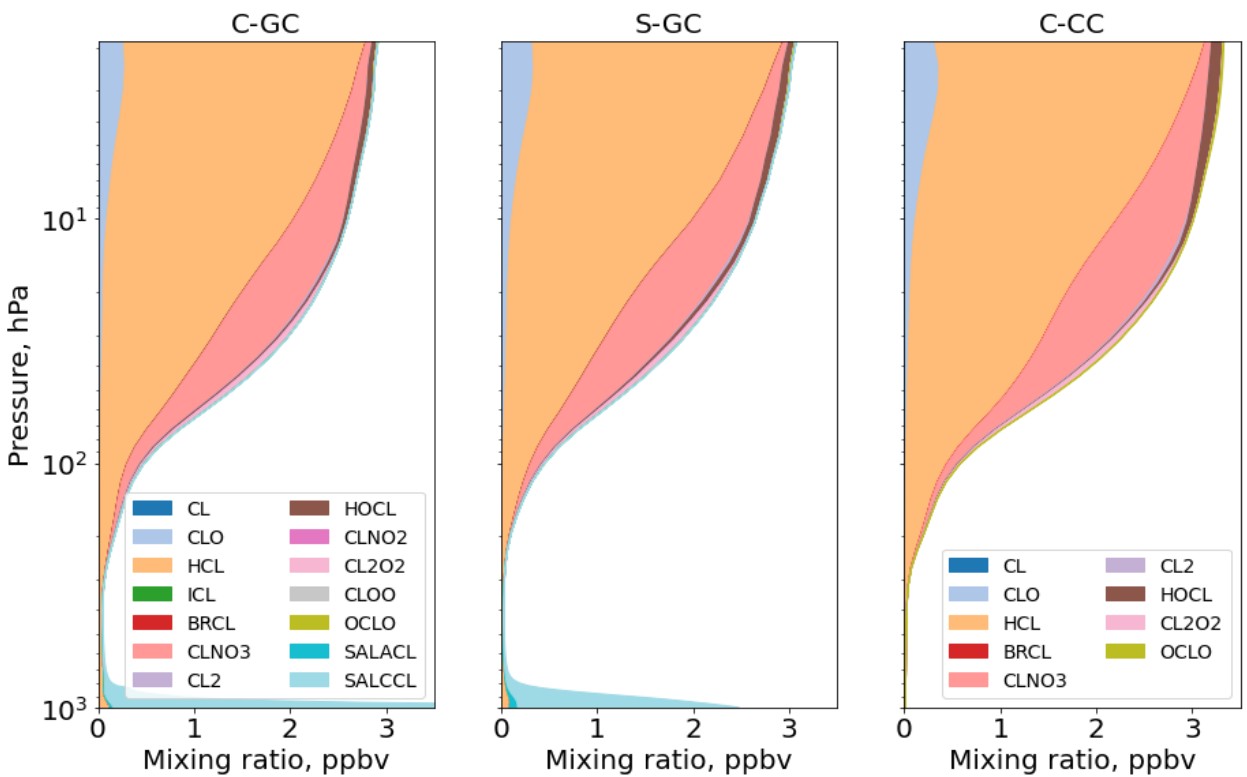


*Figure 18. Global annual mean vertical speciation of total organic and inorganic bromine in C-GC (left), S-GC (middle), and C-CC (right)*
*from the surface up to the model top (~2 hPa). Values correspond to the number of Cl atoms present, such that (e.g.) the mixing ratio of Cl₂*
*is multiplied by 2. SALACL and SALCCL correspond to chlorine in fine and coarse sea salt, respectively.*
The global mean tropospheric concentration of Cl atoms is 590 cm$^{-3}$, roughly consistent with a recent evaluation from Wang
et al. (2021) which found a value of 630 cm$^{-3}$. This is 24% greater than the value from S-GC (477 cm$^{-3}$) and 160% greater than
that from C-CC (224 cm$^{-3}$), likely due to the greater emissions of sea salt and indicating that chlorine will play a greater role
in tropospheric oxidation in C-GC.





**3.3   Comparison of model results to observations**
We now compare the results from C-GC to observational data, with results from S-GC and C-CC also provided as context.
Section 3.3.1 evaluates model performance at the surface, comparing to ground measurements of surface $NO_2$ and ozone.
Section 3.3.2 compares model results to a climatology of vertical profiles of ozone, based on ozone sonde data. Section 3.3.3
evaluates the level of agreement of simulated ozone and carbon monoxide columns to measurements from the OMI/MLS and
MOPITT satellite instruments. Finally, Section 3.3.4 evaluates the model against measurements of dry deposition fluxes and
rainwater composition measurements.
**3.3.1   Surface $NO_2$ and ozone**
Figure 19 compares surface mass concentrations of $NO_2$ as estimated by C-GC, S-GC, and C-CC for 2016 against ground
station measurements for North America, Europe, and South-East Asia (AirNow API, 2021; Environmental Numerical
Database, 2021; China Air Quality Historical Data, 2021; European Air Quality Portal, 2021). All ground station measurements
are the average value over 2016.

All three model configurations calculate lower mixing ratios than are reported by the ground observations. This is likely to be
in part due to the presence of interferants such as $HNO_3$, which cause in-situ monitors to overestimate the concentration of
$NO_2$ (Dunlea et al., 2007). However, S-GC is consistently biased lower than C-GC or C-CC. We also find that the surface $NO_2$
concentrations display variable agreement depending on the geographical location. In the U.S., Europe and South-East Asia,
the correlation coefficient equals 0.39, 0.21 and 0.42 respectively for C-GC, similar to the results of 0.38, 0.21, and 0.41 from
C-CC. S-GC provides correlation coefficients of 0.36, 0.21, and 0.41 respectively. This is expected given that the three model
configurations all use the same input wind fields and $NO_x$ emissions datasets. Nonetheless, both C-GC and C-CC estimate
higher concentrations of $NO_2$ in northern China, northern Europe, and the northeast US than S-GC. This suggests that the
representation of meteorology, photolysis, and $NO_y$ removal processes have a greater impact on simulated $NO_2$ than the
chemistry module alone.




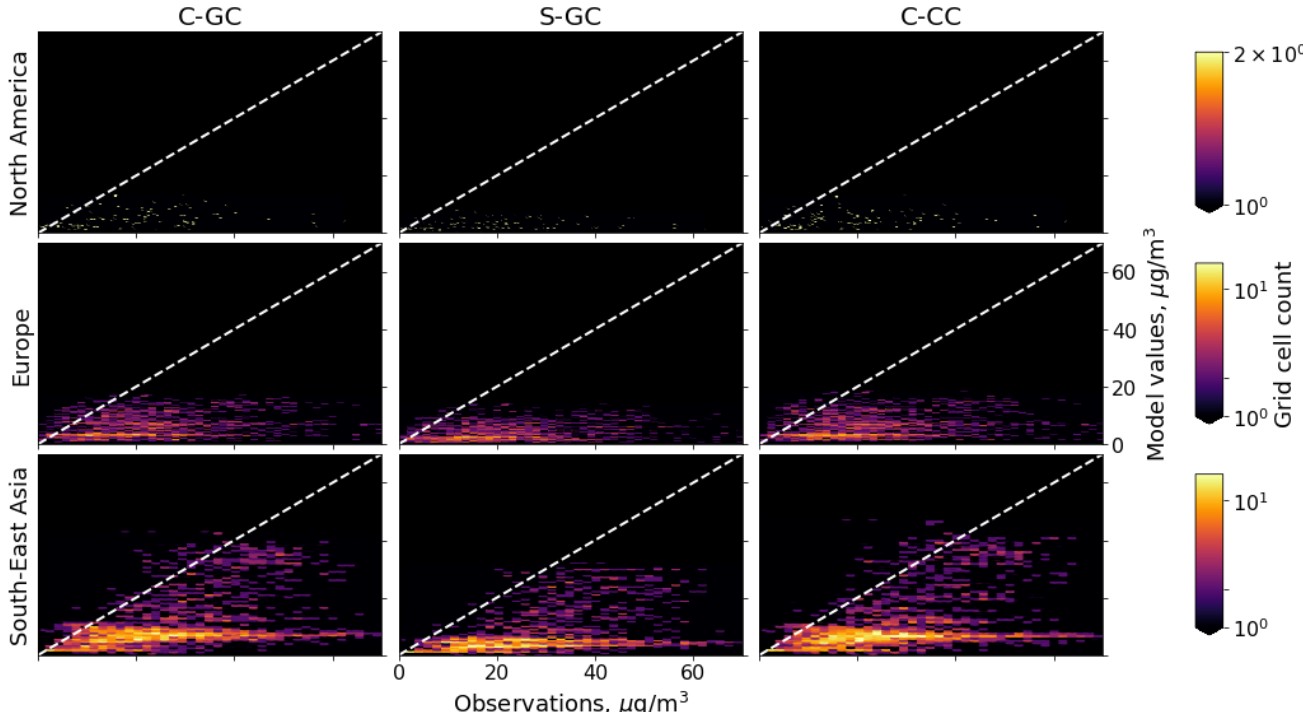


*Figure 19. Annual average surface NO₂ mass concentrations simulated by C-GC (left), S-GC (middle), and C-CC (right) for 2016*
*compared against monitor measurements in North America (top), Europe (middle), and South East Asia (bottom).*
Figure 20 shows the global distribution of $NO_2$, $NO_x$ ($NO + NO_2$), and the ratio of annual mean NO to annual mean $NO_2$, and
thus provides some insight into possible causes of these disagreements. All three configurations show enhanced $NO:NO_2$ ratios
in polluted regions such as eastern China and over icy regions such as Greenland and Antarctica. However, S-GC shows
reduced $NO:NO_2$ ratios over land compared to either C-CC or C-GC. For example, ratios over North America in S-GC range
from 0.1 to 0.2, compared to a range of 0.01 to 0.1 in C-GC and C-CC. Surface $NO:NO_2$ ratios are typically dictated by surface
ozone and the $NO_2$ photolysis rate (Seinfeld and Pandis, 2006). Given that surface ozone concentrations in S-GC are typically
between those calculated in C-GC and C-CC (see Figure 21) and that S-GC and C-GC share the same photolysis treatment,
this discrepancy may instead be caused by the differences in cloudiness calculated by CESM compared to the MERRA-2 fields
read in by S-GC.




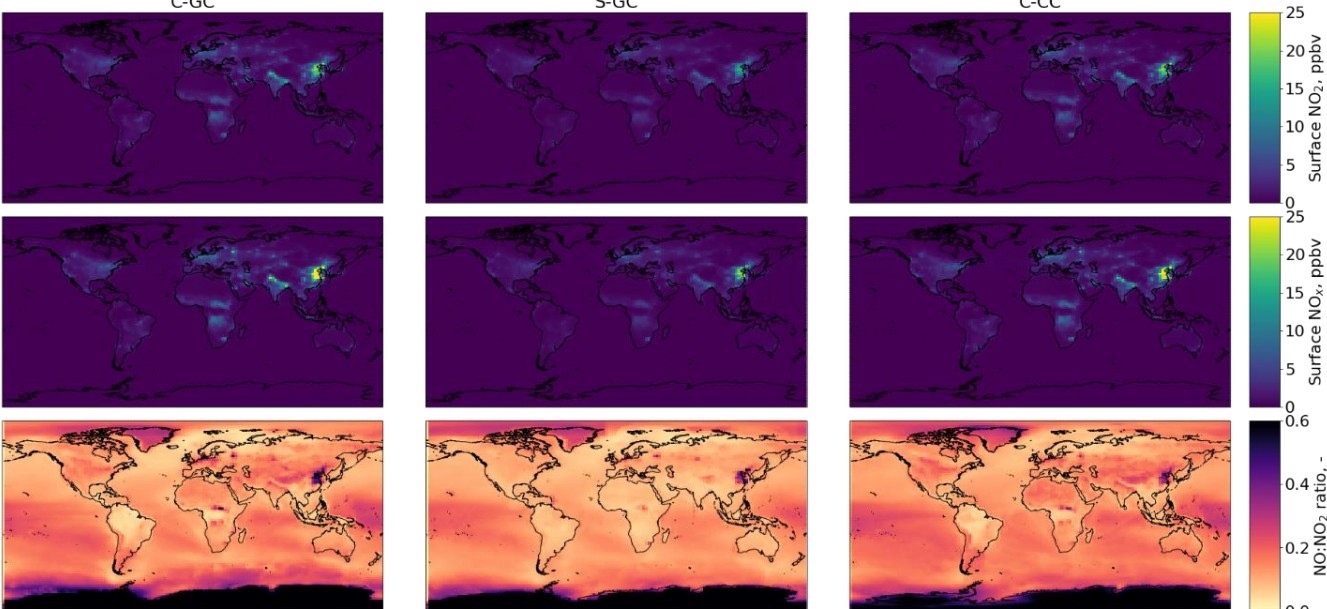


*Figure 20. Surface-level NO₂, NOₓ, and NO:NO₂ estimated by C-GC (left), S-GC (middle), and C-CC (right) for 2016. Top: annual average*
*NO₂ in ppbv. Middle: annual average NOₓ (NO + NO₂) in ppbv. Bottom: annual average NO:NO₂, calculated as annual mean NO divided*
*by annual mean NO₂ due to limited data availability.*
Differences in $NO:NO_2$ may also be related to differences in emissions and treatment of oxidants such as VOCs and bromine.
C-GC and C-CC show a reduction in $NO:NO_2$ over the Amazon and in the Congo river basin, but this pattern is not reproduced
in S-GC. Similarly, topographical features including the Andes and Himalayas are visible in the C-CC and C-GC $NO:NO_2$
ratios, but not in the S-GC data – whereas a large reduction in $NO:NO_2$ over the Arctic Ocean is more pronounced in S-GC
and C-GC than in C-CC. This latter feature may be related to differences in the response of the simulated atmosphere to
anthropogenic emissions, as ship tracks are more visible in the C-GC and S-GC $NO:NO_2$ ratios (see e.g. Cape Horn and the
Cape of Good Hope) than in C-CC.

Figure 21 compares surface ozone against monitor measurements. The correlation coefficient between the model simulation
results over the U.S. and Europe and the surface measurements for surface ozone is 0.37 and 0.44 respectively. C-CC and S-
GC predict correlation coefficients of 0.24 and 0.44, and 0.28 and 0.43 respectively, indicating a potential improvement in
correlation in the U.S. but not Europe. The geographical pattern is also consistent, with high surface ozone concentrations over
the Mediterranean sea and lower concentrations over Northern Europe.

However, the results from all models appear to be biased low. As discussed in Section 3.2.2, C-GC estimates surface ozone
mixing ratios lower than either S-GC and C-CC, and therefore exhibits the greatest mean bias. C-GC, C-CC, and S-GC show





mean biases of -15, -9, and -10 ppbv for over Europe; -10, -3, and -5 ppbv over North America; and -20, -11, and -12 ppbv
over South-East Asia.

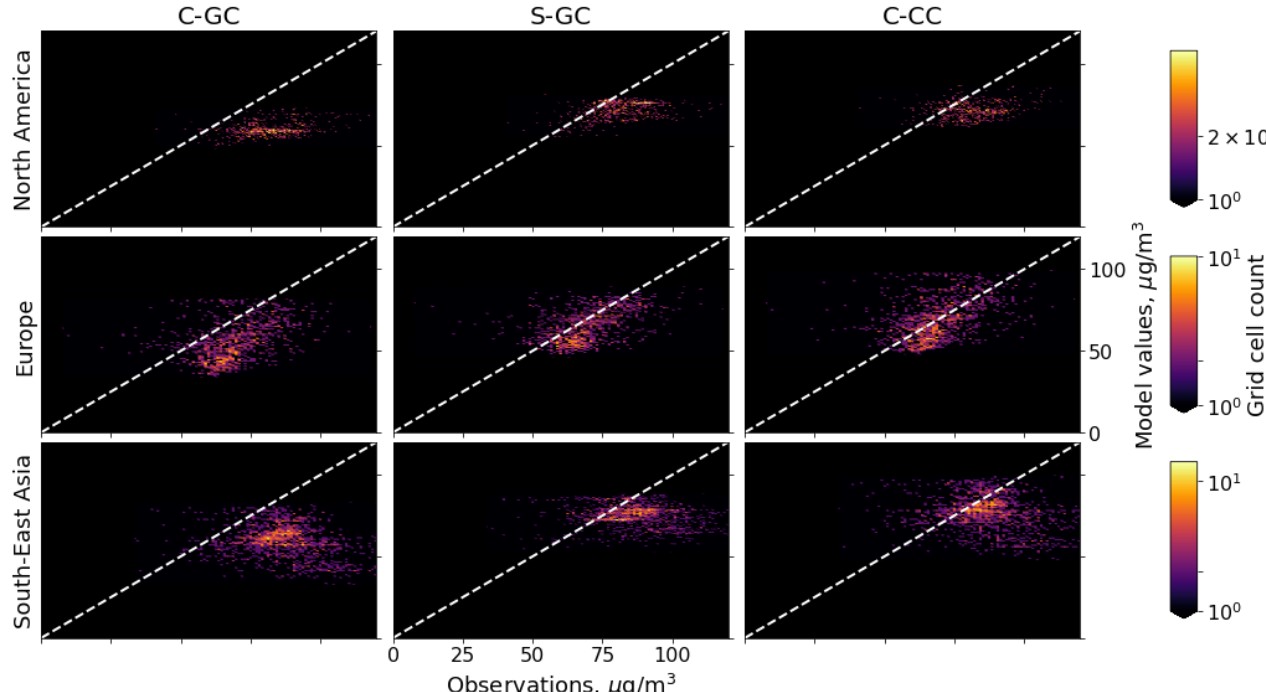


*Figure 21. Annual average surface ozone mass concentrations simulated by C-GC (left), S-GC (middle), and C-CC (right) for 2016*
*compared against monitor measurements in North America (top), Europe (middle), and South East Asia (bottom).*
The greater negative bias in simulated ozone shown by C-GC is likely related to both the different representation of
meteorology compared to S-GC and the greater bromine emissions compared to both S-GC and C-CC. However, further work
is needed to disentangle the root causes of discrepancies between the three models, and the common biases relative to
observations.
**3.3.2    Vertical profiles of ozone**
We now focus on the evaluation of the vertical profile of ozone mixing ratios by comparing C-GC, C-CC, and S-GC to a
climatology of ozone sonde observations from 1995-2010 (Tilmes et al., 2012). Over the past decades, observations from
ozone sondes in different locations provide a valuable dataset of the evolution of ozone mixing ratios in the troposphere and
stratosphere. Figure 22 provides a Taylor diagram comparison between the C-GC, C-CC, and S-GC simulations of 2016 to the
climatology.





In general, C-GC does not perform significantly better or worse than C-CC or S-GC, producing mean biases and correlations
in each region/altitude combination which are within the same range. The clearest exception is at low altitudes (900 or 500
hPa) and mid- to high latitudes. In these regions, C-GC results frequently show a smaller normalized difference from the mean
(radius) than either S-GC or C-CC, but also a weaker correlation with the observed seasonal cycle. The C-GC simulation of
tropical ozone also shows the smallest mean bias at all altitudes at or below 250 hPa, although again showing a weaker
correlation.

At high altitude (50 hPa), all three models appear to perform similarly. This may however simply reflect the lack of spin up
time. Since the three models only simulated two years in total, the simulated stratosphere will not have had time to fully
respond to the new model configuration. Longer simulations are needed to fully evaluate the performance and capability of
the C-GC stratosphere.





*Figure 22. Taylor diagrams of the comparison of C-GC (red), C-CC (blue), and S-GC (black) simulations to a present-day (1995-2010) ozone sonde climatology. Top row to bottom row: comparisons at 900, 500, 250 and 50 hPa. Left column to right column: tropics, mid latitudes, and high latitudes. The normalized mean difference between simulations and observations for each region is shown on the radius, and the correlation of the seasonal cycle is shown as the angle from the vertical.*





### 3.3.3    Total column ozone and CO

Figure 23 shows the annual mean total ozone column, expressed in Dobson Units, as measured by the Aura Ozone Monitoring Instrument (OMI) and Microwave Limb Sounder (MLS). The results from the satellite observations are compared to results from C-GC, C-CC, and S-GC. We find that on average the results from C-GC are 7.8 DU lower than the observations, mostly driven by an overestimation of stratospheric ozone depletion during the Antarctic spring of up to 16 DU. C-CC predicts a total ozone column that is 6.6 DU larger than the global mean ozone column. When broken down by tropospheric and stratospheric ozone column, we find that the bias in the stratospheric and tropospheric ozone columns for C-GC is -2 and -6 DU respectively, compared to +9.5 and -2.5 DU for C-CC. Additionally, we find that the bias in seasonal variations of total column ozone as predicted by C-GC range between -16 and -6 DU, while the variations range from -3 to +7 DU for C-CC. The model results from S-GC predict similar geographical biases in total ozone column as C-GC, although with a smaller net bias of -3.3 DU.

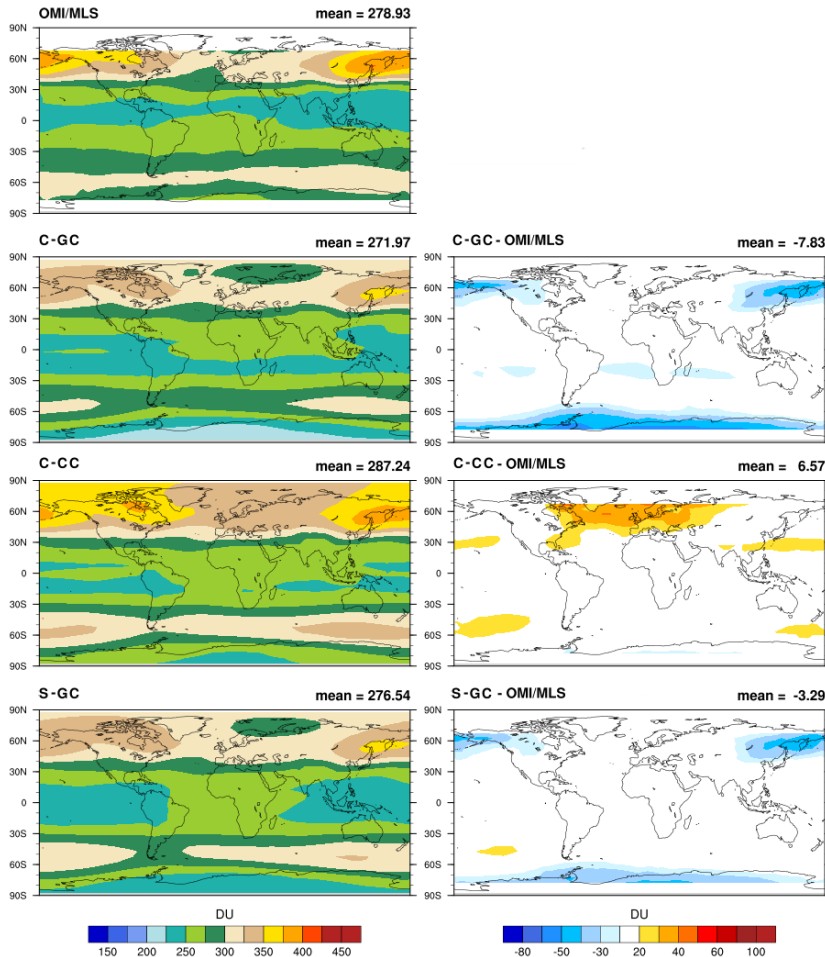

*Figure 23. Total ozone column as observed by the Aura Ozone Monitoring Instrument (OMI) and Microwave Limb Sounder (MLS) for the 2004-2010 time period (1st row), compared to the results from C-GC (2nd row), C-CC (3rd row), and S-GC (4th row) for the year 2016. The measurements and model results are presented on the left, while the model biases are shown on the right.*




912

Figure 24 compares the simulated total columns of carbon monoxide (CO) to retrievals from the MOPITT satellite instrument, averaged for each April in the period 2003 to 2012. The model results as well as the model biases are shown for April 2016. The CO model estimates using C-CC are characterized by a negative bias of -9×10$^{17}$ molec/cm$^2$ in the Northern Hemisphere that has been observed in previous model evaluations (Emmons et al., 2020). In C-GC, a negative bias still exists in the Northern Hemisphere, but is smaller at -5×10$^{17}$ molec/cm$^2$. Across all three model configurations a north-south gradient is observed in the model bias, with the bias in the southern hemisphere being approximately 10$^{18}$ molec/cm$^2$ more positive than the (negative) bias in the northern hemisphere. The results from S-GC are nearly identical to those in C-GC, with a smaller negative bias in the northern hemisphere than C-CC, but a larger positive bias in the southern hemisphere.

921

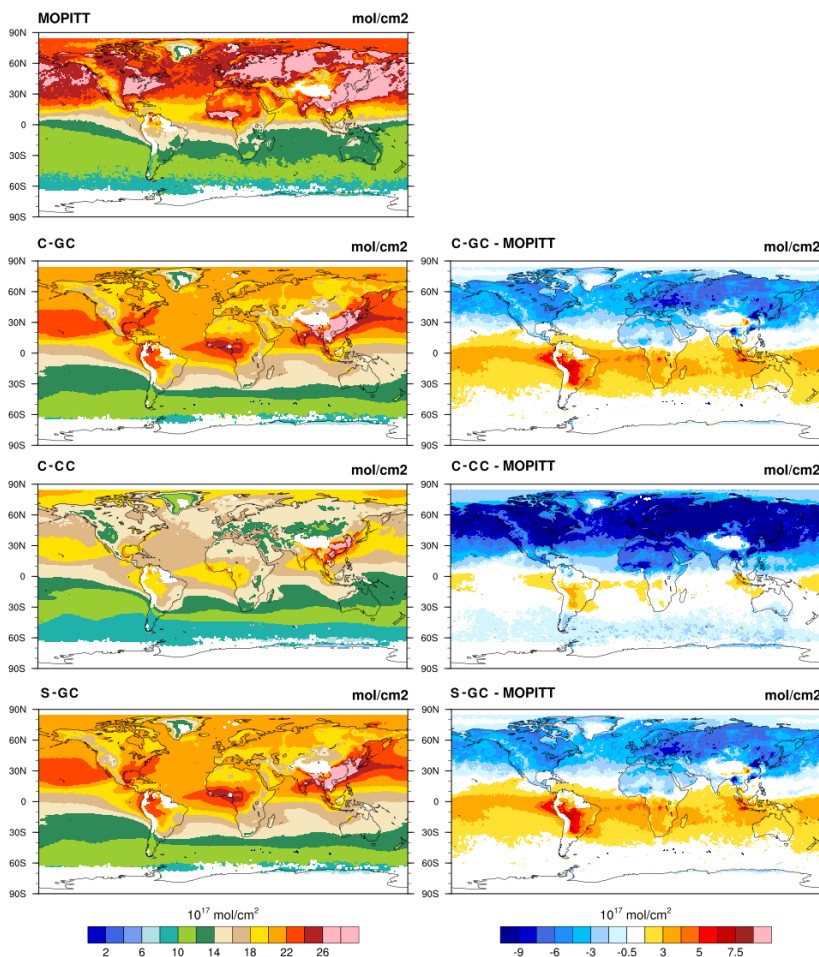

922

*Figure 24. Total carbon monoxide column during April, averaged from 2003 to 2012 and expressed in molecules per cm$^2$. The first row displays the satellite observations from MOPITT. The simulation results and biases are presented for C-GC (2$^{nd}$ row), C-CC (3$^{rd}$ row), and S-GC (4$^{th}$ row). The model evaluations are shown for April 2016.*





### 3.3.4 Wet and dry deposition tendencies

Finally, we compare simulated and observed surface deposition. Since deposition is the primary removal mechanism for atmospheric reactive nitrogen and sulfur species, the ability of a model to reproduce observed patterns of deposition provides an aggregate diagnostic for its representation of emissions, atmospheric chemistry, and the physical deposition processes.

Recent measurements have provided wet deposition rates in numerous geographical locations for the years 2005 to 2007 (Vet et al., 2014). Dry deposition fluxes are available from the same study but are limited to sulfur and nitrogen species. They are also limited to fewer geographical locations. Nonetheless we compare results from all three model configurations to the results from Vet et al. below.

Figure 25 compares the model-evaluated wet deposition rates of nitrogen at the surface for C-GC, C-CC, and S-GC. The total nitrogen flux is calculated by adding surface fluxes from each individual nitrogen compound undergoing wet deposition. Rainwater composition measurements are also displayed where available for comparison. We find correlation coefficients of 0.65, 0.66, and 0.67 for C-GC, C-CC and S-GC respectively with these observations. On average, the results from C-GC are closest to parity with a slope of 0.6, compared to 0.5 and 0.49 for C-CC and S-GC. We do not find any clear trends by location between the three models.

Comparing the dry deposition flux of nitrogen species at the surface from C-GC, C-CC and S-GC to in-situ measurements over North America from 2005 to 2007 shows that all models have positive biases. Relative to an observational mean of 1.57 kgN/ha/yr, C-GC has the best performance with a mean bias of +0.94 kgN/ha/yr, compared to +1.76 and +2.32 kgN/ha/yr from S-GC and C-CC respectively. These biases from all three models can be explained by either larger concentrations of nitrogen compounds or enhanced dry deposition velocities. However, we do not compensate for changes in nitrogen emissions between the time of the observations (2005-2007) and the simulated period, during which $NO_x$ emissions are estimated to have increased (Emmons et al., 2020).



950

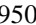

*Figure 25. Geographical distribution of the wet deposition flux of nitrogen for C-GC (top), S-GC (middle), and C-CC (bottom). The annual mean value is shown as a map for each model, with circles used to indicate observational measurements (left). A parity plot of the results against the rainwater composition measurements is also provided for each model simulation (right).*

Figure 26 displays the evaluated wet deposition rates of non-sea salt sulfur from C-GC, C-CC, and S-GC alongside measurements of sulfur in rainwater for 2005. When comparing across model results, we find a global mean deposition rate of 0.58, 0.38, and 0.50 kg S/ha/year in C-GC, S-GC, and C-CC respectively. The results from C-GC and C-CC show a correlation coefficient greater than 0.95, whereas C-GC and S-GC results show a correlation coefficient of 0.88.

Comparing to observational data, we find a mean bias of –2.40 kg S/ha/year (C-CC and C-GC) and –2.76 kg S/ha/year (S-GC) between the simulation results and rainwater composition measurements. This bias is location-dependent, with simulated data





for Asia showing a lower bias than North America or Europe. For instance, over North America, measurements indicate a
mean sulfur wet deposition flux of ~5 kg S/ha/year (for the year 2005), while the results from all three models at the same
stations reach a maximum of 1.5 kg S/ha/year (for the year 2016). This can be explained by the reduction in the sulfur wet
deposition surface flux over the past decades. Previous literature has found that the deposition rate of sulfur over the Eastern
U.S. has been decreasing at a rate of 1 kg S/ha/year[2] since 1990, with 60% of the reduction being in wet deposition rates and
40% in dry deposition rates (Zhang et al., 2018). Similar findings have been suggested for wet deposition rates over Europe
(Theobald et al., 2019). A similar, but more recent, decrease over Asia has also been observed (Aas et al., 2019).

*Figure 26. As in Figure 25, but now for non-sea salt sulfur. Rows: C-GC (top), S-GC (middle), and C-CC (bottom).*





It is difficult to say with confidence that the calculated bias is purely due to lack of recent data without new measurements to
support this conclusion. However, our results do show that the simulation of sulfur deposition in C-GC more closely follows
that in C-CC than that in S-GC. This could be due to either the simulated distribution of precipitation, the representation of
aerosol, or the representation of scavenging processes, all of which differ between C-GC (or C-CC) and S-GC.
**4    Discussion and conclusion**
We present the first implementation of the GEOS-Chem chemistry mechanism as an option in the Community Earth System
Model (CESM). In addition to allowing users of CESM to take advantage of advancements in atmospheric modeling
implemented into the GEOS-Chem model, this also allows the community to better understand why models disagree and how
progress might be made to improve model performance and accuracy.

Our results suggest that differences in the representation of tropospheric halogen chemistry – in particular the representation
and magnitude of emissions of short-lived bromine and chlorine sources – may be responsible for differences in simulated
ozone between these model configurations. However, in addition to the recognized differences in chemical mechanisms, subtle
structural differences in atmospheric models may have a significant role. Our evaluation of tropospheric ozone concentrations
suggests that one of the key drivers in differences between CAM-chem and GEOS-Chem ozone fields differences may be
differences in free tropospheric water vapor. Similarly, we show that sulfur deposition rates are approximately twice as great
when running GEOS-Chem in a standalone model as opposed to running GEOS-Chem embedded in CESM, despite the use
of identical emissions.

We also find that differences in the representation of wet scavenging are a significant contributor to differences in reactive
nitrogen and halogen species distributions between GEOS-Chem and CAM-chem. The unification of convective transport and
scavenging in GEOS-Chem helps to prevent movement of soluble species to the upper troposphere through convective
updrafts, and therefore limits the effect of near surface halogen emissions from sea salt on ozone at higher altitudes.

Our implementation of GEOS-Chem in CESM is now publicly available for use. We envision that this model can become a
powerful tool for research, forecast, and regulatory applications of global atmospheric chemistry, air quality, and climate
research. However, this is also an important step towards the Multiscale Infrastructure for Chemistry and Aerosols (MUSICA),
and thereof a truly modular Earth system model. By enabling us to fairly compare models down to individual processes, we
can begin to understand precisely why different models perform better or worse in reproducing different measurements and
accelerate our efforts to improve atmospheric modeling fidelity as a whole.





Finally, this work will foster collaboration between the GEOS-Chem and CESM-CAM-chem communities. GEOS-Chem is presently used for research by over 100 university groups worldwide, and CAM-chem similarly has numerous users and developers. The availability of GEOS-Chem as an option in CESM will stimulate broader interest in the GEOS-Chem community to use CESM, and in the CESM community to use GEOS-Chem. Indeed, we expect that on-line simulation of atmospheric chemistry will become increasingly attractive to GEOS-Chem users as the resolution of dynamical models increase, and that CESM will provide the principal vehicle for this because of its public availability and support. By enabling improvements developed for GEOS-Chem to percolate into CESM without the need for re-implementation, this work will accelerate progress in atmospheric chemistry and Earth system modeling.

**Author contributions**

TMF, SDE, HL, and EWL were responsible for the software development. TMF performed the investigation, formal analysis, and validation. SDE, LKE, SRHB, and DJJ conceived of the project and acquired funding. SDE, LKE, SG, SRHB, and DJJ supervised the work. TMF performed all visualization and prepared the original draft. Review and editing was performed by all co-authors. All contributions are defined according to the CRediT taxonomy (https://casrai.org/credit).

**Acknowledgements**

This material is based upon work supported by the National Science Foundation under Grant No. 1914920. We would like to acknowledge high-performance computing support from Cheyenne (doi:10.5065/D6RX99HX) provided by NCAR's Computational and Information Systems Laboratory, sponsored by the National Science Foundation. We would like to thank Mary Barth, Simone Tilmes, and Jean-François Lamarque for their assistance in understanding washout of aerosols in CESM. We also would like to thank Eloise Marais and Alma Hodzic for their help regarding the mapping of secondary organic aerosols.

**Code availability statement**

GEOS-Chem as an option within CESM is currently being implemented into the CESM main branch, such that no additional download will be needed to use it. However, a standalone copy of the specific implementation of CESM including GEOS-Chem which was used to generate the results in this manuscript is permanently archived at https://github.com/CESM-GC/CESM-GC-Standalone/releases/tag/v1.0.0_review (permanent DOI: https://doi.org/10.5281/zenodo.6465076). To reproduce the results of this work, the repository should be used as-is without using features such as checkout_externals to acquire any additional code.





**Competing interests**

The authors declare that they have no conflict of interest.

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
