# Peer review of "Implementation and evaluation of the GEOS-Chem chemistry module"

_EGUsphere, 2022_

## Referee Comment (RC1)

Review of "Implementation and evaluation of the GEOS-Chem chemistry module version 13.1.2 within the Community Earth System Model v2.1" by Fritz et al., submitted to EGUsphere, 2022.

Summary:
This manuscript documents the new capability of running the Community Earth System Model (CESM) with the GEOS-Chem chemistry online. Comparisons are made to the current coupled model configuration of CESM with CAM-chem chemistry module and the GEOS-Chem chemistry transport model when implemented for high performance computing ("GCHP"). This is an exciting development for both the GEOS-Chem community and the CESM community. The authors provided lengthy inter-model and observation comparisons. This work is highly relevant, and I support publication after my overall comments below are addressed as well as my minor and technical edits within the marked-up PDF are considered.

Comments:
This manuscript is very long and it is a lot to ask of your readers to commit to nearly 50 pages of figures and text. There is a lot of overlapping information in Section 2 (Coupling GEOS-Chem and CESM) and the start of Section 3 & 3.1 (Simulation setup) and I recommend the authors consider synthesizing the details, possibly restructuring these sections. The authors could then make Sections 3.2 and 3.3 their own sections. Given the authors referenced later sections when trying to explain the differences, the authors should reconsider if keeping the model intercomparisons separate from the observations is the best flow for this paper. There were times when reading the model intercomparisons I kept asking myself "which model configuration is closer to observations" and I had to wait to find out if even the model-to-observations comparison was provided. For the profiles of NOx, NOy, and some of the halogen species, there are satellite observations which could be used for validating at least the stratospheric portions (e.g., MLS, ACE-FTS). There are also ground-based and balloon-based observations of water vapor. The choice of climatologies for the sonde and satellite observation comparisons was not clear to me when observations for 2016 should be available to the authors (also no references were provided for these data sets). Be clear as to the reasoning behind the observations used for the validation section. By the end, I was also trying to find ways to reduce the figures, and suggest the authors consider if all panels and figures are necessary or could be included in supplemental information.

Especially in the model intercomparison, there is often an assumption of the reader's knowledge of atmospheric chemistry. Provide the chemistry background and references to support statements as to why different chemistry leads to differences between models.

Be careful quoting figures. I strongly suggest adding panel labels and referencing figure panels whenever possible in the main text. Often there are numbers quoted in a paragraph that I would have expected came from the figure currently being discussed but I do not find these numbers in the figure. In some places, this may have to do with number of significant digits used in the text vs the figure; but it is unclear. When results are discussed including a lot of numbers, I suggest the authors consider tables to make it easier to digest and compare the numbers between the different models (and regions). If numbers are provided but not from the figure being discussed in that paragraph state "not shown" so the reader does not spend time trying to find it.

The acronyms C-GC, C-CC, and S-GC are so similar it makes reading the comparisons hard to follow. The acronyms were only used in Section 3, not in Section 2 nor in the final Discussion and conclusion Section 4. I struggle to think of alternatives that may be better. Maybe using a lower-case **c** for CAM-**c**hem will help (e.g., C-GC, C-Cc) or include CESM instead of simply C (e.g., CESM-GC, CESM-Cc). Also, the standalone GEOS-Chem uses GCHP, so maybe simply using GCHP instead of S-GC would help it stand out from the CESM acronym. I had to keep reminding myself if it was using GCHP, not the Classic CTM, while reading the manuscript.

There are 169 comments in the following marked up version of the paper for the authors to consider addressing which I hope will help with clarity of the manuscript for final publication. In Acrobat Reader, the authors should be able to find the comments easily, but if there is any difficulty with this format, the authors can contact me.

[revised manuscript text omitted]

---

## Author Comment (AC1)

August 22nd, 2022

Dear Editor and Reviewers,

**Submission of "Implementation and evaluation of the GEOS-Chem chemistry module version 13.1.2 within the Community Earth System Model v2.1" to *Geoscientific Model Development**

Thank you for arranging this review of our work, and for your patience as we have worked to revise the manuscript. Since both reviewers noted the length of the paper, we have worked to both streamline it (by moving less critical figures to the SI) and to make it more readable by breaking specific analyses out into separate sections. We have also worked to make the paper more accessible to non-specialists through the addition of brief descriptions of key chemistry where relevant as requested by Reviewer #1, while aiming to make the paper more relevant to specialists through the addition of more technical data (in particular emissions) and diagnostic data (correlation coefficient tables). Although we recognize that the paper cannot cover all bases, we believe that these changes have helped to improve the relevance and breadth of impact of the manuscript.

As Reviewer #1's mentioned, several of their suggestions and questions took the form of comments which were placed directly on the manuscript itself. Most such comments are listed below with a response; however, for some minor comments, we have simply made the requested change without an explicit response below. These changes are reflected in the updated manuscript.

Please find below our point-by-point responses (in **bold**) to the review comments (in *italics*) and a markedup version of the revised manuscript. A "clean" revised manuscript will be provided through *GMD*'s submission system.

1

**Reviewer #1:**

**Summary:**

This manuscript documents the new capability of running the Community Earth System Model (CESM) with the GEOS-Chem chemistry online. Comparisons are made to the current coupled model configuration of CESM with CAM-chem chemistry module and the GEOS-Chem chemistry transport model when implemented for high performance computing ("GCHP"). This is an exciting development for both the GEOS-Chem community and the CESM community. The authors provided lengthy inter-model and observation comparisons. This work is highly relevant, and I support publication after my overall comments below are addressed as well as my minor and technical edits within the marked-up PDF are considered.

**Comments:**

This manuscript is very long and it is a lot to ask of your readers to commit to nearly 50 pages of figures and text. There is a lot of overlapping information in Section 2 (Coupling GEOS-Chem and CESM) and the start of Section 3 & 3.1 (Simulation setup) and I recommend the authors consider synthesizing the details, possibly restructuring these sections. The authors could then make Sections 3.2 and 3.3 their own sections. Given the authors referenced later sections when trying to explain the differences, the authors should reconsider if keeping the model intercomparisons separate from the observations is the best flow for this paper. There were times when reading the model intercomparisons I kept asking myself "which model configuration is closer to observations" and I had to wait to find out if even the model-to-observations comparison was provided.

As discussed in our opening comments, we have attempted to both shorten the paper somewhat and to make it easier to navigate. Any overlapping information between Section 2 and Section 3 has been removed from the latter. Section 3 is now purely about model setup as both Sections 3.2 and 3.3 have been moved respectively to Section 4 and Section 5. This aims to streamline the results section and make it clearer for the reader what is performed in each section. Section 4 is purely about model intercomparison, while Section 5 evaluates the model results against observations.

For the profiles of NOx, NOy, and some of the halogen species, there are satellite observations which could be used for validating at least the stratospheric portions (e.g., MLS, ACE-FTS). There are also groundbased and balloon-based observations of water vapor. The choice of climatologies for the sonde and satellite observation comparisons was not clear to me when observations for 2016 should be available to the authors (also no references were provided for these data sets). Be clear as to the reasoning behind the observations used for the validation section.

The missing references were an oversight, and we are grateful to the reviewer for pointing this out. The relevant references have now been added to the manuscript (lines 948 and 963). With regards to the use of climatologies, we chose this approach as they are part of the standard CESM evaluation package and this comparison is therefore likely to be familiar to users from the CESM community. We felt that this approach was reasonable when the focus was on evaluating differences between model configurations rather than the overall accuracy of the model. This justification is now provided on lines 922-924.

By the end, I was also trying to find ways to reduce the figures, and suggest the authors consider if all panels and figures are necessary or could be included in supplemental information.

We agree that some material is not critical to the main text. We have attempted to shorten the manuscript by moving less critical figures (including the figures showing wet deposition tendencies or surface  $NO_x$  concentrations, formerly Figures 12, 13, 14, and 20) to the Supplemental Information.

Especially in the model intercomparison, there is often an assumption of the reader's knowledge of atmospheric chemistry. Provide the chemistry background and references to support statements as to why different chemistry leads to differences between models.

We now provide some detail regarding the chemistry (including references) relevant to specific differences, with the goal of aiding the reader and supporting our statements. This includes referencing DMS chemistry in the context of oceanic sulfates (lines 626-628), how to interpret lower-stratospheric water and ozone differences (lines 552-555), the relationship between NOx and PAN (lines 715), and the importance of halogen chemistry to ozone (lines 747-749).

Be careful quoting figures. I strongly suggest adding panel labels and referencing figure panels whenever possible in the main text. Often there are numbers quoted in a paragraph that I would have expected came from the figure currently being discussed but I do not find these numbers in the figure. In some places, this may have to do with number of significant digits used in the text vs the figure; but it is unclear. When results are discussed including a lot of numbers, I suggest the authors consider tables to make it easier to digest and compare the numbers between the different models (and regions). If numbers are provided but not from the figure being discussed in that paragraph state "not shown" so the reader does not spend time trying to find it.

We have worked to make it clearer when we are not citing specifically from figures (e.g. stating "not shown" as suggested). We have also increased the number of tables in the manuscript (e.g. the new Tables 2 and 4), while trying to balance against the need to avoid further lengthening the manuscript.

The acronyms C-GC, C-CC, and S-GC are so similar it makes reading the comparisons hard to follow. The acronyms were only used in Section 3, not in Section 2 nor in the final Discussion and conclusion Section 4. I struggle to think of alternatives that may be better. Maybe using a lower-case c for CAM-chem will help (e.g., C-GC, C-Cc) or include CESM instead of simply C (e.g., CESM-GC, CESM-Cc). Also, the standalone GEOS-Chem uses GCHP, so maybe simply using GCHP instead of S-GC would help it stand out from the CESM acronym. I had to keep reminding myself if it was using GCHP, not the Classic CTM, while reading the manuscript.

Unfortunately we too were unable to think of acronyms which would work better. Our concern with using GCHP instead of S-GC is that the specific implementation of GEOS-Chem is not significant (GCHP and GC-Classic are essentially identical apart from their handling of transport), so highlighting it in such a way might be distracting. However, we have emphasized in the introduction that S-GC is the GCHP CTM, and not GC-Classic (see next comment).

**Specific comments:**

Is this referring to the GEOS-Chem Classic CTM or simply a free-running version of C-GC?

We now specify in the abstract that the acronym S-GC refers to the GEOS-Chem High Performance CTM (e.g. on line 393).

globally due to bromine or is this regional?

This statement (regarding the possible causes of differences in ozone) was incomplete. We have now clarified that there are other factors which may contribute, with variable roles in different regions (lines 32-34).

lower altitudes, but still in the stratosphere or in the troposphere?

We have now replaced "lower altitudes" with "in the troposphere" to clarify this sentence (line 32).

Could be rewritten, its a bit difficult to read.

We have clarified this sentence. The sentence now reads: "This difference in tropospheric ozone is not uniform, with tropospheric ozone in C-GC being 30% lower in the southern hemisphere when compared to S-GC but within 10% in the northern hemisphere. This suggests differences in the effects of anthropogenic emissions." (lines 35-35)

So now this sentence is referring to tropospheric ozone?

We now state that this sentence refers specifically to tropospheric ozone (lines 32-35).

i don't understand how the set up of the two versions is different from simply reading the abstract.

We now clarify that the MERRA-2 meteorology is used directly in the GEOS-Chem CTM (lines 39-40).

Change to NASA MERRA-2 Reanalysis meteorology, if that fits in the word limit for the abstract.

This change has been made (lines 39-40).

I'd say reference here but i see you have the url to GEOS-Chem on the following line. What about the Bey et al. 2001 reference?

Indeed, the URL has changed since we first submitted the manuscript. The manuscript now points to http://geos-chem.org (line 58) and we have added the relevant references for both GEOS-Chem (Bey et al. 2001 for GEOS-Chem and Eastham et al. 2018 for GCHP) and CESM (see lines 53-55).

This is true for GEOS FP but MERRA-2 resolution is coarser (0.5deg x 0.625deg).

We now state both meteorological reanalysis datasets with their respective horizontal resolution (lines 81-84).

Does it also have a world-wide network of research groups? Can you give it similar accolades as you did for GEOS-Chem? Is there a good url?

CAM-chem is widely used in the research community as it offers the capability to simulate tropospheric and stratospheric composition within CESM. In particular, users can choose to run CAM-chem with specified meteorology or as "free-running" (with climate feedback). We have now added URLs for both CAM-chem (line 116) and GEOS-Chem (line 58).

I usually recommend only using acronyms after they are defined but are these here so common to the community that they do not need definitions?

The acronyms in question (WACCM, ACCMIP, CCMI, POLMIP, HTAP2, GeoMIP, and CMIP6) admittedly vary in terms of how well they are known. However, all are better known by their acronym than by their full name, and none are used again in the manuscript. We have therefore opted to keep them as acronyms (lines 110-111) rather than potentially compromise readability. However, we would be happy to revisit this decision if necessary.

Is there a url which lists CAM-chem participating in all of these activities?

We have added the following URL (line 116): https://www2.acom.ucar.edu/gcm/cam-chem

I suggest you move sentence to start of next paragraph to better connect the ideas.

**We agree with the reviewer's comment. This sentence has now been moved to the beginning of the next paragraph (line 118).**

"...the only implementations of GEOS-Chem which are currently publicly available are either designed to run "offline"... or operate at regional scale and do not extend to global simulation": What does this mean? A researcher outside of GMAO can only run GEOS-Chem Classic CTM or GCHP, or WRF-GC?

This is correct. The only other relevant implementations are a frozen (and now outdated) version of GEOS-Chem in the Beijing Climate Center's model, and the implementation in GEOS which is (to our knowledge) only used (or usable) by researchers at NASA.

But you're going to try [to disentangle the root causes of these differences] here, right?

Throughout this study, we have tried to disentangle the source of the differences between the C-GC and C-CC model results.

but you just said you were comparing 2016. is there a spin-up?

We perform a spin up for each simulation setup but only compare the results for the 2016. We now state explicitly the spin up periods used (lines 402-406).

This is a good description of the schematic, but there are some aspects that are not clear. Why are there dashed boxes and arrows in the Chemistry section on the right, versus the solid arrows on the left. Why is Driver/Mediator in a dashed box? Why is the GEOS-Chem source code outside the box with the chemistry options? Are CAM-Chem and GEOS-Chem headers in a table? Dynamics is not connected to anything.

Figure 1 has been regenerated with the same kind of boxes and arrows to avoid any confusion. The large blue box on the right represents a blow-up version of the data exchange between the atmosphere state and the chemistry module (whether CAM-chem or GEOS-Chem). The GEOS-Chem source code stands outside of that blue box because it is not part of the CAM source code but rather downloaded for the GEOS-Chem Github repository. Dynamics are not directly applied through the control layer but rather acts on a "dynamics container" which are then translated to tendencies.

This is not labelled as such in Figure 1.

We now state in the manuscript that this "coupling interface" is labeled as "CESM2-GC interface" in Figure 1.

I assume this is a subfolder within the top level directory of the CESM2 code. Can this be written out in more layman's terms?

We now clearly state that the interface code is kept in a subfolder of the chemistry source code, and we have decided to keep the relative path to this subfolder within parentheses.

Using ESMF or NUOPC was supposed to make sure you could do clean testing within a modular system. How will this be accomplished otherwise?

At this time we are relying on the principle that the GEOS-Chem code base is common between all implementations, relying on interface codes which are now supplied as part of the GEOS-Chem base. We did this because we knew that CESM development is moving towards NUOPC, at which point we intend to update our interfaces to leverage that capability.

define SOA since the acronym is used later.

The acronym is now defined on first occurrence (line 199).

is this a general Modal Aerosol Model acronym or should it be MAM4? My brain defaults to spring March-April-May MAM.

The official acronym of the Modal Aerosol Model is MAM, with MAM3, MAM4 and MAM7 describing the number of modes used in the simulation. The CESM default is to run with MAM4.

Not sure if there is a journal preference but I recommend changing this to m-3

All occurrences of these in the manuscript have been modified to "m-3" (instances highlighted – e.g. line 229).

Is there a word missing here? "When implemented as the standalone model,"

Indeed, a word was missing. The sentence now reads: "When implemented as the standalone model, wet deposition is calculated for large-scale precipitation using separate approaches for water-soluble aerosols (Liu et al., 2001) and gases (Amos et al., 2012) with calculation of convective scavenging performed inline with convective transport."

Define

HNO3 is now defined as nitric acid.

This link did not work for me.

The new URL is now https://geos-chem.seas.harvard.edu/.

add (H2SO4) afterwards to correspond with use in Line 252

We have now added (H2SO4) after gas-phase sulfuric acid.

not yet defined.

H2SO4 is now defined (see previous comment).

what does this mean here? Are you comparing C-CC and C-GC in this table? Try to be consistent with acronyms.

The table lists the mapping in place between the GEOS-Chem and CAM-chem chemical representations. C-GC and C-CC refer to the fully-coupled CESM alongside its atmospheric chemistry module (GEOS-Chem or CAM-Chem).

Neither of these products have been introduced before. Change to "using the meteorological fields from NASA Modern-Era Retrospective analysis for Research and Applications, Version 2 (MERRA-2; Gelaro et al., 2017) and GEOS Forward Processing (FP; Lucchesi, 2018).

Since the MERRA-2 and GEOS-FP acronyms are now defined previously in the manuscript, the sentence now reads "using the NASA MERRA-2 and GEOS-FP meteorological fields" (line 309).

Trying to keep track of all the different module names versus the ESM name is not easy. CAM-Chem here doesn't have its own versioning, it is linked to the version of the CESM?

The Community Atmospheric Model (CAM) does have its own versioning system (C-GC and C-CC both use CAM6), but we are not aware of any subversioning system for CAM-chem. Each release of CESM is tied to its version of CAM. We have attempted to be more explicit in the revised manuscript (e.g. line 331).

Isn't this already stated on Lines 185?

Indeed, we have removed the double occurrence.

what does this mean? C-GC or C-CC? (this refers to "Whichever scenario is chosen")

The expression "Whichever scenario is chosen" refers to the choice between CMIP6 scenarios (e.g. historical or one of the SSPs). This is now clarified.

What does this stand for?

CIME stands for the Common Infrastructure for Modeling the Earth. The sentence now reads: "Additionally, we modify CAM, CLM and the Common Infrastructure for Modeling the Earth (CIME) such that the land model can pass land type information and leaf area indices to the atmosphere model to compute dry deposition velocities."

**when is this expected to happen? And on who's authority/guidance?**

We now say that this is currently being discussed with the CESM team. Although we interact on a monthly basis with the team, there are no hard guarantees regarding an implementation timeline (lines 370-371).

**Explain why [a complete copy of the source code is downloaded]**

To make sure that the most recent version of the GEOS-Chem chemical representation is used, a full copy of the code is downloaded when first setting up C-GC. Later updates of the GEOS-Chem code can be brought in using the Git version control tool. The sentence now is: "Although a complete copy of the GEOS-Chem source code is downloaded from the version-controlled remote of GEOS-Chem repository (to ensure that the most-recent release of GEOS-Chem is used), not all files present in the GEOS-Chem source code directory are compiled" (lines 377-379).

There's a fair bit of repetition in Section 3 opening paragraph and Section 3.1 to was already provided in previous Section 2. Given how long this paper is, I recommend revisiting these sections to reduce any redundant information/model descriptions. You may find you can change Section 3.1 to be a subset of Section 2 and then Sections 3.2 and 3.3 can be more stand-alone "result" sections.

Section 3.1 (which is now just "Section 3") only describes the model setup and we made sure that no overlap exists with Section 2, which focuses on the description of each module and their interface with the whole code. Additionally, Sections 3.2 and 3.3 have been moved to Section 4 and Section 5 respectively.

Should you be using C-GC and C-CC throughout Section 2?

We now use the C-GC and C-CC acronyms through all the sections in the manuscript for consistency.

this is odd wording. Did you preform a two-year simulation which you will evaluate against two other model simulations?

We understand that this sentence caused confusion, so we have removed the last few words. A two-year C-GC simulation was performed and evaluated against the other models and observations. The evaluations are described in the same paragraph.

It's odd to reference Section 3.2 and 3.3 without referencing Section 3.1. I advise rewriting this opening paragraph to introduce the models and then describe first a model intercomparison to establish the differences of C-GC to C-CC and then the C-GC evaluation against observations

**Sections 3.2 and 3.3 have now been moved to their own Section. Section 3 now just consists of the old "Section 3.1".**

How much does the choice of initial conditions impact the difference between the model simulations? How close are the initial conditions between CESM and S-GC? If the CESM initial conditions are used where S-GC doesn't have the fields, could they not have been used entirely to start the C-GC run to be consistent?

The initial conditions used for C-GC have been obtained using the existing C-CC restart file (which contains initial conditions for MAM aerosols, CAM-chem species, and some other meteorological data). However, any species involved in the GEOS-Chem chemistry scheme has been added to this C-GC initial file, using the S-GC restart file. This S-GC restart file is obtained after a decade-long S-

GC simulation. The simulation we performed with C-GC used a one year spin up period, using the same "default" restart file which is expected to provide a reasonable representation of the early 21st century (private communication). A brief description is now provided on lines 402-411.

Our approach allows for the troposphere to be reasonably well spun-up in all three configurations. However, given the model differences between S-GC and C-GC, we realize that stratospheric concentrations might need longer timescales to reach a quasi-equilibrium. Further work would be needed to quantify the impact of the initial spin up time.

Change to 0.01 hPa to be in the same units as the CESM v2.1.1 description in line 393.

The corresponding change has been made in the manuscript.

This is first mentioned at line 299, so it should be defined at first use.

We now define the MERRA-2 acronym on its first use in the main text (lines 84-85). Although we use the term MERRA-2 in the abstract, we do not define it there due to length concerns.

Add references for these schemes

These citations (Bogenschutz et al., 2013; Zhang and McFarlane, 1995) are now included.

not defined yet

The acronym CFC has now been defined. The sentence now reads as "Finally, for long-lived species such as chlorofluorocarbons (CFCs) we use the shared socio-economic pathway 2-4.5 (SSP2-4.5) set of surface boundary conditions in both C-GC and C-CC" (lines 478-479).

have these been defined as H2O and OH?

The sentence now states "water vapor (H2O) and the hydroxyl radical (OH)".

Based on this comparison being the third panel, I was expecting the middle panel to be discussed first.

We now generally evaluate the differences between C-GC and S-GC first, and then between C-GC and C-CC, thus following the order of the panels.

This result statement should come later where you discuss bromine, or possibly at Line 506, where you discuss reasons for all these differences.

This sentence has been moved to the corresponding paragraph and we now refer to Section 4.4 as a link to the NOx and bromine comparison Sections.

I encourage the authors to label the panels a, b, c.

We have regenerated each figure with a panel label as requested, and where possible refer to specific figure panels in the main text.

Are you trying to say [the absence of a specific pattern in the ozone delta] is likely related to transport in the online vs offline GEOS-Chem?

Not exactly – we meant more that differences might be related to different treatments of meteorological variables, rather than differences in transport (since all three model configurations use the same prescribed wind fields). We now specify that we are referring to, for example, the differing treatment of water vapor (lines 511-512).

Again, comparing first to C-CC but it is the third panel. Is there a reason to put S-GC in the middle?

We have aimed for consistency in panel ordering throughout the manuscript. However, in this case, we felt that the comparison between C-CC and C-GC resulted in a more logical flow for the comparison to S-GC, and thus began with that rather than a comparison to S-GC.

the next section specifically evaluates the surface ozone. I suggest changing this instead of "at the surface" but "from the surface"

**We agree with the reviewer's comment. This change has been implemented in the manuscript.**

this figure doesn't show an average. I suggest changing this to "across"

**We have made this change in the manuscript (line 30).**

The KORUS-AQ campaign is one location, not at all latitudes. Is there a better comparison, such as in Keller et al. (2021) evaulation, where we looked at ozonesondes across all latitudes. If you are specifically trying to highlight the GC vs CC, at least state the caveat that this is regional comparison and not global. [AND] Is this related to the KORUS-AQ study or the current study?

To further clarify that these observations are all made in one geographical location and to more clearly communicate the source of the observed differences, we have added the following sentence: "This suggests that discrepancies observed in KORUS-AQ may be related to chemistry rather than the treatment of meteorology, but a more focused regional analysis would be needed to confirm this" (lines 533-534).

I assume later you'll tell us why this might be. If not, can you provide here a reason?

The sections later in the paper go into detail as to why we observe this difference. We now state on line 544 that this will be discussed in Section 5.2.

**Explain why this would impact ozone. Is this seasonal signal that is dominating the annual differences?**

To further explain the role of stratospheric water vapor in ozone change, we have added the following sentence at the end of the paragraph: "This is unlikely to be due to  $HO_x$  catalytic cycles depleting ozone, as OH in this region is lower in C-GC than in S-GC (panel e) and  $HO_x$  cycles are in any case a minor contributor to ozone depletion in the lower stratosphere (Brasseur and Solomon, 2006). The greater water vapor (and therefore humidity) may instead result in faster heterogeneous chemistry, including the liberation of  $NO_x$  from  $HNO_3$ " (lines 552-555).

use "approximately" or "about" instead of a tildas.

**We have replaced all tildes with the word "approximately".**

[Figure 3 middle panel] Is this in reference to the red plume that goes up from the surface to 600 hPa? From the colorbar it is hard to know which color is the 10% to know if this is limited to 900 hPa to the surface.

Yes, this increase of 10% in northern mid-tropospheric OH mixing ratio is a reference to the plumelike feature in the figure. The percentage represents the relative difference between the two model results.

add "generally", as there are some red hot spots in both difference plots

**We have made the corresponding change.**

I don't see this. It looks darker over the southern ocean.

**This sentence was incorrect and has now been removed.**

What color is zero? It looks like a slight negative bias to me, with a slight positive bias over the eastern pacific near central america.

This comment refers to Figure 4, middle panel. This panel compares surface ozone as simulated by C-GC and S-GC. We agree that there is a small difference, rather than no difference. The sentence now reads as "In the Northern Hemisphere, we observe a small difference in surface ozone mixing ratio over the oceans (less than 1 ppbv), while a difference of approximately 3 ppbv can be found over North America, Europe and East Asia."

Is there such an asymmetry in bromine from sea salt between southern and northern hemispheres? What is the rational behind this statement?

Given that C-CC does not model sea salt bromine and that ozone is mostly lower over the oceans in C-GC compared to C-CC, we suggest that this difference may be induced by sea salt bromine. On the other hand, the difference between C-GC and S-GC mostly occurs in the Southern Hemisphere, which wouldn't be just explained by differences in sea salt bromine. We therefore believe that this difference may be driven by other factors, such as the larger anthropogenic influence in the Northern hemisphere.

the hemispheric differences seem to be on the order of 2.5 ppb, just in different directions depending on the model comparison. Are you arguing that the C-GC and C-CC difference is not asymmetric?

We agree that there is an asymmetry, but that it is weaker. We now clarify (line 590) that this is specifically in absolute terms, since the asymmetry should also be considered against the baseline hemispheric asymmetry in ozone calculated by C-GC.

*Is this suppose to quote the top left panel? The r2 value is 0.91 in Figure 5. i don't see this number in Figure 5. Is it supposed to quote the r2 value from top right?*

We indeed had a typo in this sentence. The new sentences are: "In January, we find a correlation coefficient of 0.91 and slope of 0.91 between C-GC and S-GC. In July this agreement is worsened, with a correlation coefficient of 0.80 but a slope of 0.93" (lines 599-600).

Is this an eye ball estimate from the figure or did you output this range? It looks like it starts before 15 ppbv in C-GC to me.

The range provided in the manuscript indeed started approximately around 11 ppbv. The sentence has been modified to the following: "There is also a distinctive "hot spot" in the July parity plot, with a large cluster of grid cells showing mixing ratios in the range 20-25 ppbv in S-GC but 10-20 ppbv in C-GC" (lines 602-603).

DMS has not been defined. Explain to the readers the chemistry behind this connection of DMS to Sulphate aerosols.

We have now properly defined the acronym DMS. We now provide a brief description of the relevant chemistry (lines 626-628).

These large saturated differences start at 700 hPa so I would say at least from the mid-troposphere. This would look lower down in the atmosphere if you used a log-scale for the pressure.

We agree with the reviewer's comment. We have replaced "upper troposphere" with "tropical midto-upper troposphere and extratropical lower stratosphere" (line 631).

Is there a reason that surface ozone got its own subsection but surface aerosols are lumped with the zonal means?

We chose to analyze surface ozone because of its importance to oxidation as well as its importance in air quality. We had originally planned to perform a similar analysis for PM2.5, but chose not to do so in order to save space. Such an analysis could however be included if desired.

Not all of your readers are likely meteorologists and will know where the ITCZ is located. I suggest describing this instead as "off the west coast of southern hemisphere continents" or something like that.

**We have made the recommended change on lines 639-640.**

Do you think this is because of the emissions in the southern latitudes as you state in line 587? Why do you think this may be? In Line 588 you claim "despite emissions of DMS from oceans".

We are not certain that these differences in surface sulfate concentrations in the ITCZ are caused by the same factors as the difference in sulfate concentrations between C-GC and C-CC in the Southern Hemisphere, in part because DMS emissions are calculated in identical fashion in all three simulations. The differences in surface sulfate concentrations could be explained by differences in a combination of factors such as deposition tendencies, aerosol representation, and OH abundance, but we have not been able to draw a firm conclusion thus far. Further work would be needed to identify the underlying factors causing these differences in surface sulfate concentrations. We now state that this issue is unresolved in Section 4.3.

I suggest this is changed to "mid- and upper tropical troposphere and throughout the lower stratosphere"

We agree with the reviewer and have implemented the corresponding change (lines 653-654).

NOx was first used at line 307 but never defined. It should be written out as NO and NO2.

 $NO_x$  is now first defined on first use (line 312) as NO and  $NO_2$ . In the rest of the manuscript, we refer to nitrogen oxides as  $NO_x$ .

Define which species you are adding up here to be NOy

We agree that this can lead to confusion given that GEOS-Chem and CAM-chem have different species included in  $NO_y$ . We have added the following sentence in the manuscript: "A full list of the species included in the lumped  $NO_y$  reservoir species can be found in the legend of Figure 10 for each model configuration" (lines 663-664).

For those readers who may be red/green color blind, I suggest replotting with different color choices or using dashed styled lines. Also could use the same colors as in the tailor diagrams.

We agree. Where possible, we have modified the plots so that red and green are not used in the same Figure (e.g. Figure 9).

How does the vertical resolution of the model configurations differ and impact the zonal mean comparisons here (and in previous sections).

All three models use the same vertical resolution. The only difference is that S-GC has a higher model top. Indeed, S-GC uses a 72-layer grid where C-GC and C-CC both use a 56-layer grid. In any case, the first 56 layers have identical pressure edges. This is stated in Section 3 (lines 419-423).

I tend to read from left to right so it does throw me off a bit that you usually discuss the right panel first.

We have rearranged our discussion such that the comparison is now made in the same order as the panels (see e.g. Section 4.4.1).

The units on the y-axis are hPa and I think it stops between 2 and 1 hPa.

There was indeed a typo in the manuscript. Figure 10 displays the speciation of  $NO_y$  as a function of altitude up to approximately 1 hPa.

I cannot see this subtle difference from Figure 10.

This comment refers to the following sentence: "At 10 hPa,  $HNO_3$  constitutes 20% of total  $NO_y$  in C-GC but 23% in both C-CC and S-GC.". These percentages were not read from Figure 10 but rather calculated from the simulation output. We now state in the manuscript (line 689) that these values are not shown explicitly in the figure, although they can in theory be read from the figure data.

Why are there two different legends? Does this mean your NOy estimates have different make up between the GEOS-Chem and CAM-Chem models? What impact does that have on your profile comparisons in the previous figure?

The list of species that make up NOy differs between GEOS-Chem and CAM-chem. Thus, S-GC and C-GC have the same list of species included in NOy, while it is a different one for C-CC. We now specify this when first mentioning Figure 10 (lines 663-664). The most significant difference is the lack of nitrate aerosols in CAM-chem, which affects the treatment of HNO3 (see e.g. lines 688 to 695) and is discussed throughout the manuscript. Otherwise the speciation of NOy does not appear to vary significantly between C-CC and C-GC (see e.g. lines 681-684).

**I don't see this purple band on Figure 11.**

This was a miscommunication; although  $N_2O_5$  is an important component at altitudes above (pressures less than) 200 hPa, at altitudes below (pressures greater than) this point, PAN is the key component. The discussion has been simplified (lines 700-702) to no longer discuss  $N_2O_5$  and instead focus on PAN.

Sometimes the authors use NOx and sometimes NO and NO2 but I don't think the authors have yet to define NOx as NO and NO2.

 $NO_x$  is now first defined early in the manuscript (line 312) as NO and  $NO_2$ . In the rest of the manuscript, we refer to nitrogen oxides as  $NO_x$ .

Can the C-CC legend be moved down slightly or stacked vertically so it doesn't overlap with the NOy speciation

We have regenerated this figure such that the legend does not overlap with the speciation.

For example, this seems redundant with what is in line 657. Can you better connect these two paragraphs as there is overlap in the message.

We have merged this paragraph into the previous one and attempted to reduce redundancy (lines 700 – 718).

Connect this paragraph with the previous paragraph. You give the reasoning for the result before you show/discuss the result. Alternatively, could Figure 12 and 13 be in a supplemental to support the reasoning you provide in Lines 680-686?

In order to reduce the length of the main manuscript, we have decided to move Figures 12, 13 and 14 to the Supplementary Information. The two paragraphs describing the results have been merged into the main text.

How does the difference in species between GC and CC impact these profiles? e.g., Br2

This is an important question given that GEOS-Chem and CAM-chem have different representation of bromine chemistry. The most critical difference is the lack of short-lived bromine source species in this version of CAM-chem, including sea salt, which means that bromine-related tropospheric ozone depletion is largely not captured. We now refer to this difference in the manuscript (lines 761-765).

I got tripped up reading this sentence because of the similar acronyms. Can you change this to "the C-GC total Bry concentration exceeds C-CC by 1000 % at the surface".

**We agree with the reviewer's comment and have made the corresponding change.**

**What do you mean here? Is the slope flattening or steepening?**

We agree that this sentence was confusing. We have now rewritten it as "Above 100 hPa, the averaged  $Br_y$  mixing ratio levels off, with values between 20 hPa and 2 hPa remaining roughly constant in the range of 16-20 pptv" (lines 767-768).

**above 80 hPa for C-CC comparison.**

We have added the following statement in the manuscript: "(and even above 80 hPa when compared to C-CC)" (lines 770-771).

**in GEOS-Chem?**

We now mention that this rapid washout occurs in C-GC and S-GC.

**The BrNO3 also looks bigger, less BrCl.**

We have added a sentence comparing BrNO3 and BrCl in the mid stratosphere. This sentence reads as "Larger mixing ratios of BrNO3 are also present in C-CC (approximately 10 ppbv at 30 hPa) compared to C-GC and S-GC (approximately 7.4 and 7.0 ppbv respectively at 30 hPa). Smaller mixing ratios of BrCl are observed in C-CC, with a mean value of 1.8 ppbv at 30 hPa, while they reach 3.1 ppbv in C-GC and S-GC at 30 hPa. The base causes of these differences are not clear, but may be related to the presence of more complex tropospheric and stratospheric halogen chemistry in the GEOS-Chem chemical mechanism (Wang et al., 20210" (lines 786-790).

"vertical" is redundant as profiles are not in the horizontal.

We agree with the reviewer and we have removed the word "vertical".

Why?

We decided to exclude long-lived chlorocarbons from the analysis of the profiles of reactive chlorine because they have longer lifetimes. However, they are of course included in the simulation.

How do the differences in species representing Cly impact the comparisons in Figure 17?

We find that above 500 hPa, all three simulations have similar chlorine speciation and species not included in CAM-chem do not have major contributions above 500 hPa. Below 500 hPa, the main differences are due to chlorine from sea salt, which is not represented in CAM-chem (~90% of the surface chlorine content, excluding chlorocarbons) (line 829-830).

**I do not follow this connection.**

We clarified the sentence. It now reads as "As with total  $Br_y$ , total  $Cl_y$  follows the same vertical distribution as S-GC up to 10 hPa. Above this pressure, the vertical distribution in C-GC is closer to that of C-CC" (lines 815-816).

I read the figures to have negative differences in the middle panel above 10 hPa. **and** Isn't this a negative difference in the right panel of Figure 17? and starting at 200 hPa for C-GC minus C-CC

These are indeed negative differences. The sentence now reads as "Above 10 hPa, the relative difference in  $Cl_y$  between C-GC and S-GC increases slowly from -2% at 10 hPa to -5% at 2 hPa, while the difference relative to C-CC remains at approximately -20% above 200 hPa" (lines 817-819).

How can you say this [a statement on production] when looking at a global annual mean? **and** The HCl and CINO3 look similar to me when comparing the left and right plots below 200 hPa (upper troposphere). What are you considering for the region to be lower stratosphere?

We find higher mixing ratios of HCI in C-CC between 200 hPa and 50 hPa. For instance, at 50 hPa, we find a mean HCI mixing ratio of 0.8 ppbv in both C-GC and S-GC, but 1.1 ppbv in C-CC. By the expression "appears to occur faster", we meant to say that there's greater mixing ratios of HCI at these altitudes in C-CC. We now state that "larger mixing ratios of upper tropospheric and lower stratospheric HCI from chlorine source compounds are observed in C-CC" (lines 831-832).

**Is there a reason to not use CO? Has it been defined yet?**

Carbon monoxide (CO) had not been defined previously. This sentence now reads as "Section 5.3 evaluates the level of agreement of simulated ozone and carbon monoxide (CO) columns to measurements from the OMI/MLS and MOPITT satellite instruments".

I missed the fact at first that each row was a different region. I highly recommend adding a detail to this sentence like "for North America (top row), Europe (middle row), and South-east Asia (bottom row)." Are the bounding boxes for these regions provided else where in the paper? If so remind the reader, if not add these details.

We now state that each row corresponds to a specific region. The sentence now reads "Figure 17 compares surface mass concentrations of  $NO_2$  as estimated by C-GC, S-GC, and C-CC for 2016 against ground station measurements for North America (top row), Europe (middle row), and South-East Asia (bottom row)".

**Have you tried comparing it to a "NOy" from the model to see if you get a better comparison?**

This is an interesting point and worth studying. However, we have not considered this possibility for the present study. Evaluating NOy model results against NO2 observations would maybe bypass the consistent overestimation that in-situ monitors generate due to HNO3. In Section 5.1, we have added the following sentence: "Comparing in-situ NO2 measurements against NOy model results could potentially remove the effect of interferants in the observations of surface NO2 concentrations, but is not considered here" (lines 870-872).

While Figure 19 clearly shows the models underestimate observations across the board, this regional discussion could be provided as a Table and I would then recommend Figure 19 to a supplemental. Other statistics like mean bias, RMSE to go with the correlation values should be considered (see Figure 3 of Keller et al., 2021 JAMES 10.1029/2020MS002413). Also keep in mind that comparing model grid boxes to point source measurements should always come with a caveat in the text that we do not expect perfect matches. I think you are saying something like that in line 828 but not specifically highlighting the grid-box representation errors.

I see later on you have the mean biases. I highly recommend a table for when you have this many numbers that you are quoting in the main text.

We agree with the reviewer's comment and now state in Section 5.1 "By comparing model results at an approximately 2° horizontal resolution to point observations, we expect some differences due to grid-box representation errors.". Additionally, we provide new tables (Tables 8 and 9) listing correlation coefficients of surface-level NO2 and ozone mass concentrations.

Are the NO and NO2 panels discussed? If not, can they be moved to a supplemental?

We now only present the panels corresponding to the NO:NO2 ratio, while the previous figure has been moved to the SI (Figure S3).

**What about referencing Figure 4 instead?**

**We now reference Figure 4 as suggested.**

Should we be able to see these in the NO2 or NO plots? Would a better color bar show this (e.g., log scale or less saturation below 5 ppbv)?

**The new Figure in the SI (Figure S3) plots $NO_2$ and $NO_x$ on a log scale. Ship tracks can be observed for all three models.**

Again, maybe a table would be more useful? or at least put the r values as insets on the panels?

We have added a new table listing the correlation coefficients between surface-level simulation results for ozone and measurements (Table 9).

The authors have already presented a lot of comparisons between the two models. Can the authors be more specific as to what further work needs to be done.

We now state throughout the paper different points that could be envisioned for future work (e.g. NUOPC-based interface, bromine sea salt emissions scaled with sea salt emissions, source of the differences in surface sulfate concentrations, role of heterogeneous chemistry on nitrogen speciation).

**Why not use available ozonesonde data for 2016?**

Since our goal was to show that CESM-GC produces reasonable results compared to CESM and GEOS-Chem, we used the analysis and processing codes (and datasets) which are typically used in evaluations of CESM and CAM-Chem. Several of the figures shown in the manuscript (including the ozonesonde comparison) have been generated using the CESM post-processor, which directly compares ozone profiles to a climatology of ozone sonde observations from 1995 to 2010. Even though a new climatology came out in 2016, the CESM post-processor still uses the previous dataset. To clarify, the manuscript now includes the time range of the ozone sonde measurements (line 922). We also clarify in the same section the reason for using climatology rather than a specific year.

We found that the choice of restart really made a difference on our stratospheric ozone in the GEOS-CF (GEOS ESM with GEOS-Chem chemistry) (see Section 4, Knowland et al., 2022 JAMES 10.1029/2021MS002852). How well spun up was the stratosphere in the restart file provided from version 13.1.2 GEOS-Chem (line 390)? What is the stratosphere like in the CESM initial conditions (line 389)?

The GEOS-Chem restart file was obtained from a 10-year simulation using GEOS-Chem Classic on a 4°x5° horizontal grid, which was then regridded to the horizontal grid used for this study. We agree that performance of the model stratosphere will be influenced by the restart file chosen, but we do not evaluate that influence in this work. In Section 3, we now state "For C-CC, the standard restart file provided with CESM is used to provide initial conditions. For S-GC, we use a restart file provided with version 13.1.2 of the GEOS-Chem chemistry module, which was obtained from a 10-year simulation. The CESM restart file is intended to represent the early 21st century, so we have followed the lead of previous studies which have used a 1-2 year spin up period (Schwantes et al., 2022; He et al., 2015)" (lines 402-406).

Figure 23 caption states that the observations are for a different period than the model but i do not read that here. State that these comparisons are to climatologies, like in the ozonesondes. My question again though is why not use the actual observations for this year as they are available?

We now address this issue directly in the text (lines 922-924 and 947-956). We agree however that an additional evaluation looking at the performance for a specific year would be a useful supplementary analysis.

Is this shown in Figure 23? If not, state so.

It is not, and we now state this explicitly in the text.

not shown

We now state that these data are not shown in Figure 23 (now Figure 20).

not shown. Again, would these numbers make for a good table?

We now state that this number is not shown in Figure 23 (now Figure 20). Although we agree that a table would be helpful, we have chosen not to create one so as to avoid further lengthening the manuscript.

This should have been defined at first use and then CO can be used throughout. The authors do not make a confusing reference to Colorado which would make using CO for carbon monoxide confusing.

Carbon monoxide (CO) is now defined on first use in Section 5. All other references to carbon monoxide have been replaced with CO.

**of CAM-Chem**

The sentence has been modified accordingly and now reads as "The CO model estimates using C-CC are characterized by a bias of -9×1017 molec/cm2 in the Northern Hemisphere, consistent with previous evaluations of CAM-chem" (lines 962-970).

Capitalize [Southern Hemisphere]

We have now capitalized Southern and Northern Hemisphere throughout.

I don't follow this statement.

We have clarified the sentence. It now reads as "Across all three model configurations a northsouth gradient is observed in the model bias, with the bias in the southern hemisphere being approximately 1018 molec/cm2, which is of greater magnitude than the (negative) bias in the northern hemisphere."

**Are these numbers on Figure 25?**

The numbers are included in Figure 25 (now Figure 22), but respectively rounded to 0.6, 0.7 and 0.7.

**Is this in a figure?**

We have decided to not show the dry deposition fluxes of nitrogen for all three simulations, but rather state the results in the manuscript. The dataset we used has few measurements of dry deposition fluxes and we decided to not include this figure, as the results can be best described in a few sentences. This is now stated in the text in Section 5.4.

**Are these numbers from the Figure 26?**

These numbers represent the global mean for each of the simulations and they represent thus the mean of the panels on the left in Figure 26 (now Figure 23). This is now stated in the text in Section 5.4.

Are these numbers in the Figure?

The correlation coefficients between model results are not displayed in the figure.

Did the authors expect these numbers to match?

Wet deposition of non-sea salt sulfur is handled by MAM in C-CC and C-GC. Since the same representation of aerosol microphysics and wet deposition is used for sulfates in both C-CC and C-GC, we expected these numbers to be similar.

Am I reading this correctly. Right hand panels, mean of the red dots? The maximum for the model seems less for S-GC than the other two models.

The reviewer is correct. The maximum for S-GC is lower than for other models. We have modified the sentence to "For instance, over North America, measurements indicate a mean sulfur wet deposition flux of approximately 5 kg S/ha/year (for the year 2005), while the results at the same stations are lower with the slope of the linear fit equal to 0.2, 0.1, and 0.2 for C-GC, S-GC and C-CC respectively" (lines 1015-1017).

not only university groups but also government agencies for research

The sentence has been modified accordingly and now reads as "GEOS-Chem is presently used and developed worldwide for research by over 100 university groups and government agencies" (line 1058).

A Data Availability Statement is missing. In the main text, there is no reference to the source of the observation datasets for the ozonesondes, OMI, MLS or MOPITT. Web address and DOIs should be provided.

A Data Availability Statement is now included below the Code availability statement. Web addresses and DOIs are now provided both in said statement, and we provide references (e.g. Deeter et al., 2014 for MOPITT data and Ziemke et al., 2011 for OMI/MLS data) in the main text.

**Incomplete reference [Fast-JX]**

The reference in question has been removed, as there is no journal paper to cite.

**Reviewer #2:**

Fritz et al present a landmark overview of a well-designed and implemented configuration of the Community Earth System Model incorporating the GEOS-Chem module. This constitutes a significant technical achievement and represents a very impressive step forward in model capability. This paper fits well within the scope of GMD and I believe is suitable for publication after consideration of the manuscript structure and some further thought is given to the level of detail of the discussion.

As the authors note on L377, the use of the same host ESM allows the differences in results to be attributed to the two chemistry modules. The new configuration offers the possibility to perform interesting chemistry module intercomparisons and offers the possibility to work towards a better understanding of the role of the chemistry scheme, and other processes connected to chemistry (aerosol processes, wet and dry deposition), in determining model performance and intermodel differences.

At this stage, that goal is still someway off, which is understandable given that this is the first paper from this project. There remains a number of differences in the implementation of key processes that inevitably lead to intermodule differences, and it will require further work to unpick the role of, say, the different dry deposition or aerosol schemes in driving differences between the two modules. The CESM-GEOS-Chem framework does allow this work to begin, but the authors might wish to say more about what possibility exists to harmonise further these key processes between modules and to further increase the modularity of the chemistry schemes. This would better facilitate being able to swap between chemistry module process-level treatments to improve attribution which is an important goal, and I would say is the most important potential outcome of this work.

This is a big paper that is doing the work of two or three: it is a description of the technical changes required, a description of the model configurations, a model/module intercomparison paper and a model evaluation paper. This is not to criticize, but it does serve to illustrate the rather huge task of the essential role of model description and evaluation. However, I do wonder if the paper has become rather overlong.

We have worked to reduce the length of the paper by moving several figures to the Supplementary Information, while also aiming to improve the paper's clarity and structure. Section 3 now only discusses the model setup, while Section 4 (previously 3.2) and Section 5 (previously 3.3) go into detail about the model intercomparison and evaluation against observations respectively. Our hope is that this will help the reader to navigate and reduce fatigue.

The evaluation itself is often rather cursory and little is gone into in detail. This puts the success of the paper in some jeopardy - the scope is impressive but the level of detail occasionally leaves the reader hanging, and for specialists it does run the risk of being rather unsatisfactory.

We understand the reviewer's concern, and addressing this has been a significant focus of our revisions. As mentioned above, we have aimed to improve the structure and have moved material which is not critical to the paper into the SI, with the goal of streamlining the reader's experience. However, we have also aimed to improve the level of detail, as will hopefully be evident from our response below. This includes providing more information on total emissions; providing additional context on the likely chemistry causing differences observed between models; and breaking out diagnostic data such as correlation coefficients into separate tables so that they can be more readily accessed and interpreted by specialists. We hope that this has helped both to make the work more accessible to a broad audience and to improve its interest to specialist readers.

The paper performs a comparison between model configurations using zonal mean O3, surface O3, aerosol mass concentration, NOy, Bry and Cly, as well as an evaluation agains observations of surface NOx,O3, ozone profiles (2016 model year vs climatology), satellite O3 (2004-2010 period for troposphere, stratosphere and total column), total CO column (2016 vs 2003-2013 climatology) and wet/dry deposition

(fluxes at various stations 2005-2007). The use of different observational periods for the intercomparison could presumably be addressed with a longer transient, but the text is reasonably caveated on this point.

We understand the reviewer's concern on this point. We have tried to, wherever possible, clarify which observations and which time period are used to compare against model results. In our study, we have used some tools provided by NCAR to post-process results from C-GC and C-CC. Since our goal was to show that CESM-GC produces reasonable results compared to CESM and GEOS-Chem, we used the analysis and processing codes (and datasets) which are typically used in evaluations of CESM and CAM-Chem. Several of the figures shown in the manuscript (including the ozonesonde comparison) have been generated using the CESM post-processor, which directly compares ozone profiles to a climatology of ozone sonde observations from 1995 to 2010.

The intermodule/model comparison is really interesting. The paper describes a whole atmosphere chemistry scheme, and so some whole-atmosphere evaluation is performed, particularly for O3/NOy/Bry/Cly. I think the impact of the structural differences in the model is probably the main result in this paper - wet deposition and Cly/Bry sources are frequently mentioned - so breaking discussion down into C-GC vs C-CC for most of the evaluation and considering in a separate section the offline S-GC runs might make things a bit simpler to follow, not least as there are huge differences arising from the different meteorologies that frequently dominate the S-GC runs, making the comparison not one between modules but more between models at a high level, i.e. between CTM-style offline meteorology and GCM-style free-running experiments, which is interesting but perhaps muddies the waters.

In light of this comment, we experimented with multiple different possibilities for the document structure. Although we agree that the proposed structure (C-GC v C-CC first, with S-GC comparisons separated out) would likely make parts of the analysis much cleaner, we found that it was difficult to avoid repetition in diagnosing model differences. We have therefore opted instead to split the intermodel and intermodule comparison into its own Section (now Section 4), with the comparison of model results to observations now separated into Section 5. We have also worked to improve the degree to which the sections are "signposted", so that interested readers can more easily find the component of the analysis which is most relevant to them.

Better understanding the drivers of inter-module differences would be welcome. I think the manuscript would be improved significantly by examining not just the levels of key species but also the factors controlling the level of their reservoirs in more detail. The manuscript would be improved significantly if this would go further and address the species' budgets, quantifying the inputs and outputs between the modules. While biogenic emissions are compared, it would help - from an ozone evaluation point of view - to add data on other ozone precursors such anthropogenic, soil and LNOx to this table. Similarly, sink terms in the ozone budget would also be beneficial. A table similar to Table 1 in Tilmes et al. Geosci. Model Dev., 9, 1853–1890, 2016 would be ideal for the purposes of comparison. Putting more results into such tables would be helpful for the specialist reader.

We agree and think that this is a valuable addition to the paper. We have thus sought to extract more information from our model configuration data which might inform at least the source magnitudes. Unfortunately, we have only limited output data available from the original simulations (which we do not have the resources to re-run), and are thus not able to extract much additional data such as loss rates and lifetimes (in particular, we do not have this data consistently across all three). However, we were able to provide more information on total emissions. The new Table 2 provides total annual NOx emissions from anthropogenic, soil, and lightning sources; the new Table 4, shows total annual surface emissions of key aerosols (sulfates, primary organic matter, and black carbon). The values from the model results are consistent with the ones provided in the study from Tilmes et al. (2016). These complement the existing data and tables showing biogenic VOC emissions (Table 3), sea salt emissions (Table 5), dust emissions (Section 3 text), and the burden data shown throughout the results sections.

Similarly, ideally, where key parameters or processes are identified, it may be useful to add references that indicate how the model configurations/chemistry was tuned/optimised when that model configuration was produced (e.g. is it possible to say how the sulfate dry deposition was evaluated originally in CESM2 and GEOS-Chem that means the deposition rates are so different?). This would give some traceability of the model configuration to the evaluation paper.

We have added throughout the manuscript references to the key model papers which describe the technical basis and implementation of specific processes. This includes references to the relevant implementations of stratospheric chemistry (lines 487-488) and halogen chemistry (lines 493-495), and specific references to key differences where appropriate (e.g. lines 788-790).

**Specific points**

The level of detail is rather variable in section 2.1

We have modified Section 2.1 to try and maintain a more consistent level of detail. In addition, we have attempted to streamline the manuscript by avoiding duplication of information in Sections 2 and 3.

Figure 1 and L161-165 - I am not sure of the timing of the various calls to dynamics, physics and chemistry - can the authors expand on why dynamics does not modify the atmospheric state in the diagram? What order are the routines (physics/chemistry) called in?

The result of dynamics is not directly applied through the control layer but rather acts on a "dynamics container" which are then translated to tendencies. This is now stated in the caption of Figure 1. We have now added a brief but more detailed description of the order of calls in the Supplemental Information.

Section 2 would benefit from a summary table that lists configurations side by side, e.g. aerosol scheme, dry deposition, as in e.g. the supplementary to Turnock et al., Atmos. Chem. Phys., 20, 14547–14579, 2020.

We have added a table summarizing the model configuration (Table 6).

Section 2.3.4 repeats some of the detail in L328 and L185.

We agree that this was somewhat redundant. We have now removed the second occurrence of this statement.

L366 missing words after to ensure

The sentence now reads as "Although a complete copy of the GEOS-Chem source code is downloaded from the version-controlled remote of GEOS-Chem repository (to ensure that the most-recent release of GEOS-Chem is used), not all files present in the GEOS-Chem source code directory are compiled" (lines 377-379)

L470-481 if the authors prefer to keep the three-panel structure (see comment above) it would be helpful to describe the figures in the same order that they are presented (L-R)

We agree with the reviewer. We have kept the same order for all Figures (C-GC, S-GC, C-CC) and we have thus modified the text to describe the results in that order. We also now label all figure panels.

L587 'emission' regions?

We have made the corresponding modification (line 626).

L617 what understanding of the ozone and aerosol do the subsequent analyses aim to improve? What beyond assessment of model skill does the comparison with observation aim to do?

The subsequent analysis (i.e. Section 5) aims to evaluate the model results against both surface ozone concentration measurements and vertical profiles so that we can understand what the dominant factors are in surface ozone when simulated by C-GC, and whether C-GC broadly moves us away from or towards the observations relative to either model. In this sense the reviewer is correct that this paper is concerned more with model skill than with a deeper understanding of atmospheric chemistry, which we hope will be gained through future applications of the model.

**L648 are the aerosol reactive tendencies stored? Can this be further assessed?**

The aerosol reactive tendencies were unfortunately not stored when the simulations were performed. Further assessment of the role of heterogeneous chemistry on  $NO_y$  partitioning in C-GC would be an interesting follow-up to the present study. We now state this in Section 4.4.1.

**L657 reads strangely**

The corresponding sentence has been split in two and now reads as : "However, between 200 and 900 hPa the dominant contributors are  $HNO_3$  and PAN [peroxyacetyl nitrate]. In this pressure range, the C-GC and S-GC simulations also show a significant contribution from nitrate aerosol (NIT) and  $BrNO_3$ ."

We would again like to thank the reviewers for their time and insight, and believe that their input during this review process has improved the paper substantially. Thank you again for considering our manuscript for publication in *Geoscientific Model Development*.

Best wishes,

Sebastian Eastham

**1** Implementation and evaluation of the GEOS-Chem chemistry module**

**2 version 13.1.2 within the Community Earth System Model v2.1**

Thibaud M. Fritz1, Sebastian D. Eastham1,2\*, Louisa K. Emmons3, Haipeng Lin4, Elizabeth W.
 Lundgren4, Steve Goldhaber3, Steven R. H. Barrett1,2, Daniel J. Jacob4

[revised manuscript text omitted]

---

## Referee Report (RR1)

Second Review of "Implementation and evaluation of the GEOS-Chem chemistry module version 13.1.2 within the Community Earth System Model v2.1" submitted to Geoscientific Model Development by Fritz et al.

The resubmitted draft describing the new implementation of GEOS-Chem within CESM is much improved over the initial submission. I recommend it for publication after my minor comments and technical edits listed below are considered. The line numbers refer to the *marked-up version* that followed after the authors' comments to the reviewers and not line numbers in the submitted draft as I did not have the new draft when I started my review and was unfortunately without internet at that time.

Line 55: The authors need to check references throughout the manuscript are in alphabetical or chronological order. Often there is a mix. For example, "Community Earth System Model (CESM) (Hurrell et al., 2013; Tilmes et al., 2016; Lamarque et al., 2012; Emmons et al., 2020)"

Line 64 : extra space at the end of sentence before the period.

Line 82: GEOS FP does not have a hyphen ( see https://gmao.gsfc.nasa.gov/GMAO_products/NRT_products.php) and this should be corrected throughout the manuscript.

Line 83-85: GEOS FP and MERRA-2 should have references to Lucchesi 2018 and Gelaro et al. 2017 (https://doi.org/10.1175/JCLI-D-16-0758.1), respectively. Also, is there an extra space after output and before provides? ("output provides data")

> Lucchesi, R., 2018: File Specification for GEOS FP. GMAO Office Note No. 4 (Version 1.2), 61 pp, available from http://gmao.gsfc.nasa.gov/pubs/office_notes.

Line 87: Should ESM be redefined, or is it assumed folks know it from the CESM definition?

Line 99: The Keller et al. 2017 reference is incomplete. My assumption is it is redundant to the Hu et al reference so the Keller et al. 2017 should be removed and only the Keller et al 2021 reference should be included.

Line 101: Goddard Earth Observing System (GEOS) (Bey et al., 2001; Eastham et al., 2018) is redundant (to text above as GEOS already defined) and confusing (as readers could mistake Bey and Eastham references for GEOS meteorology references). I suggest removing references and "Goddard Earth Observing System".

Line 105: CESM already defined and references provided.

Line 107: "observed meteorology including from GEOS", I suggest changing observed to analyzed, unless you can actually nudge towards observations.

Line 119: Earth system models is written out instead of using "ESM"

Line 141: "Sections 4 and 5 present a one-year simulation (not including spin up)". I know I suggested to make it clear that there was a spin-up vs the year used for evaluation in my initial review but I think this is confusing as it is written, such that one might wonder if no spin-up was included. Instead, maybe something like "Results from the one-year simulation (following the appropriate spin-up) performed for each model configuration 1) CESM with GEOS-Chem, 2) CESM with CAM-chem, and 3) the stand-alone GEOS-Chem CTM are presented with includes model intercomparison in Section 4 and evaluation against surface and satellite measurements in Section 5. Final discussion and conclusions are in Section 6.". Note, I restructured the sentence and included a final sentence for Section 6.

Line 182: Should HEMCO have a Keller et al. reference? Line 301 redefines HEMCO and provides the Keller et al. 2014 reference.

Line 202: "OH, ozone and NO3". OH and NO3 should be defined. OH is defined later on line 486.

Line 206,229: The unit has a period after gram instead of a space.

Line 249: MAM4 already defined so not necessary to do here.

Line 315: NOx is defined as NO+NO2 but then NO is used a few lines later without being first defined as nitric oxide.

Line 322: Sulfur dioxide not defined first before SO2.

Line 331: Should this say "since iodine chemistry is not explicitly modeled" or do the authors mean only "iodine"

Line 346: "(methane, N2O, and chlorofluorocarbons)". When are these used again and if so should they be defined? N2O is not defined yet. CFCs defined at line 478.

Line 377: The previous paragraph also begins with "Although". I suggest changing one of these to an alternative word, like "While" (or "Whilst" if you prefer the British English)

Line 387: "Then, we perform a comparison of its output to that generated by two other model configurations (Section 4)". I suggest adding something to this sentence so it reads to the effect of "the two conventional configurations for CESM-chem and GEOS-chem". Otherwise, the "other model configurations" is ambiguous as the next sentences states "by comparing these results to C-CC", it is unclear if the "results" is just the C-GC or C-GC plus these two other model configurations.

Line 398: I suggest changing "data" to "output" as this is from model simulations, and not observations. Does "This section" in Line 399 refer to Section 3 or Section 5?

Line 402: I suggest changing "All simulations cover January 1st to December 31st 2016, with an additional 6 months (S-GC) or 1 year (C-GC/C-CC) of spin up" to "Following a model spin-up period (6 months for S-GC and 1 year for C-GC and C-CC), the 1-year period of January 1 to December 31, 2016 was suitable for multi-model evaluation."

Line 415: could state this is the stratopause

Line 423: Are the truncated levels the first 56 layers of the GEOS grid and then are they somehow regridded to match the 56 hybrid pressure levels of CESM? Can the authors clarify this?

Line 438: CEDS should be defined and reference provided, especially as a new version of CEDS was recently released.

Line 453: Is there a reference for AeroCom?

Line 506: What is the pressure range in Figure 2? It is hard to know from the y-axis labels. Some of the other vertical plots have the top of the y-axis given in the Figure caption. Check this is given in all captions.

Line 515: Remove "hydroxyl radical" as OH already defined.

Line 531: Should "ozone" be clarified to be tropospheric ozone or UTLS or which levels did the Park et al study look at?

Line 546-557: I am struggling with the structure of these two paragraphs. The bigger differences are in panels b and i, yet this first paragraph focuses on (h) and then the second on (i). I suggest reconsidering the flow of the paragraphs to something like:

> There is a clear link between the ozone distributions and water vapor. Outside of the tropics and below the tropopause (Figure 3i), water vapor concentrations are up to 30 % greater in C-GC than in S-GC…<continue as written>…(an indirect sink for ozone). While ozone concentrations are uniformly lower for C-GC than for C-CC (Figure 3c), water vapor concentrations are uniformly greater for C-GC than for C-CC (Figure 3d). This is not surprising since the representation of moist physics in the two models is identical. However…<continue as written>…HNO3. Differences in ozone related to tropospheric NOy and halogens will be explained in detail in Section 4.4.

Lines 565-569: Is it differences in moist processes or transport?

Line 576-577: I am struggling to see this link.

Line 590-595: What do the authors mean by "does not show the same hemispheric asymmetry in absolute terms"? The difference over the southern hemisphere is about the same, it is really the northern hemisphere that is different. Is the emission of bromine from sea salt limited to either hemisphere or could it be driving the differences seen in the arctic which looks bluer to me in panel c than panel b. In the northern hemisphere, the C-GC and S-GC are fairly close (2.2 ppbv difference). Are the anthropogenic emissions driving the differences in the northern hemisphere or the southern hemisphere?

Line 602-604: This hot spot is also seen in both C-CC v C-GC plots. So is it almost more remarkable that January S-GC v C-GC does not have it? I wanted to link it to the hot spot over the amazon, but I got my sign wrong. Is there a clear feature in the C-GC minus C-CC in Figure 4 that is consistent all year round that could be linked to this hot spot in the bottom panel of Figure 5 that could partially explain July top panel? (if not, then this falls under the need for future work).

Line 626: reference Figure 7 when claiming the differences are greatest over the oceans, as this cannot be determined in Figure 6.

Line 626-628: The link to OH differences, is that still for below 800 hPa or throughout the troposphere? I do not see a clear link between Figure 3f and Figure 6c regarding the higher sulfate aerosol mass in the southern latitudes, but I can see patterns that match the OH zonal plots elsewhere in the troposphere.

Line 629: Can you quantify the "more closely follows"? I think this is the differences of +/- 25% quoted at the end of the paragraph but I think it should be quoted earlier. Can you give a reason by the C-GC and S-GC differ so much? Is it the difference in the bulk representation vs summing across all aerosol size bins? Oh, based on my latter reading of line 654 saying it is the convective scavenging, I think you may need to check here in line 630 if this should be broken into two sentences or at least properly reference (b) and (c) as I missed here the authors were referring to the 'differences in the representation of convective scavenging" was now discussing panel (b) and not still panel (c).

Line 644: In the previous paragraph there was a link to DMS for these greater concentrations and OH. Does that not apply here?

Line 655-657: it is not clear to me why POM would be greater in C-CC if the emissions of POM in C-CC are 29% lower.

Line 674: except the two points just above 50% (as mentioned in line 677). May be best to change your upper bound to match or give the range (looks like about -25% to 55%).

Line 688-691: Is the difference at 10 hPa really significant? At 200 hPa, the 60 and 63% are also only 3% different and read like they are about the same, but the striking difference is the much

greater 78% in C-CC at 200 hPa. Is it fair to open the paragraph with "partitioning between NOx and HNO3 differs significantly between the three models"? Is there a point where the difference becomes significant?

Line 693: Can this statement on sulfate aerosol size distribution be linked to Figure 6 at all?

Line 704: At 500 hPa, I cannot see any NOx in the figures. Given the previous sentences say that from 200-900 hPa HNO3 and PAN are dominant and the following sentence says NOx once again becomes significant contributor, is there much NOx contributing at 500 hPa that it is worth listing it here in the 78%, 85% and 97%? Or is this to be consistent with what is in line 702?

Line 706-707: This sentence is confusing as it opens talking about lower altitudes but then ends referencing 200 to 300 hPa and then the new text again refers to lower altitudes and is possibly redundant. Can the authors revisit these last two sentences to make sure the message is clear.

Line 710: Lightning NOx emissions are not identical in Table 2 for C-GC and C-CC, but they are calculated using the same online parameterization. This should be clearer in this sentence.

Line 715: Make sure there are not two Table 3 references. May be an error in the marked-up version only.

Line 717-718: Is the percentage of NIT about the same between C-GC and S-GC?

Line 723: Is there a reference you can put for the Neu scheme? It is a bit confusing given the Neu and Prather reference after MAM.

Line 736: What is the scheme used in S-GC? It is not referenced earlier in this paragraph. Can the authors remind the readers?

Line 747-749: This kind of introduction of halogens (referring to both bromine and chlorine) may be better suited for the start of Section 4.4 (Line 660) instead of in the start of the bromine section.

Line 759: It looks to me like at 1000 hPa, the difference is exactly if not slightly less than 100%. Is it correct to say "exceeding … by 100%"? Add reference to Figure 12b in the sentence. If Figure 12 is not referenced again, this paragraph only focuses on the result of surface concentrations and this could likely be summed up in a table instead.

Line 762: If Figure 12 is kept, reference Fig 12c in this sentence.

Line 765: This sentence is misleading as we do not have information on CH3Br and OH in Figure 13. If this is a statement of chemistry knowledge, about Bry increasing with height due to this reaction, it should go with Figure 12. I encourage the authors to reconsider the reference to

Figure 13 for the description of Bry profiles (Line 765-771).  Possibly this was a Figure reference that was not updated correctly (line 764) and was intended to go with Figure 12 all along.

Line 800: Could refer back to Figure 12b to quantify this large difference.

Line 804:  In the marked-up copy there is a double "mid-tropospheric mid-tropospheric" and then at the end of the sentence another "mid troposphere".  Can the authors reduce it to just one?

Line 812:  Could remove "vertical" before "profiles".  Cly has been defined a few times.  I am not sure if you need to redefine it here.  The authors did not redefine Bry in line 750.

Line 815:  Add C-GC to this sentence "total Cly for C-GC follows…". Is this true, given there are swings in the % difference with values over 100 % near the surface, then negative, then back to over 100%?  The distribution is about the same from 100 to 2 hPa (Figure 14b).  Also, I do not agree with the "above this pressure, the vertical distribution of C-GC is closer to C-CC", and this statement does not agree with what is stated in Lines 818-819.  The authors need to revisit the discussion of Figure 14.  Would it flow better if the authors followed a similar description as for Figure 12 (i.e., first the extreme differences at the surface and then the rest of the profile)?

Line 831:  Can the authors quote altitude range here for UTLS HCl (e.g., 400 to 100 hPa)?  I am trying to estimate by eye comparing panel c to b and I do not know how far up to the range should be.  Does it go over 50 hPa, given the sentence that follows picks this height out for further discussion?

Line 832:  While this statement is true, my mind went to Figure 14 looking for the percent difference, not from Figure 15.  It would likely be easier on the reader if the 15% came off Figure 14, even if you reference Figure 14a which clearly shows C-CC is greater than C-GC and S-GC.  I wanted to get this number off panel b or c, and that is less straightforward. As for the difference in HCl, this is not shown in a figure like panels b and c of Figure 14.

Line 840:  Add C-GC to this first sentence.  Can the authors link these values to the percent differences shown in Figure 14b and c?

Line 849: MOPITT is on TERRA.

Line 902-903:  Does this sentence refer to Figure 4 or another figure (possibly moved to the supplemental) that compares model to observations somehow spatially so the readers could infer the biases over the Mediterranean Sea and Northern Europe.  And if there is a high bias, how does that line up with the sentence that follows on 907 that the models are biased low.

Line 909:  These numbers could go nicely in Table 9 next to the correlation values.  Might help the reader to follow that the greatest bias is found for the C-GC as quoted in line 916.

Line 930-931:  Can the authors provide panel references in the text here.  I expect this plot is produced by a CAM-Chem post-processing script so it is not something to change as folks familiar with these plots are familiar with this layout, but it is counter-intuitive to me to have the plots nearer the surface at the top and the stratospheric levels in the bottom rows.   This figure is very busy; if the 1, 2, 3, 4 key in the bottom left of each plot could be removed except for say in the bottom panels (j, k, l) that would help where there is overlap with some of the plots.

Line 936:  Can you refer back to Figure 2 where the authors present the zonal mean ozone.  The values at that height all appear to be about the same across the models so it does not surprise me that all three models appear to perform similarly.

Line 947-948:  Is this again a familiar comparison for CAM-Chem users to a climatology for 2004-2010?  I suggest repeating this reason for using this other reference period if it is the case.  Also, could merge these first two sentences together because Figure 20 shows more than just TOC from OMI/MLS.  "Figure 20 shows …<continue as written> …(Ziemke et al., 2011) (panel a) compared to results from C-GC, C-CC, and S-GC (panels b, d, and f, respectively)".

Line 956: Out of curiosity, is the difference in S-GC driven also by the spring Antarctic? Or is there something fundamentally different about the seasonality of the S-GC vs C-GC runs that leads to the 4.5 DU mean bias difference (7.8 vs 3.3)?

Line 958, Figure 20:  The difference color bar labels are a bit odd.  Any way to force it so the labels for white are -20 to 20 DU?  It does not look like you need all the colors in the color bar.  Similarly, Figure 21 difference color bar labels could be improved.

Line 964:  Why only April and not the whole year as in Figure 20?  Is it possible to have the mean values in the top right title label like in Figure 20 instead of the units mol/cm2?

Line 1053:  Is there a reference for MUSICA (publication or website) which could be included here?

---

## Referee Report (RR2)

Fritz et al. 2022 egusphere-2022-226.

I am content that the authors have attempted to address my comments, and have also clarified their aims for the paper in their responses.

---

## Author Response (AR2)

**Dr. Sebastian D. Eastham**
Principal Research Scientist
Joint Program on the Science and Policy of Global Change
Associate Director of the Lab. for Aviation and the Env.

[Figure]

**Massachusetts Institute of Technology**
77 Mass. Ave. Office E19-439F, Cambridge MA 02139, USA
http://globalchange.mit.edu | http://lae.mit.edu
seastham@mit.edu | (617) 253-2170

Editorial Office

Geoscientific Model Development

October 23rd, 2022

Dear Dr. O'Connor,

**Re: submission of "Implementation and evaluation of the GEOS-Chem chemistry module version 13.1.2 within the Community Earth System Model v2.1" to *Geoscientific Model Development***

Thank you for arranging this review of our work. We have used the time since we received our reviews to address the reviewers' comments, and the manuscript has been improved as a consequence. We believe that we have now addressed all comments where possible.

Please find below our responses (in **bold**) to each of the review comments (in *italics*) and our revised manuscript enclosed.

*Referee #1*

*The resubmitted draft describing the new implementation of GEOS-Chem within CESM is much improved over the initial submission. I recommend it for publication after my minor comments and technical edits listed below are considered. The line numbers refer to the marked-up version that followed after the authors' comments to the reviewers and not line numbers in the submitted draft as I did not have the new draft when I started my review and was unfortunately without internet at that time.*

**We thank the reviewer for their time and effort and for their helpful comments. We have done our utmost to respond to each comment in turn (below).**

*Line 55: The authors need to check references throughout the manuscript are in alphabetical or chronological order. Often there is a mix. For example, "Community Earth System Model (CESM) (Hurrell et al., 2013; Tilmes et al., 2016; Lamarque et al., 2012; Emmons et al., 2020)"*

**References have now been changed to chronological ordering throughout.**

*Line 64 : extra space at the end of sentence before the period.*

**This has now been corrected.**

*Line 82: GEOS FP does not have a hyphen ( see https://gmao.gsfc.nasa.gov/GMAO_products/NRT_products.php) and this should be corrected throughout the manuscript.*

**GEOS FP is now written without a hyphen throughout the manuscript.**

*Line 83-85: GEOS FP and MERRA-2 should have references to Lucchesi 2018 and Gelaro et al. 2017 (https://doi.org/10.1175/JCLI-D-16-0758.1), respectively.*

**These references have now been added.**

*Also, is there an extra space after output and before provides? ("output provides data")*

**The extra space has been removed.**

*Line 87: Should ESM be redefined, or is it assumed folks know it from the CESM definition?*

*ESM is now defined as Earth system model on first use (line 48).*

*Line 99: The Keller et al. 2017 reference is incomplete. My assumption is it is redundant to the Hu et al reference so the Keller et al. 2017 should be removed and only the Keller et al 2021 reference should be included.*

**The reference in question now includes only Keller et al. (2021).**

*Line 101: Goddard Earth Observing System (GEOS) (Bey et al., 2001; Eastham et al., 2018) is redundant (to text above as GEOS already defined) and confusing (as readers could mistake Bey and Eastham references for GEOS meteorology references). I suggest removing references and "Goddard Earth Observing System".*

**The references have been removed, as has the re-definition of GEOS.**

*Line 105: CESM already defined and references provided.*

**The re-definition and references have been removed.**

*Line 107: "observed meteorology including from GEOS", I suggest changing observed to analyzed, unless you can actually nudge towards observations.*

**This has been changed to "analyzed" (line 107).**

*Line 119: Earth system models is written out instead of using "ESM"*

**We now write "ESM" (line 118).**

*Line 141: "Sections 4 and 5 present a one-year simulation (not including spin up)". I know I suggested to make it clear that there was a spin-up vs the year used for evaluation in my initial review but I think this is confusing as it is written, such that one might wonder if no spin-up was included. Instead, maybe something like "Results from the one-year simulation (following the appropriate spin-up) performed for each model configuration 1) CESM with GEOS-Chem, 2) CESM with CAM-chem, and 3) the stand-alone GEOS-Chem CTM are presented with includes model intercomparison in Section 4 and evaluation against surface and satellite measurements in Section 5. Final discussion and conclusions are in Section 6.". Note, I restructured the sentence and included a final sentence for Section 6.*

**We agree and have adopted the proposed changes in full (lines 139-142).**

*Line 182: Should HEMCO have a Keller et al. reference? Line 301 redefines HEMCO and provides the Keller et al. 2014 reference.*

**We now include Keller et al. (2014) on line 181.**

*Line 202: "OH, ozone and NO3". OH and NO3 should be defined. OH is defined later on line 486.*

**Both OH and NO3 are now defined on line 201.**

*Line 206,229: The unit has a period after gram instead of a space.*

**The erroneous period has now been replaced with a space.**

*Line 249: MAM4 already defined so not necessary to do here.*

**We now simply state on line 249 that CESM uses MAM4 with a minor edit to make it clear that the 4 in MAM4 refers to the number of modes (lines 248-250).**

*Line 315: NOx is defined as NO+NO2 but then NO is used a few lines later without being first defined as nitric oxide.*

**NO and $NO_2$ are now defined as nitric oxide and nitrogen dioxide on first use (lines 312-313).**

*Line 322: Sulfur dioxide not defined first before SO2.*

*SO₂ now defined as sulfur dioxide on first use (line 321).*

*Line 331: Should this say "since iodine chemistry is not explicitly modeled" or do the authors mean only "iodine"*

**We now state that "iodine species and chemistry" are not explicitly modeled (line 330).**

*Line 346: "(methane, N2O, and chlorofluorocarbons)". When are these used again and if so should they be defined? N2O is not defined yet. CFCs defined at line 478.*

**All three are now defined here (lines 345-346). Although we do not discuss methane or N₂O further, we felt it was helpful to include the chemical formulae since both (especially N₂O) are so frequently discussed in such terms.**

*Line 377: The previous paragraph also begins with "Although". I suggest changing one of these to an alternative word, like "While" (or "Whilst" if you prefer the British English)*

**We chose to go with "While" (although I like the sound of "Whilst", I must bite my tongue and recognize that I am in a US institution and have otherwise used American English).**

*Line 387: "Then, we perform a comparison of its output to that generated by two other model configurations (Section 4)". I suggest adding something to this sentence so it reads to the effect of "the two conventional configurations for CESM-chem and GEOS-chem". Otherwise, the "other model configurations" is ambiguous as the next sentences states "by comparing these results to C-CC", it is unclear if the "results" is just the C-GC or C-GC plus these two other model configurations.*

**We have adopted this change as suggested in full (lines 386-388).**

*Line 398: I suggest changing "data" to "output" as this is from model simulations, and not observations.*

**We now refer to "model output" instead (line 397).**

*Does "This section" in Line 399 refer to Section 3 or Section 5?*

**The reference was to Section 3, which is now stated explicitly on line 398.**

*Line 402: I suggest changing "All simulations cover January 1st to December 31st 2016, with an additional 6 months (S-GC) or 1 year (C-GC/C-CC) of spin up" to "Following a model spin-up period (6 months for S-GC and 1 year for C-GC and C-CC), the 1-year period of January 1 to December 31, 2016 was suitable for multi-model evaluation."*

**We have changed the lines in question (401-402) to read "Following a model spin-up period (6 months for S-GC, 1 year for C-GC and C-CC), the one-year period of January 1st to December 31st 2016 is simulated and used for multi-model evaluation".**

*Line 415: could state this is the stratopause*

**We have adopted this change (line 415).**

*Line 423: Are the truncated levels the first 56 layers of the GEOS grid and then are they somehow regridded to match the 56 hybrid pressure levels of CESM? Can the authors clarify this?*

**We now clarify (lines 423-424) that the "upper 16 layers from MERRA-2 are removed, leaving a truncated 56-layer vertical grid which is used unmodified by CAM6".**

*Line 438: CEDS should be defined and reference provided, especially as a new version of CEDS was recently released.*

**CEDS is defined previously on line 320 and includes the relevant reference. We now repeat the reference on line 439.**

*Line 453: Is there a reference for AeroCom?*

**Thank you for catching this oversight! We now reference the relevant source (line 455).**

*Line 506: What is the pressure range in Figure 2? It is hard to know from the y-axis labels. Some of the other vertical plots have the top of the y-axis given in the Figure caption. Check this is given in all captions.*

**The range is 300 hPa to 1.65 hPa (the model top for C-CC and C-GC). This is now specified in the caption (line 510). A similar specification is also now provided for all other figures.**

*Line 515: Remove "hydroxyl radical" as OH already defined.*

**This redundant definition has been removed.**

*Line 531: Should "ozone" be clarified to be tropospheric ozone or UTLS or which levels did the Park et al study look at?*

**Park et al. (2021) were focused on the surface and free troposphere, so this has been modified to state "tropospheric ozone" (line 532).**

*Line 546-557: I am struggling with the structure of these two paragraphs. The bigger differences are in panels b and i, yet this first paragraph focuses on (h) and then the second on (i). I suggest reconsidering the flow of the paragraphs to something like:*

*There is a clear link between the ozone distributions and water vapor. Outside of the tropics and below the tropopause (Figure 3i), water vapor concentrations are up to 30 % greater in C-GC than in S-GC…<continue as written>...(an indirect sink for ozone). While ozone concentrations are uniformly lower for C-GC than for C-CC (Figure 3c), water vapor concentrations are uniformly greater for C-GC than for C-CC (Figure 3d). This is not surprising since the representation of moist physics in the two models is identical. However…<continue as written>...HNO3. Differences in ozone related to tropospheric NOy and halogens will be explained in detail in Section 4.4.*

**We have attempted to restructure this section following the reviewer's advice. (lines 548 to 577).**

*Lines 565-569: Is it differences in moist processes or transport?*

**Since both could be relevant, we now say "since the representations of transport and tropospheric moist physics in the two models are identical" (lines 558-559).**

*Line 576-577: I am struggling to see this link.*

**The original text read "…in the Northern mid- and upper latitudes below 900 hPa, OH concentrations are 10-20% greater in C-GC than in S-GC. This reflects the greater water vapor concentrations and roughly equal ozone concentrations between the two models". We now clarify that we are referring to Figure 3e on line 567.**

*Line 590-595: What do the authors mean by "does not show the same hemispheric asymmetry in absolute terms"? The difference over the southern hemisphere is about the same, it is really the northern hemisphere that is different. Is the emission of bromine from sea salt limited to either hemisphere or could it be driving the differences seen in the arctic which looks bluer to me in panel c than panel b. In the northern hemisphere, the C-GC and S-GC are fairly close (2.2 ppbv difference). Are the anthropogenic emissions driving the differences in the northern hemisphere or the southern hemisphere?*

**We have rephrased this to instead say that "The comparison between C-GC and C-CC (panel c) shows a similar difference in Southern Hemispheric ozone over oceans, but the relative difference now also extends to Northern Hemispheric oceans." (lines 591-592). After a brief discussion of the role of bromine and biogenic emissions (lines 593-597), we now also state that differences in the response to anthropogenic emissions may play a role in the different Northern Hemispheric ozone concentraitons (lines 597-598).**

*Line 602-604: This hot spot is also seen in both C-CC v C-GC plots. So is it almost more remarkable that January S-GC v C-GC does not have it? I wanted to link it to the hot spot over the amazon, but I got my sign wrong. Is there a clear feature in the C-GC minus C-CC in Figure 4 that is consistent all year round that could be linked to this hot*

*spot in the bottom panel of Figure 5 that could partially explain July top panel? (if not, then this falls under the need for future work).*

**Based on additional review, we agree that the July hot spot is present in both C-CC and S-GC, and now state as much in the text (line 606). However, we believe that the feature in the January C-CC vs C-GC comparison is distinct, and not necessarily related. This feature is now commented on at the end of the same paragraph (lines 607-608).**

*Line 626: reference Figure 7 when claiming the differences are greatest over the oceans, as this cannot be determined in Figure 6.*

**We now do so (line 630).**

*Line 626-628: The link to OH differences, is that still for below 800 hPa or throughout the troposphere? I do not see a clear link between Figure 3f and Figure 6c regarding the higher sulfate aerosol mass in the southern latitudes, but I can see patterns that match the OH zonal plots elsewhere in the troposphere.*

**We agree that the link between Figures 3f and 6c is not especially clear, and that OH alone is unlikely to explain this discrepancy. However, we had overlooked the possibility of differences due to in-cloud sulfur processing, which is handled differently in the C-GC/S-GC compared to C-CC. We now mention this possibility on line 633 and refer the reader to the original description in Section 2.3.1.**

*Line 629: Can you quantify the "more closely follows"? I think this is the differences of +/- 25% quoted at the end of the paragraph but I think it should be quoted earlier. Can you give a reason by the C-GC and S-GC differ so much? Is it the difference in the bulk representation vs summing across all aerosol size bins? Oh, based on my latter reading of line 654 saying it is the convective scavenging, I think you may need to check here in line 630 if this should be broken into two sentences or at least properly reference (b) and (c) as I missed here the authors were referring to the 'differences in the representation of convective scavenging" was now discussing panel (b) and not still panel (c).*

**We have moved the quantification (+/-25%) to immediately follow the statement that C-GC more closely follows C-CC, and also direct the reader to the panel c. The sentence in question has been split into two as suggested, and panel b is now indicated in the discussion of the differences between C-GC and S-GC (lines 634-637).**

*Line 644: In the previous paragraph there was a link to DMS for these greater concentrations and OH. Does that not apply here?*

**It does apply, but we have not been able to definitively identify the root causes of differences. Since we already mention several possible causes for differences in the prior paragraph, we chose not to repeat those explanations here.**

*Line 655-657: it is not clear to me why POM would be greater in C-CC if the emissions of POM in C-CC are 29% lower.*

**This was an oversight, and we thank the reviewer for catching it. We now state (lines 661-663) "[a] lthough emissions of POM in C-CC are 29% lower they occur as accumulation-mode rather than primary organic mode aerosol, which may extend their lifetime".**

*Line 674: except the two points just above 50% (as mentioned in line 677). May be best to change your upper bound to match or give the range (looks like about -25% to 55%).*

**The range has been updated to read "-26 to +55%" (line 683).**

*Line 688-691: Is the difference at 10 hPa really significant? At 200 hPa, the 60 and 63% are also only 3% different and read like they are about the same, but the striking difference is the much greater 78% in C-CC at 200 hPa. Is it fair to open the paragraph with "partitioning between NOx and HNO3 differs significantly between the three models"? Is there a point where the difference becomes significant?*

**We agree that the differences in partitioning are predominantly between the GC and CC-based models. 10 hPa is highlighted because at this pressure the total $NO_y$ is consistent between C-CC and C-GC, but the**

**partitioning is more consistent between S-GC and C-GC. We have re-written the paragraph opening to highlight this (lines 697-699).**

*Line 693: Can this statement on sulfate aerosol size distribution be linked to Figure 6 at all?*

**Our intent in this statement ("the CESM-driven simulation includes a more detailed representation of the sulfate aerosol size distribution") is to highlight that the explicit size distribution representation may affect the rates of hydrolysis. Unfortunately Figure 6 shows only total sulfate aerosol mass, so we could not draw a clear link to this statement. However, we do now draw the reader's attention back to Figure 6 on lines 703-704.**

*Line 704: At 500 hPa, I cannot see any NOx in the figures. Given the previous sentences say that from 200-900 hPa HNO3 and PAN are dominant and the following sentence says NOx once again becomes significant contributor, is there much NOx contributing at 500 hPa that it is worth listing it here in the 78%, 85% and 97%? Or is this to be consistent with what is in line 702?*

**We include $NO_x$ for consistency with the prior statement (previously line 702, now line 710) as surmised by the reviewer. To try and make this clearer we now say "the same combination of species ($NO_x$, $HNO_3$, PAN)" (lines 712-713).**

*Line 706-707: This sentence is confusing as it opens talking about lower altitudes but then ends referencing 200 to 300 hPa and then the new text again refers to lower altitudes and is possibly redundant. Can the authors revisit these last two sentences to make sure the message is clear.*

**On review these final two sentences seem redundant, and as such have been removed.**

*Line 710: Lightning NOx emissions are not identical in Table 2 for C-GC and C-CC, but they are calculated using the same online parameterization. This should be clearer in this sentence.*

**We now state instead that the same parameterization is used (line 720).**

*Line 715: Make sure there are not two Table 3 references. May be an error in the marked-up version only.*

**The redundant reference has been removed.**

*Line 717-718: Is the percentage of NIT about the same between C-GC and S-GC?*

**Similar but not identical. We now state (lines 728-729) that "[t]he ratio of nitrate in aerosol compare to in gaseous HNO3 is similar at low altitudes (below 900 hPa) between C-GC and S-GC, with nitrate mixing ratios being lower than HNO3 at 900 hPa but greater than HNO3 at the surface".**

*Line 723: Is there a reference you can put for the Neu scheme? It is a bit confusing given the Neu and Prather reference after MAM.*

**We now cite Neu and Prather (2012) immediately after "Neu scheme" instead of after "MAM4" (line 734). We also state on the first usage of "Neu scheme" that this is the name most used for the scheme described by Neu and Prather (2012) (line 194).**

*Line 736: What is the scheme used in S-GC? It is not referenced earlier in this paragraph. Can the authors remind the readers?*

**We now cite Luo et al. (2001) again on line 747.**

*Line 747-749: This kind of introduction of halogens (referring to both bromine and chlorine) may be better suited for the start of Section 4.4 (Line 660) instead of in the start of the bromine section.*

**We have moved this text accordingly (now lines 666-668).**

**We respond to the following three reviewer comments collectively:**

*Line 759: It looks to me like at 1000 hPa, the difference is exactly if not slightly less than 100%. Is it correct to say "exceeding … by 100%"? Add reference to Figure 12b in the sentence. If Figure 12 is not referenced again, this paragraph only focuses on the result of surface concentrations and this could likely be summed up in a table instead. Line 762: If Figure 12 is kept, reference Fig 12c in this sentence.*
*Line 765: This sentence is misleading as we do not have information on CH3Br and OH in Figure 13. If this is a statement of chemistry knowledge, about Bry increasing with height due to this reaction, it should go with Figure 12. I encourage the authors to reconsider the reference to Figure 13 for the description of Bry profiles (Line 765-771). Possibly this was a Figure reference that was not updated correctly (line 764) and was intended to go with Figure 12 all along.*

**We agree both that Figure 12 was under-utilized and that the initial discussion of $CH_3Br$ belonged with Figure 12 rather than Figure 13. We have extended the discussion of Figure 12 (lines 767-773), more clearly referencing the role of OH. We believe that Figure 12 is also more appropriate for the discussion of total $Br_y$ than Figure 13, and have moved the discussion of variation with altitude to this location.**

*Line 800: Could refer back to Figure 12b to quantify this large difference.*

**We now do so (lines 810-811).**

*Line 804: In the marked-up copy there is a double "mid-tropospheric mid-tropospheric" and then at the end of the sentence another "mid troposphere". Can the authors reduce it to just one?*

**Done (line 815).**

*Line 812: Could remove "vertical" before "profiles". Cly has been defined a few times. I am not sure if you need to redefine it here. The authors did not redefine Bry in line 750.*

**We now simply state "[a]nnually-averaged profiles of $Cl_y$" (lines 821-822).**

*Line 815: Add C-GC to this sentence "total Cly for C-GC follows…". Is this true, given there are swings in the % difference with values over 100 % near the surface, then negative, then back to over 100%? The distribution is about the same from 100 to 2 hPa (Figure 14b).*

**We agree that "the same" was too strong. We now instead state that "Comparing C-GC to S-GC (Figure 14b), differences in total $Cl_y$ follow a similar pattern to $Br_y$ (Figure 12b) up to 10 hPa" (line 825).**

*Also, I do not agree with the "above this pressure, the vertical distribution of C-GC is closer to C-CC", and this statement does not agree with what is stated in Lines 818-819. The authors need to revisit the discussion of Figure 14. Would it flow better if the authors followed a similar description as for Figure 12 (i.e., first the extreme differences at the surface and then the rest of the profile)?*

**This statement was inaccurate – our intent had been to point out that the relative difference in $Cl_y$ above 10 hPa was stable between C-GC and C-CC, but increasing when comparing C-GC to S-GC. We have followed the reviewer's advice and restructured this paragraph, doing away with the misleading initial summary and instead following the same structure as was used when describing Figure 12 (lines 825-830).**

*Line 831: Can the authors quote altitude range here for UTLS HCl (e.g., 400 to 100 hPa)? I am trying to estimate by eye comparing panel c to b and I do not know how far up to the range should be. Does it go over 50 hPa, given the sentence that follows picks this height out for further discussion?*

**We now state "specifically, between 200 and 50 hPa" (line 843). Although the comment is true at lower altitudes also, this range is relatively clear in Figure 15.**

*Line 832: While this statement is true, my mind went to Figure 14 looking for the percent difference, not from Figure 15. It would likely be easier on the reader if the 15% came off Figure 14, even if you reference Figure 14a which clearly shows C-CC is greater than C-GC and SGC. I wanted to get this number off panel b or c, and that is less straightforward. As for the difference in HCl, this is not shown in a figure like panels b and c of Figure 14.*

**We now reference Figure 14a in this location (now line 844).**

*Line 840: Add C-GC to this first sentence. Can the authors link these values to the percent differences shown in Figure 14b and c?*

**We now specify C-GC (line 851). We chose not to reference Figure 14 or 15, as the contribution of Cl atoms to total Cl$_y$ is negligible.**

*Line 849: MOPITT is on TERRA.*

**Thank you for catching this! We now state "Terra MOPITT" (line 860).**

*Line 902-903: Does this sentence refer to Figure 4 or another figure (possibly moved to the supplemental) that compares model to observations somehow spatially so the readers could infer the biases over the Mediterranean Sea and Northern Europe. And if there is a high bias, how does that line up with the sentence that follows on 907 that the models are biased low.*

**We realized that this wording was confusing; we had intended only to say that geographical patterns were consistent. When we stated that ozone was "high" over the Mediterranean and "low" over Northern Europe, this referred to value rather than bias. We have removed the sentence in question as it did not add to the discussion. Instead, we now refer readers to Figure 4, which shows the surface ozone in C-GC and the differences when comparing to the other two configurations (lines 914-916).**

*Line 909: These numbers could go nicely in Table 9 next to the correlation values. Might help the reader to follow that the greatest bias is found for the C-GC as quoted in line 916.*

**We agree and have moved these numbers into Table 9.**

*Line 930-931: Can the authors provide panel references in the text here. I expect this plot is produced by a CAM-Chem post-processing script so it is not something to change as folks familiar with these plots are familiar with this layout, but it is counter-intuitive to me to have the plots nearer the surface at the top and the stratospheric levels in the bottom rows. This figure is very busy; if the 1, 2, 3, 4 key in the bottom left of each plot could be removed except for say in the bottom panels (j, k, l) that would help where there is overlap with some of the plots.*

**Unfortunately we have limited control over the plot, as it is (as the reviewer correctly surmised) a standard output from CAM-chem post-processing software. However, we now explicitly state which panel we are referring to in each location (lines 937-947).**

*Line 936: Can you refer back to Figure 2 where the authors present the zonal mean ozone. The values at that height all appear to be about the same across the models so it does not surprise me that all three models appear to perform similarly.*

**We now do so (lines 956-957).**

*Line 947-948: Is this again a familiar comparison for CAM-Chem users to a climatology for 2004-2010? I suggest repeating this reason for using this other reference period if it is the case.*

**We now state that this is a standard output from CAM-chem processing software (line 958).**

*Also, could merge these first two sentences together because Figure 20 shows more than just TOC from OMI/MLS. "Figure 20 shows …<continue as written> …(Ziemke et al., 2011) (panel a) compared to results from C-GC, C-CC, and S-GC (panels b, d, and f, respectively)".*

**These sentences have been merged to ensure that it is clear to the reader what is being shown (lines 956-958).**

*Line 956: Out of curiosity, is the difference in S-GC driven also by the spring Antarctic? Or is there something fundamentally different about the seasonality of the S-GC vs C-GC runs that leads to the 4.5 DU mean bias difference (7.8 vs 3.3)?*

**We are not sure of the drivers behind the differences, but suspect that behavior at the South Pole may indeed be relevant. To this end we now refer the reader to the SI, where we explore the differences in Antarctic ozone depletion between C-GC and S-GC (lines 966-967).**

*Line 958, Figure 20: The difference color bar labels are a bit odd. Any way to force it so the labels for white are -20 to 20 DU? It does not look like you need all the colors in the color bar. Similarly, Figure 21 difference color bar labels could be improved.*

**Unfortunately we are again constrained by the CAM-chem postprocessing software, which automatically picks the color bar and range. As such, we have left this as-is.**

*Line 964: Why only April and not the whole year as in Figure 20? Is it possible to have the mean values in the top right title label like in Figure 20 instead of the units mol/cm2?*

**Again we are showing a standard output in this case which does not generate an estimate of the mean bias. Although analyses are produced for four different months (January, April, July, and October) we concluded that it would be unwieldy to show all possibly combinations and therefore only show a single month.**

*Line 1053: Is there a reference for MUSICA (publication or website) which could be included here?*

**Yes – we now cite Pfister et al. (2020) on line 1064.**

*Referee #2*

*I am content that the authors have attempted to address my comments, and have also clarified their aims for the paper in their responses.*

**We thank the reviewer for their time and effort.**

We would again like to thank the reviewers for their time and insight, and believe that their input during this review process has improved the paper substantially. Thank you again for considering our manuscript for publication in *Geoscientific Model Development.*

Regards,

Sebastian Eastham